# The influence of soil properties and nutrients on conifer forest growth in Sweden, and the first steps in developing a nutrient availability metric

Kevin Van Sundert[1], Joanna A. Horemans[1], Johan Stendahl[2], and Sara Vicca[1]

[1]Centre of Excellence PLECO (Plants and Ecosystems), Biology Department, University of Antwerp, Wilrijk, 2610, Belgium
[2]Department of Soil and Environment, Swedish University of Agricultural Sciences, P.O. Box 7014, Uppsala, 75007, Sweden

*Correspondence to*: Kevin Van Sundert (Kevin.VanSundert@uantwerpen.be)

**Abstract.** The availability of nutrients is one of the factors that regulate terrestrial carbon cycling and modify ecosystem responses to environmental changes. Nonetheless, nutrient availability is often overlooked in climate-carbon cycle studies because it depends on the interplay of various soil factors that would ideally be comprised into one (in the best case globally applicable) metric. Such a metric does not currently exist. Here, we use a Swedish forest inventory database that contains soil data and tree growth data for > 2500 forests across Sweden to test which combination of soil factors best explains variation in tree growth, and to take the first steps in developing a nutrient availability metric. Specifically, we aimed to provide a flexible, and hence updateable nutrient metric, applicable for Sweden and with potential for elaboration to other regions and biomes. Our analyses revealed that SOC and the ratio of soil carbon to nitrogen (C:N) were the most important factors explaining variation in "normalized" (climate-independent) productivity across Sweden. Normalized productivity was significantly negatively related to soil C:N ratio ($R^2 = 0.02$–$0.13$), while SOC exhibited an empirical optimum ($R^2 = 0.05$–$0.15$). For the metric, we started from a (yet unvalidated) metric for constraints on nutrient availability that was previously developed by IIASA (Laxenburg, Austria). This IIASA-metric – initially developed for evaluating potential productivity of arable land – consists of soil properties that are indicative of nutrient availability and is based on theoretical considerations that are also generally valid for non-agricultural ecosystems. Nonetheless, the IIASA-metric was unrelated to normalized productivity across Sweden ($R^2 = 0.00$–$0.01$), because the soil factors under consideration were not optimally implemented, and because the C:N ratio was not included. Using two methods (each one based on a different way of normalizing productivity for climate), we upgraded this metric by incorporating soil C:N and adjusting the relationship between SOC and nutrient availability in view of the observed relationships across our database. In contrast to the IIASA-metric, the upgraded metrics explained some variation in normalized productivity in the database ($R^2 = 0.03$–$0.21$; depending on the applied method). A test for five randomly selected local fertility gradients in our database revealed a significant and stronger relationship between the upgraded metrics and productivity for each of the gradients ($R^2 = 0.09$–$0.38$). The upgraded nutrient availability metrics thus open up new opportunities for further validations and improvements with other datasets, from forests and from other ecosystem types, to develop a metric that enables comparisons of nutrient availability at large spatial scales.

## 1 Introduction

Nutrients determine structure and functioning at all levels of biological organization. The availability of mineral elements influences for example plant growth (von Liebig, 1840), patterns of biodiversity (Fraser et al., 2015) and ecosystem processes (e.g. Janssens et al., 2010; Vicca et al., 2012; Fernández-Martínez et al., 2014). Moreover, nutrient availability can modify ecosystem responses to global atmospheric and climatic changes, such as nitrogen (N) deposition (From et al., 2016), increasing $CO_2$ levels (Norby et al., 2010; Terrer et al., 2016), warming (Dieleman et al., 2012) and drought (Friedrich et al., 2012). Given the crucial role of nutrients in terrestrial carbon cycling and in shaping the magnitude and direction of its feedbacks to climate change, nutrient availability should be taken into account in global analyses and in Earth system models (Goll et al., 2012; Thomas et al., 2015; Wieder et al., 2015). This is, however, not yet common practice because we often lack the soil data and metrics needed to accurately account for nutrient availability.

Comparing nutrient availability among terrestrial ecosystems is thus difficult for two reasons: comprehensive and harmonized data on soil properties and nutrients are not usually available from experimental and observational sites, and no standardized quantitative metric exists to compare the nutrient statuses of terrestrial ecosystems at the global scale, or even at a national scale (e.g. for Sweden, which is considered in the present paper). In the absence of a standardized nutrient availability metric, studies comparing nutrient availability across sites have previously described soil fertility related approximations such as the height of 100 year old trees (which, however, also depends on other factors such as soil depth and hydrology - Hägglund and Lundmark, 1977) or have manually classified sites as low, medium, and high nutrient availability based on existing site information (Vicca et al., 2012; Fernández-Martínez et al., 2014). The absence of a more nuanced expression impedes elucidating the role of nutrient availability in ecosystem processes and functioning (Cleveland et al., 2011) and how these respond to global change, and precludes investigating non-linear effects of nutrient availability.

Although various proxies exist to estimate soil N and phosphorus (P) availability at the local scale (e.g. "snapshots" of extractable pools), no perfect method exists to quantify N and P availability in a comparable way across ecosystems (Binkley and Hart, 1989; Holford, 1997; Neyroud and Lischer, 2003), which limits the potential for inter-site comparisons based on these data alone (Cleveland et al., 2011). Soil properties like soil texture, soil organic matter (SOM) quantity and quality, and pH, on the other hand, are more indicative of the general nutrient status, because together with environmental factors (temperature and moisture - Binkley and Hart, 1989), they control the size of the soil solution pool, exchange sites and unavailable soil pools and fluxes between them over longer time scales (Roy et al., 2006). For instance, a high clay fraction corresponds to a high cation exchange capacity (CEC), i.e. the soil's potential to retain positively charged, exchangeable ions such as $NH_4^+$, $K^+$, $Ca^{2+}$ and $Mg^{2+}$ (Chapman, 1982; Chapin et al., 2002), while SOM has a positive influence on nutrient availability by acting as a nutrient reserve (Grand and Lavkulich, 2015) and provides cation as well as anion exchange sites (IIASA and FAO, 2012). Finally, soil pH strongly influences availability of P and base cations ($K^+$, $Ca^{2+}$ and $Mg^{2+}$). At low pH, P is bound to Fe and Al oxides, while at high pH, P is typically unavailable because of complex formation with Ca. P availability is thus maximal at intermediate pH (Chapin et al., 2002; Bol et al., 2016), while enhanced leaching of base cations

occurs in acidic soils, thus lowering the amount of total exchangeable bases (TEB = cation equivalent of summed K, Ca, Mg and Na - IIASA and FAO, 2012). Hence, unlike temperature or precipitation, nutrient availability cannot be assessed by measuring one single parameter. It is determined by the interplay of various nutrients and soil properties. A nutrient availability metric should thus combine critical soil properties and nutrients, while considering important non-linearities. To be widely applicable, such a metric is preferably constructed only of easy-to-obtain variables.

Only a few exploratory attempts to find an expression for nutrient availability at the global scale have been made. The most recent one was developed by the International Institute for Applied Systems Analysis (IIASA, Laxenburg, Austria) and FAO, who provide a simple index in their Global Agro-ecological Zones report of 2012 (IIASA and FAO, 2012). It is a worldwide applicable metric for constraints on nutrient availability, principally meant for agricultural purposes. This metric represents, for a particular crop species, the percentage of the maximum attainable productivity that could be reached given constraints imposed by environmental characteristics such as climate, rooting conditions and soil oxygen availability, but absent nutrient limitation:

$$\text{Actual productivity} = \frac{\text{Metric score [\%] x Attainable productivity}}{100} \text{ ,} \qquad (1)$$

The species-specific score of the metric depends on four measurable soil variables, related to soil fertility: soil organic carbon concentration (SOC - %), texture, total exchangeable bases (TEB - $cmol_+$ $kg^{-1}$ dw) and pH measured in water ($pH_w$). The metric score combines the scores of each of these four attributes (provided in a look-up table), but giving more weight to the attribute with the lowest score. Together with the non-linear relationships (e.g. for pH and SOC - see Methods), this increases the realism of the metric.

To our knowledge, the accuracy of the IIASA-metric has not yet been tested against data from natural ecosystems, and it is not known to what extent the metric – aimed at describing constraints on nutrient availability – can describe variation in nutrient availability of non-agricultural soils. Evaluation of the IIASA-metric, and further development of a widely applicable metric of nutrient availability, requires datasets that combine the necessary information on soil properties and nutrients with data on plant productivity, while also covering a substantial variation in nutrient availability. Such a unique dataset – that comprises > 2500 conifer forest plots and thus provides sufficient statistical power for an evaluation of the metric – is provided by the Swedish forest inventory service. Moreover, it contains an additional variables of interest related to N availability, such as soil total N stock and concentration, and especially the soil C:N ratio, which we expected to be an important factor in explaining variation in nutrient availability. In this paper, we present two upgrades of the original IIASA-metric, that can describe some of the variation in nutrient availability in Sweden, and that can be further evaluated and upgraded in the future with other datasets to test and improve its performance also outside Sweden.

Specifically, we used the Swedish dataset to address the following questions:

*Question 1:* which single soil variables can explain variation in normalized (i.e. climate-independent) productivity across Sweden? Which combination of soil factors best explains variation in normalized productivity?

*Question 2:* can the IIASA-metric of constraints on nutrient availability explain variation in normalized productivity? Are the soil variables already included in the metric (SOC, texture, TEB and $pH_w$) accurately implemented?

*Question 3:* can the IIASA-metric be upgraded to characterize nutrient availability in Swedish forests?

**2 Methods**

**2.1 The Swedish forest and soil inventories (national database)**

We combined a Swedish forest soil (Olsson, 1999; Lundin, 2011) and inventory database for the period 2003–2012 (Lundin, 2011) with a database with soil texture and climate information across Sweden. Precipitation data were extracted from the European Commission Joint Research Centre Monitoring Agricultural Resources dataset (EC–JRC–MARS, based on ECMWF

model outputs and a reanalysis of ERA–Interim; see http://spirits.jrc.ec.europa.eu/), based on the geographic location of each site. The dataset's spatial resolution is 0.25° and averages were calculated for the period 1989–2012. The resulting data collection thus incorporated information on location, climate, soil and vegetation for about 2500 forested plots ($n = 1099$ for spruce, $n = 1422$ for pine), spread over Sweden (Table 1).

Many of the (mostly managed) forest plots were not monocultures, but contained both Norway spruce (*Picea abies* (L.) H.

Karst.) and Scots (or Lodgepole) pine (*Pinus sylvestris* L. or *Pinus contorta* Douglas) trees, as well as other species. In order to contrast spruce and pine forests, we classified forests with ≥ 50 % basal area of spruce (pine) trees as spruce (pine). To quantify the influence of climate on productivity across Sweden (*question 1*), we first determined the annual growing season temperature sum (TSUM) following Odin et al. (1983) (www.kunskapdirekt.se):

TSUM [°C days]

= 4203.212488 - 40.21083 * latitude [° N] - 2.564434 * elevation [m]

   + 0.030492 * latitude [° N] * elevation [m] - 0.117532 * latitude² [° N] + 0.00188 * elevation² [m]

   – 0.000000556 * latitude² [° N] * elevation² [m] ,                     (2)

In order to facilitate between-site comparisons and to allow calculating the nutrient availability metric, we converted the soil measurements (SOC, texture, TEB, $pH_w$, $pH_{KCl}$, total nitrogen and C:N ratio) taken per horizon to values representative of the

120 upper 10 cm (i.e. the 0–10 cm layer) and the upper 20 cm (i.e. the 0–20 cm layer). To this end, we first calculated bulk densities (BD) as

$$BD_{organic\ horizon}\ [kg\ m^{-3}] = \frac{humus\ stock\ [kg/m^2]}{humus\ depth\ [m]}\ ,$$                     (3)

for the organic horizons and

$$BD_{mineral\ horizon}\ [kg\ m^{-3}] = 1546.3 * exp(-0.3130 * \sqrt{total\ carbon\ [\%]})\ , \tag{4}$$

for the mineral soil (Nilsson and Lundin, 2006).

Conversions of soil data ("variables") per horizon to data per depth interval (layer x–y cm) were then performed as follows (soil mass per m² [kg m$^{-2}$] = BD [kg m$^{-3}$] * thickness$_{horizon\ or\ layer}$ [m]):

$$Variable_{x-ycm} = (soil\ mass_{horizon1}/soil\ mass_{x-ycm}) * variable_{horizon1}$$
$$+ (soil\ mass_{horizon2}/soil\ mass_{x-ycm}) * variable_{horizon2} + \ldots \tag{5}$$

The IIASA-metric of constraints on nutrient availability, originally meant for use on arable land, incorporates four crop specific scores (estimated for SOC, texture, TEB and pH$_w$) that can be assigned to a soil (IIASA and FAO, 2012). These scores, which can be found in look-up tables (http://webarchive.iiasa.ac.at/Research/LUC/GAEZv3.0/soil_evaluation.html), were derived from crop growth data on different agricultural soils. Given that we were analyzing boreal forests and not crops, we averaged the scores of the different crop species for each of the four soil properties. As such, we thus removed crop specific requirements, but generally known relationships between the soil variables and plant performance (not only valid for agro-ecosystems), such as an optimum for pH, remained. In addition, we replaced the look-up table derived step functions by continuous empirical formulas, to facilitate their calculation as well as their modification (Fig. 1):

$$SOC\ Score\ [\%] = 38.94 + (100 - 38.94) * (1-exp(-1.4192 * SOC[\%]))\ , \tag{6}$$

$$Texture\ Score\ [\%] = max(100 + 0.4911 * (1-exp(0.0522 * SAND[\%])), 35)\ , \tag{7}$$

$$TEB\ Score\ [\%] = 28.05 + (100 - 28.05) * (1-exp(-0.4508 * TEB[cmol_+\ kg^{-1}]))\ , \tag{8}$$

$$pH\ Score\ [\%] = max(-17.228 * (pH_w - 4.04) * (pH_w - 8.84), 0)$$
$$= max(-17.228 * (pH_w - 6.44)^2 + 99.32, 0)\ , \tag{9}$$

The total score for nutrient availability, which can be interpreted as the expected actual yield (i.e. aboveground productivity) proportional to the maximum attainable yield (i.e. without nutrient constraints), was then calculated as follows (IIASA and FAO, 2012):

$$Total\ IIASA\ Score\ [\%] = 0.5 * Lowest\ Score + 0.5 * Average\ of\ other\ Scores\ , \tag{10}$$

## 2.2 General approach

Forest productivity across Sweden depends not only on nutrient availability, but also on climate. Before evaluating the metric, we removed the influence of climate on forest productivity ("PRE" in Fig. 2). Normalized productivity was calculated in two alternative ways: (1) as the residuals of the regression model (of PRE; from here on referred to as "method 1") and (2) as the ratio of the original productivity relative to the theoretical maximum productivity (from here on referred to as "method 2"). This theoretical maximum productivity, which was extracted from a map provided by Bergh et al. (2005) with ArcGIS (ESRI, 2011), indicates the productivity that could be obtained under non-nutrient-limited conditions and is further referred to as

attainable productivity. The second method is thus very similar to the IIASA approach (cf. Eq. (1)), but because an estimate

for attainable productivity was only available for spruce, it could only be applied to this species. The two alternative methods for normalizing productivity were used to verify the robustness of analyses, and because each method has its own advantages and disadvantages (the main disadvantage of method 1 is that not only the direct influence of climate on productivity is removed, but also its indirect effect through nutrient availability, so that only effects of regional variation in nutrient availability on productivity remain, while for method 2, there is for example uncertainty related to the estimates of attainable productivity).

Regression analysis was then used to elucidate how different available soil variables were related to normalized productivity (Q1). In addition, normalized productivity was fitted against the IIASA-metric to test its performance. The correlation between the residuals of this relationship and each of the four variables of the metric then indicated whether or not the variables were well implemented (Q2). Finally, the associations found in Q1 indicated how the metric could be upgraded (Q3). Two upgraded metrics were then evaluated in the same way as the original IIASA-metric in Q2, and by investigating if they could explain

variation in productivity for five randomly selected local gradients in nutrient availability. An overview of the methodology is presented in Fig. 2.

## 2.3 Data analyses

The two methods for normalizing productivity each follow a different approach. Method 1 uses the residuals of a general linear model with productivity (Fig. S1a,b) dependent on climate and tree species (Fig. 3a, Tables S1 and S2 and Eq. (S1)). This

approach considers the residuals to reflect deviations in productivity imposed by spatial variation in nutrient availability and in the absence of climate effects. However, residuals deviated more strongly from zero towards the warmer south (Fig. 3a), thus causing heteroscedasticity and a potential bias in the further analyses if not properly accounted for. For further analyses, we therefore split the database into three TSUM groups (north, middle and south; Fig. 3a). For method 2, actual productivities for spruce (Fig. S1b) were divided by hypothetical attainable productivities (Fig. S1c). In other words, while method 1 uses as

response variable the residuals of the climate model per species, method 2 considers the ratio actual/attainable productivity (Fig. 3b).

### 2.3.1 Identifying potentially confounding factors

In order to understand the correlation structure of the database, and avoid multicollinearity in the subsequent analyses, we examined correlations among the soil variables (SOC concentration, total N concentration, total N stock, C:N ratio, sand

fraction, clay fraction, TEB, pHw and $pH_{KCl}$). We performed a principal component analysis (princomp function, package MASS - Venables and Ripley, 2002) in R (R Core Team, 2015) for a visualization and constructed a correlation matrix with Pearson's *r* as correlation coefficients for each variable pair.

Soil moisture and soil type (available as categorical variables) may act as confounding factors for associations between productivity and other soil properties (e.g. in wet soils, the rooting environment is anoxic and decomposition is inhibited

(Olsson et al., 2009), leading to reduced productivity and accumulation of SOM). We therefore tested if the selected soil variables and normalized productivity differed among soil moisture classes (dry, fresh, fresh-moist and moist, as available from the database) and the most common soil types (peaty histosols, wetland gleysols, weakly developed regosols, gravel-rich leptosols and sandy podzols) using two-way ANOVA with soil moisture/type and tree species as fixed factors.

Numerous studies have shown the strong influence of N deposition on forest productivity (e.g. Binkley and Högberg, 2016;
From et al., 2016). Although N deposition can influence the soil properties considered in our analyses, it may also influence productivity without immediate changes in these soil properties (i.e. there is a time lag - Novotny et al., 2015). In other words, for a given set of soil characteristics and climate, productivity may vary depending on N deposition, which would weaken the link between soil properties and normalized productivity. To verify whether N deposition confounded our analyses, we extracted     N    deposition    data    from    a    map    available    at
http://www.smhi.se/sgn0102/miljoovervakning/kartvisare.php?lager=15DTOT_NOY     (Swedish Meteorological and Hydrological Institute, 2018), using the ArcGIS software (ESRI, 2011). We then tested whether N deposition correlated with productivity and soil variables, using Pearson's correlation coefficient ($r$), and performed regression analyses on normalized productivity vs N deposition, stratified by soil moisture and type.

### 2.3.2 *Question 1* - Normalized productivity vs single and combined soil variables

Simple regression analysis was used to determine the relationship between single, continuous soil variables and normalized productivity. We performed these analyses on all data, and on the data stratified by soil moisture and soil type, to test the robustness of the observed relationships in the absence of potentially confounding effects of soil moisture and type, independent of nutrient availability. Then, we tested which combination of continuous soil variables best explained variation in normalized productivity across Sweden (multiple regression analysis). Starting from the full model, non-significant variables
were removed one by one, the order based on significance, after which the mean squared error (mse), based on cross-validation (package DAAG - Maindonald and Braun, 2015) each time indicated whether the variable in question could be removed. Interaction effects up to the first order were added if suggested by regression trees (package tree - Ripley, 2015). For method 1 (Fig. 3), first-order interactions of continuous variables with region as a factor (levels: N, M, S) were included in the selection procedure (i.e. an ANCOVA was used for this approach).

### 2.3.3 *Question 2* - Evaluation of the IIASA-metric

Irrespective of the method applied, a well-functioning nutrient availability metric would be recognized by a clear, positive relationship with productivity. We used linear model analysis to test the significance of the relationship between the metric and normalized productivity, and its explanatory power ($R^2$). To test whether the variables included in the metric were accurately implemented, we also examined the correlation between the residuals of this linear model and each of the variables

included in the metric (SOC, texture, TEB and $pH_w$). A significant correlation suggests that the soil variable under consideration is not optimally implemented in the metric.

### 2.3.4 *Question 3* - Upgrades of the IIASA-metric

Outcomes of *question 1* indicated which soil variables best explained variation in normalized productivity. This information was further used to i) assess if the relationships for variables already included in the IIASA-metric should be altered, ii) remove
soil variables from the metric if their empirical associations with normalized productivity were opposite from their relationships in the original IIASA-metric due to indirect effects of other underlying mechanisms, complicating parametrization and iii) include additional soil variables to improve performance of the metric. Two new metrics were developed: "upgraded metric 1" and "upgraded metric 2", referring to the respective methods of normalizing productivity (Fig. 3). As a starting point for upgraded metric 1, half of the dataset from southern Sweden (where productivity varied most, cf.
Fig 3a) was used as a calibration set to derive regression equations, while half of the complete national dataset for spruce served as a calibration set for new metric 2. The best predictors of normalized productivity as indicated by the analyses in *Question 1* were then adopted as partial metric scores (cf. the original Eqs. (6–9)). Moreover, for new metric 1, the minimum and maximum normalized productivities observed in southern Sweden were included as lower and upper boundaries to the partial metric scores to avoid possible unrealistic values for future applications to other datasets. For method 2, the minima
and maxima were, as in the IIASA-metric, set to 0 and 100%, respectively (units for this metric [%] remained the same as in the original IIASA-metric, while for new metric 1, the unit was [$m^3$ $ha^{-1}$ $yr^{-1}$]). Finally, the two improved metrics for nutrient availability were calculated as in Eq. (10).

Performance of the upgraded metrics was evaluated by (i) testing normalized productivity in the database against the metrics and inspecting the implementation of the variables, and by (ii) testing productivity against the metrics and examining variable
implementation for five randomly selected local gradients in nutrient availability. For (i), the metrics were thus evaluated as described for the IIASA-metric under *question 2*, with the exceptions that validation datasets were used (i.e. the data that were used for developing the metrics were not included for the evaluations), and that the same analyses were also performed after stratifying by soil moisture and type, to assess robustness. For (ii), two gradients with spruce, and three gradients with pine (locations indicated in Fig. S2) were manually selected in ArcGIS (ESRI, 2011). Each of these randomly selected gradients
included at least 40 data points from the Swedish database that were (i) located in the same region, without showing substantial spatial variation in climate, and (ii) showed high spatial variation in either soil moisture, TEB or productivity (we also searched specifically for clear soil C:N gradients, but found none for which climate did not vary; variables like soil C:N or SOC did however sufficiently vary in the selected gradients). We thus not only evaluated the upgraded metrics against normalized productivity across the complete database, but also tested their performance for local gradients, which had the advantage that
no normalization of productivity for climate was needed.

We examined the validity of the linear models' assumptions (linearity, normality of residuals, no influential outliers, homoscedasticity) with standard functions of R (R Core Team, 2015), including diagnostic plots and additional tests from packages. For all regressions, potential non-linearities were detected with histograms of all variables' distributions and generalized additive models from the mgcv package (Wood, 2006). Data were accordingly log-transformed if their distribution was right-skewed, while polynomial (e.g. quadratic) functions were included in the model selection procedure where the general additive models suggested non-linear patterns. The variance inflation factor (package car - Fox and Weisberg, 2011) assessed possible multicollinearity. Whenever confidence intervals are given, they represent standard errors of the mean. For all analyses, $\alpha = 0.05$ was taken as significance level, whereas $P$-values between 0.05 and 0.10 were considered as borderline significant.

## 3 Results

### 3.1 Identifying potentially confounding factors

Correlations among soil properties and nutrients were investigated to acquire a better understanding of the data, and to decide if certain variables could be excluded in the subsequent analyses due to redundancy. In this database, $pH_w$ and $pH_{KCl}$ were strongly correlated. As $pH_{KCl}$ has the practical advantage of showing less seasonal variation than $pH_w$ (Soil Survey Staff, 2014), we opted to use only $pH_{KCl}$ in the analyses for research question 1. Similarly, TN and SOC largely shared the same information. We included SOC in the analyses and discarded TN because SOC is a component of the IIASA-metric of constraints on nutrient availability. Moreover, soil organic matter acts as a nutrient store and provides cation and anion exchange sites, while total N is merely correlated to SOM but only a (small) proportion of total N is available to the plants. Collinearity among other variables was minor (|Pearson's $r$| < 0.65; Fig. 4), and they were thus all included in the analysis.

Relationships between soil variables and normalized productivity might vary depending on factors such as soil moisture and soil type. Therefore, we first examined how these factors influence soil properties and normalized productivity. Soil moisture, for example, may influence nutrient availability of ecosystems by – among others – affecting the rate of decomposition, and consequently change other soil characteristics. In the database, each forest was originally assigned to a soil moisture category. Using these categories, we found that SOC and C:N ratio increased from dry to moist. A similar trend was observed for TEB, while the sand fraction and $pH_{KCl}$ decreased from dry to moist. For clay, no significant differences among soil moisture classes occurred (Fig. S3). Lastly, normalized productivity was highest in the "fresh" soil moisture class and lowest for the wettest forests (Fig. 7c). This pattern was most pronounced in southern Sweden (north - $F_{3,568} = 22.43$, $P < 0.01$; middle - $F_{4,844} = 39.47$, $P < 0.01$; south - $F_{4,1056} = 35.23$, $P < 0.01$; moisture x region - $F_{7,2468} = 3.77$, $P < 0.01$).

Soil properties not only differed among soil moisture classes, but also among soil types. Especially histosols and podzols could be distinguished from the other soils: histosols were characterized by a low $pH_{KCl}$, high SOC and C:N ratio, while podzols were sandy and had a low TEB stock (Fig. S4). Differences in normalized productivity among soil types were observed as

well. Histosols in particular showed reduced productivities compared to other soil types (Fig. 6). Hence, the wetness of a site and its type of soil could confound observed patterns in productivity associated with the soil variables and are therefore taken into account in the further analyses and their interpretation.

Besides soil moisture and soil type, N deposition also may confound associations between normalized productivity and soil data. In our Swedish database, N deposition correlated significantly with all soil variables. Especially total N stock and concentration, and SOC correlated somewhat positively with N deposition (Fig. 4b). N deposition was also strongly positively correlated with productivity (Pearson's $r = 0.73$); both variables increased from north to south, as did the growing season temperature sum (TSUM - which was therefore also highly correlated with (ln) N deposition - Pearson's $r = 0.91$). However,

N deposition did in general not have a significant effect on productivity when the latter was normalized according to method 1 (i.e. residual productivity - Tables S3 and S4). However, normalized productivity according to method 2 (i.e. actual/attainable productivity - Tables S3 and S4) did reveal a strong positive relationship with N deposition. The increasing N deposition along the north-south gradient in Sweden (e.g. Olsson et al., 2009) should thus be kept in mind when interpreting effects of soil variables on productivity, especially when productivity was normalized according to method 2.

**3.2 *Question 1* - Normalized productivity vs single and combined soil variables**

In order to elucidate how soil variables affect nutrient availabilities across Sweden, we used their single and combined relationships with normalized productivity. For method 1, we found that most single soil variables were significantly related to normalized productivity (Table 2), however, the coefficients of determination were small (0.002–0.146). Normalized productivity was significantly negatively correlated with the soil C:N ratio (Fig. 7b), for which the effect became more

pronounced towards the south (i.e. slopes and $R^2$s increased; $F_{2,2274} = 34.23$, $P < 0.01$). For both SOC (Fig. 7a) and pH$_{KCl}$, the relationship with normalized productivity was quadratic, while the associations with soil N stocks and clay were weak yet significantly positive. Normalized productivity and TEB did not significantly correlate, but the trend was weakly positive. The strongest relationships were found for normalized productivity versus SOC, pH$_{KCl}$ and soil C:N ratio and consequently these were among the variables selected for the model with multiple covariates (Table 3).

Results of method 2 were qualitatively similar to those of the other approach for SOC (Fig. 8a), N stock, C:N ratio (Fig. 8b), clay fraction and TEB, although the N stock explained a larger proportion of the variation here and the curve for actual/attainable productivity decreased logarithmically rather than linearly with increasing C:N ratio. However, the function for pH$_{KCl}$ was not quadratic, but linear with a significantly positive slope (Table 2). In summary, SOC and the soil C:N ratio were the only soil factors that showed a similar trend according both methods with an $R^2$ of at least a few percent, and were

thus included in the multiple regression models for both methods 1 and 2 (Table 3).

Since soil moisture and soil type influenced both soil properties and normalized productivity, we also stratified the analyses above by these factors. In general, these separate analyses confirmed the robustness of the observed patterns across the database (despite low $R^2$s), as the results and parameter estimates were similar to those of the previous analysis (Tables S5 and S6).

### 3.3 *Question 2* - Evaluation of the IIASA-metric

Both methods agreed on the poor performance of the IIASA-metric to elucidate patterns in nutrient availability, as the weakly positive correlation between normalized productivity and the metric was rarely significant, and explained < 1 % of the variation in normalized productivity in northern Sweden for method 1 (Fig. 9). Residual values of the relationship between normalized productivity of method 1 and the metric score (Fig. 9a) were significantly associated with all four input variables of the metric (SOC, texture, TEB and $pH_w$ - Table S9). SOC and TEB correlated negatively with these residuals, while sand was significantly

positively related to these same residuals, and productivities at low $pH_w$ were overestimated (the quadratic functions were concave; not shown in Table S9). Residuals of method 2 (Fig. 9b) confirmed the negative trend with TEB, but showed no statistically significant relationship with SOC, texture or $pH_w$ (Table S9). Overall, the fact that residuals were still correlated with the variables in the metric suggests that the input variables were not optimally implemented in the formula.

### 3.4 *Question 3* - Upgrades of the IIASA-metric

From the statistical analyses for *question 1*, we deduce that SOC, soil C:N and pH each play a role in influencing nutrient availability in Sweden. Based on their relationships with normalized productivity in southern Sweden according to method 1 (Table S10), and in entire Sweden according to method 2 (Table S11), the following formulae were implemented in two upgraded nutrient availability metrics (Figs. S5 and S6):

$$\text{SOC Score [m}^3\text{ ha}^{-1}\text{ yr}^{-1}] = \max(-0.18 * (\ln(\text{SOC}_{0-20cm}\text{ [%]}) - \ln(2.3))^2 + 0.525, -5.65) , \tag{11}$$

$$\text{C:N Score [m}^3\text{ ha}^{-1}\text{ yr}^{-1}] = \max(-0.08 * \text{C:N}_{0-20cm} + 2.1, -5.65) , \tag{12}$$

$$\text{pH Score [m}^3\text{ ha}^{-1}\text{ yr}^{-1}] = \max(-0.9 * (\text{pH}_{w,0-20cm} - 4.67)^2 + 0.6, -5.65) , \tag{13}$$

for the metric based on method 1 ("upgraded metric 1"), and

$$\text{SOC Score [%]} = \max(-2.8 * (\ln(\text{SOC}_{0-20cm}\text{ [%]}) - \ln(8.1))^2 + 43.5, 0) , \tag{14}$$

$$\text{C:N Score [%]} = \max(-19 * \ln(\text{C:N}_{0-10cm}) + 102, 0) , \tag{15}$$

$$\text{pH Score [%]} = \max(2 * \text{pH}_{w,0-20cm} + 31, 0) , \tag{16}$$

for the metric based on method 2 ("upgraded metric 2").

In the same way as for the IIASA-metric, Eqs. (11–13) and (14–16) were combined in Eq. (10) to calculate the final nutrient availability score for each metric. Soil texture and exchangeable bases were not included here, as their empirical relationships with normalized productivity showed opposite trends as compared to their implementation in the IIASA-metric (Fig. 1 vs

Tables 2 and S9), likely due to indirect effects of soil moisture and related organic matter accumulation.

In contrast to the IIASA-metric of constraints on nutrient availability, the upgraded metrics were significantly related with normalized productivity (Figs. 10 and 11), albeit with low $R^2$s. The same analyses stratified by soil moisture (Tables S12 and S14) gave similar results for the intermediate "fresh" and "fresh-moist" moisture classes (i.e. those with the majority of data

points), while stratification by soil type generally weakened relationships between the metrics and normalized productivity (only for podzols and and regosols, the metrics could always describe variation; Tables S13 and S15). Only on few occasions did the soil variables included in metric 1 show a (borderline) significant correlation with the residuals of the relationship between normalized productivity and the upgraded metrics (and the associated $R^2$s were always low; Table S16). We therefore conclude that SOC, soil C:N and pH are generally well implemented in this upgraded metric, at least for the database considered here. For upgraded metric 2, however, significant associations with higher $R^2$s emerged, thus indicating suboptimal implementation of the variables in the metric, but the sign of the significant slope differed depending on whether normalization method 1 or 2 was used (Table S17).

Five nutrient availability gradients were selected to evaluate the performance of the upgraded metrics in the absence of confounding climate and N deposition effects (Fig. S2). Both metrics were capable of describing variation in productivity for all gradients, with $R^2$s of 0.092–0.383 (Tables 4 and Y5). Variable implementation was generally good, except for SOC in upgraded metric 2. There, SOC was significantly negatively associated with the residuals of the productivity-metric relationship (for four out of five gradients; Tables S18 and S19). Both the results of the national database and the gradients thus indicate that the upgraded metrics explain part of the spatial variation in productivity. Further upgrades, for example with other soil variables, may be needed to increase their performance.

## 4 Discussion

### 4.1 Identifying potentially confounding factors

Soil moisture varies from dry to very wet across Sweden, and may obfuscate associations between nutrient related soil properties and (normalized) productivity. Across our database, we indeed observed that certain soil properties (SOC, C:N ratio, TEB) were related with soil moisture (Fig. S3), and also normalized productivity depended on soil wetness (Fig. 5.): productivity was highest for intermediate soil moisture levels, while it was significantly reduced for the most dry and wet soils. The influence of soil moisture on productivity can be explained as follows: at high water content, the anoxic rooting environment inhibits root and microbial respiration. Tree productivity is thus suppressed, both directly due to the lack of oxygen for the tree itself, and because nutrient supply is limited due to the inhibition of mineralization (Gorham, 1991). For relatively dry soils, on the other hand, productivity is reduced because of water limitation (which has been shown to occur in southern Sweden - Bergh et al., 1999), lower nutrient inputs through groundwater and less frequent periods with easily available nutrients in the soil solution (Qian and Schoenau, 2002) and lower retention (Larcher, 2003; Roy et al., 2006) and supply (Binkley and Hart, 1989) of nutrients by organic matter. In summary, any associations between a soil variable and productivity should be interpreted in view of the fact that soil moisture may act as a factor influencing both this soil variable and productivity. We therefore performed our analyses not only for the complete set of data, but also stratified by soil moisture to assess whether relationships between soil properties and productivity would change.

In the same way as for soil moisture, stratification by soil type might help in resolving nutrient-productivity relationships. Soil properties and productivity differed among the five most common soil types in the database (i.e. peaty histosols, wetland gleysols, weakly developed regosols, gravel-rich leptosols and sandy podzols - Fig. S4). To some extent, these differences among soil types overlapped with these observed for soil moisture classes (e.g. wet histosols had the highest SOC, C:N and the lowest productivity), but additional patterns emerged as well (e.g. podzols had a particularly low TEB stock). Although actual differences in nutrient availability among soil types will in part underlie the variations in productivity, other factors related to soil type (e.g. wetness, soil depth or the rooting environment) may also influence productivity (Binkley and Hart, 1989). The main analyses of the current study were therefore stratified by both soil moisture and type to test the robustness of associations between nutrient related soil properties and normalized productivity.

Many studies have shown the strong influence of N deposition on forest productivity (e.g. Binkley and Hogberg, 2016; From et al., 2016). As expected, N deposition correlated to some extent with some of the soil variables considered in the present study, such as the total soil N stock and concentration (Fig. 4b). Furthermore, N deposition was strongly positively related to productivity. However, this effect of N deposition on productivity cannot be separated from the influence of climate and light, as all these factors increase together in the north-south direction. Nevertheless, we argue that for the goals of this study, i.e. investigating soil nutrient-productivity relationships across Sweden and developing a nutrient metric, the spatially varying N deposition is not problematic, since the normalization for climate and species according to method 1 (Fig. 3a) at the same time

also removed the influence of the confounding N deposition on productivity. Accordingly, "Residual productivity" was generally not correlated with N deposition (Table S3). The response variable derived from method 2 (i.e. actual/attainable

productivities for spruce - Fig. 3b) in contrast, correlated strongly with N deposition (Table S4), because both actual/attainable productivity and N deposition increased from north to south. Consequently, relationships between actual/attainable productivity and soil data for this method were unavoidably confounded by N deposition.

### 4.2 *Question 1* - Normalized productivity vs single and combined soil variables

Soil C:N ratio had a negative effect on normalized productivity across both methods (Figs. 7b and 8b). Apart from high N

concentrations at low C:N, increased productivities with decreasing C:N ratio can follow from its influence on litter decomposition and mineralization, and thus on nutrient availability: when the ratio in organic matter is high, microbes more strongly immobilize N to adjust their internal C to N stoichiometry. As a consequence, N is not easily released and made available for plant uptake. A low C:N ratio, on the other hand, facilitates N mineralization (Roy et al., 2006) and thus enhances N availability (Wilkinson et al., 1999). Last, the logarithmic relationship between actual/attainable productivity and soil C:N

ratio indicates that in the lower range, close to the internal C:N stoichiometry microbes pursue (i.e. 5–17:1 - Cleveland and Liptzin, 2007), a small shift in C:N can have a large effect on mineralization rates and subsequent N supply to sustain plant biomass production. Where the C:N ratio is high, in contrast, microbial nutrient release is always low, and a shift in C:N would be of little importance. In other words, intuitively, we can hypothesize that a shift in the C:N ratio from, for instance, 25:1 to 20:1 will make a larger difference to the equilibrium between mineralization and immobilization as compared to a change from

45:1 to 40:1.

The quadratic relationship of ln SOC with normalized productivity (Figs. 7a and 8a) is partly explained by the role of SOM in storing and exchanging nutrients, but also partly by the confounding effect of soil moisture acting. At high moisture levels, SOC most likely increases because decomposition is reduced in water saturated soils, leading to organic matter accumulation (Fig. S3a). Anoxic soils impede productivity, because of the aforementioned prevention of root respiration and reduced supply

of newly available nutrients through mineralization. At low SOC, on the other hand, productivity supposedly decreases with decreasing SOC because of water limitation and low availability of organic matter, which acts as a nutrient store. Together, these results suggest that the empirical relationship between SOC and productivity might have an optimum below which soil fertility is reduced due to a lack of sufficient organic matter itself, and above which high SOC indicates hostile rooting conditions and limited nutrient supply through slow mineralization. The first aspect is thus included in the IIASA-metric (Fig.

1), while the decreasing part of the curve should be included in the empirical relationship of SOC with nutrient availability if the effect of reduced decomposition is not captured by any of the other soil variables in an updated metric.

Soil factors other than the soil C:N ratio and SOC either exhibited only a marginal influence on normalized productivity or their effect depended on the approach (Table 2). N stocks could explain variation across both methods, but their explanatory power was rather modest for method 1. We anticipate that including N stock in a nutrient availability metric will be of limited

value, as this variable is only loosely related to N availability at the continental or global scale: readily available N only represents a small portion of the total N stock, because most soil N is incorporated in organic matter (Binkley and Hart, 1989). Mineral soil clay fractions had a weak but significantly positive effect on normalized productivity. Even though clay particles can protect SOM from decomposition (Xu et al., 2016), clay soils in the Swedish database in all likelihood positively influence nutrient availability by means of their negative charges that serve as cation exchange sites (i.e. for $NH_4^+$, $K^+$, $Ca^{2+}$ and $Mg^{2+}$ - IIASA and FAO, 2012). Effects of TEB and pH were dependent on the method, possibly reflecting differences between regional (method 1) and national (method 2) variation in nutrient availability.

All equations resulting from multiple regression analysis combining different soil variables contained the soil C:N ratio and SOC (Table 3), confirming that, in the absence of direct soil nutrient data, these are key and complementary determinants of nutrient availability in northern coniferous forests. Qualitatively considered, associations of C:N ratio (-), SOC (concave quadratic after log-transformation), N stock (+) and clay fraction (+) with normalized productivity were consistent for both approaches (Table 2). Together with their abilities to explain variation, the consistent effects of C:N and SOC suggest these soil variables have most potential for inclusion in an improved nutrient availability metric.

### 4.3 *Question 2* - Evaluation of the IIASA-metric

Although the IIASA-metric of constraints of nutrient availability was originally designed for evaluating constraints on nutrient availability of arable lands, we opted to start with this metric for a few reasons. Apart from the fact that to our knowledge, it represents the only attempt so far to develop a generic nutrient metric, the structures of its formulas (Eqs.(6-9)) reflect general mechanisms that link soil properties to nutrient availability, which are also valid for non-agricultural ecosystems. Soil pH for example shows a typical optimum effect on nutrient availability, while SOC and TEB have a direct positive non-linear influence (IIASA and FAO, 2012). The final weighing of the four partial scores (Eq. (10)) finds its rationale in the idea that if a certain soil property is particularly suboptimal, it will be the most important nutrient-related determinant of productivity, with less influence of the other soil properties that are within or close to the optimal range. This way of weighing can be considered as a type of interaction, but one that cannot be implemented in a simple linear regression model. Hence, our main reason for adopting the IIASA-metric as a starting point is that, in spite of its simplicity, it is based on theoretical considerations. Moreover, adopting this structure allows for updating with other datasets - something that can probably not be achieved with multiple regression equations (see below).

The IIASA-metric of constraints on nutrient availability does not clarify much variation in normalized productivity among Swedish forests: only in the north, the metric was significantly positively associated with normalized productivity of method 1, but the explained variation was minor ($R^2 = 0.008$). In middle and southern Sweden, and for actual/attainable productivity across the country (method 2), the IIASA-metric was not significantly correlated with the response (Fig. 9). A low performance of the IIASA-metric in its current form for the Swedish database was expected, as it was initially developed for evaluating (constraints on) the soil fertility of agricultural ecosystems. Species and soil conditions of such ecosystems indeed greatly

differ from the boreal forests investigated in the present study. Many Swedish forest soils are for instance coarse-textured, and in addition, the database contains wet-soil forests, while arable soils are typically not water saturated. Especially the future inclusion of variables closely related to N availability (e.g. C:N, see below) could potentially improve performance of the metric for N limited boreal forests.

Although the IIASA-metric was not or only weakly related to normalized productivity, the variables it includes did exhibit significant relationships with the residuals obtained from the relationship between normalized productivity and the metric. In other words, SOC, soil texture, TEB and $pH_w$ were not optimally implemented in the IIASA-metric. For example, SOC and TEB were negatively associated with productivity (residuals) instead of positively as suggested by the metric (Table S9). In Sweden, the high organic matter contents are very likely not the direct reason for the suppressed productivity. As reasoned under *question 1*, organic matter probably accumulated in places where decomposition rates are low (Minderma, 1968), giving rise to almost purely organic topsoils. This slow decomposition, in turn, may arise from high water tables and related low soil oxygen (Olsson et al., 2009 - see *question 1* for evidence) and/or low temperatures (Larcher, 2003). Similarly, the organic matter typically retains exchangeable base cations (IIASA and FAO, 2012), explaining the association of TEB with residuals.

**4.4 *Question 3* - Upgrades of the IIASA-metric**

Based on results of the analyses for *question 1*, the nutrient availability metric was improved by i) including an empirical optimum in the influence of SOC on normalized productivity, and ii) including soil C:N, thus more explicitly incorporating the availability of N. In the current analysis, soil texture and TEB were excluded from the metrics, as they exhibited negative instead of the expected positive associations with normalized productivities (IIASA and FAO, 2012), probably due to indirect effects of low soil oxygen, reduced decomposition and suppressed productivity where the proportion of sand is low and TEB is high.

In contrast to the original metric developed by IIASA, the upgraded metrics described some variation across all approaches using the full database (Figs. 10 and 11). Variables were generally properly implemented, at least for upgraded metric 1 (Table 5), while for metric 2, significant associations emerged between residuals of normalized productivity and SOC and pH (Table S17). These associations had, however, opposite trends depending on whether normalization method 1 or 2 was considered, meaning that the methods did not agree on whether the metric score was either under- or overestimated for the soil variable considered. The stratified analyses confirm that the metrics are an improvement, at least for those soil moisture classes and soil types with sufficient data points (Tables S12-15). Moreover, each metric could describe spatial variation in productivity for five randomly selected nutrient availability gradients (Tables 4 and 5). The coefficients of determination were generally higher for these gradients than for the database analyses, likely because the gradients did not require a normalization for climate (the latter increased the uncertainty on the response variable, see section 4.5 on sources of uncertainty and future challenges). Lastly, the gradients generally confirmed the correct implementation of soil variables in upgraded metric 1 (Table S18), whereas for metric 2, scores for high SOC might be overestimated (Table S19).

Variation in normalized productivity explained by the upgraded metrics ($R^2 = 0.03–0.21$ and $R^2 = 0.06–0.18$) was similar to the variation explained by multiple regression equations ($R^2 = 0.18–0.22$) that contained the same (and more) soil variables than the metrics. The metrics can, however, easily be updated, while equations derived from multiple regressions would be less useful for developing a nutrient availability metric because such an entirely empirical approach does not allow for later updates of parameters or model structures based on data from other ecosystems. In order to further enhance performance of the metrics, and to test to what extent they can describe variation in nutrient availability outside of Swedish conifer forests, additional datasets with productivity and soil information are needed. Such datasets include large-scale inventories such as the one considered in the present study, but also local gradients and nutrient manipulation experiments. The latter two have lower generalizability, but offer the advantage that normalization for climate is not needed. Moreover, the usefulness of additional nutrient related data often not available in large datasets, such as foliar nutrients (e.g. Cleveland et al., 2011), stable N isotope signatures (e.g. Craine et al., 2009, 2015; Wolf et al., 2011), ion exchange resin/membrane derived supply rates (e.g. Lundell, 2001; Qian and Schoenau, 2002), Ellenberg values (Andrianarisoa et al., 2009) etc. may be assessed.

## 4.5 Sources of uncertainty and future challenges

Even though normalized productivity was significantly related to soil properties, and to our upgraded metrics, much of the variation in normalized productivity remains unexplained. The considerable unexplained variation may have multiple reasons. Apart from a possible lack of soil and nutrient data more closely related to N availability than the ones available in our database, another possible factor reducing $R^2$s could be the quality of the data in the database. This could for instance be due to an insufficient number of replicates sampled per data point ($n = 3$ for the soils), although this is probably of limited importance because of the large number of data points in the database itself. A more important source of uncertainty is probably the inevitable uncertainty related to the response variable, i.e. "climate-normalized" aboveground productivity. This not only includes uncertainty in the original productivity estimates (for which for example differences in management or disturbances likely increased variability), but also uncertainty related to the normalization for climate. By taking residuals of the productivity vs climate regression model (method 1), we for instance unintentionally not only removed the direct effect of climate on productivity, but also its indirect effect through nutrient availability. Normalized productivity based on this method thus mainly represents productivity as influenced by regional variation in nutrient availability. The approach taking actual/attainable productivity as a response variable (method 2) does not suffer from this issue, but there the estimates of attainable productivity come with a high uncertainty. As a consequence, the low $R^2$ values are party due to shortcomings of the normalization procedure that can only be overcome by using datasets where climate does not vary but nutrient availability does. Such datasets are provided by local gradients, such as the five local nutrient availability gradients that we randomly selected from our database for additional evaluation of our upgraded metrics.

Despite the limited $R^2$, similar significant results for different methods (1 and 2) and subsets of the database (regions, soil moisture classes and soil types) indicated that the findings about the soil properties and nutrients are generally robust. The

upgraded metrics explained up to 21 % of the variation in normalized productivity. It is unclear to what degree the influence of nutrient availability is covered by this percentage. Future studies, where additional soil data can be included, will need to verify this. In any case, the significant relationships with normalized productivity, the better implementation of the soil variables and the capability of the metrics to explain up to 38 % of the variation in productivity across different gradients imply
a significant improvement compared to the original IIASA-metric for this database.

A key challenge in the further development of a metric describing spatial variation in nutrient availability both within and outside the boreal biome is differential nutrient limitation. Eventually, we want to be able to compare for example N-limited and P-limited systems. The original structure of the IIASA-metric, which was kept in our upgraded metrics, facilitates this by allowing the inclusion of multiple soil variables such as C:N (mainly relating to N availability), pH (among others a critical
factor controlling P availability) and exchangeable bases in one single metric. In fact, the IIASA-metric is particularly useful in this regard, as it gives more weight to the soil factor with the lowest score. This corresponds to reality and enables accounting for the type of nutrient limitation. For instance, if C:N is high, indicating N limitation, the metric score will be substantially reduced by this high C:N, while at low C:N other limiting factors can dominate the metric score.

## 5 Conclusions

In our database, the soil properties explaining most variation in tree productivity across Swedish conifer forests were SOC and the soil C:N ratio. The empirical relationship between SOC and normalized productivity showed an optimum, reflecting the soil characteristic's direct positive effect on nutrient availability only at low soil carbon concentrations, whereas at high SOC, its effect was masked by other environmental factors (soil moisture and oxygen, and temperature), affecting both SOC and productivity through their role in regulating organic matter formation and decomposition rates. The soil C:N ratio showed the
expected negative correlation with normalized productivity in the present database. Based on the resulting regression equations, we upgraded the IIASA-metric for Swedish conifer forests by adjusting the relationship between soil organic carbon concentration and nutrient availability, and by incorporating soil C:N.

The current nutrient availability metrics were developed based on data from Swedish conifer forests only, and are therefore not yet ready for extrapolation outside the boreal biome. In order to eventually obtain a sufficiently accurate nutrient
availability metric that enables comparison of the nutrient status across experimental and observational sites also beyond the boreal biome, the upgraded metrics developed in this study now need to be validated and further improved based on other forests for which the necessary soil information is available. In a later stage, this approach can then be expanded to other ecosystem types.

**Code and data availability**

The Swedish national database and R scripts with statistical analyses are available at https://www.dropbox.com/s/llbz1p6rtkrccjh/KevinVanSundert_etal_Biogeosciences_2018.7z?dl=0.

**Supplementary information is available online at doi: XXX.**

**Author contribution**

SV and KVS conceived the study. KVS performed the analyses and wrote the manuscript. JAH provided statistical advice and JS provided data. All authors contributed to the discussions and the writing of the manuscript.

**Competing interests**

The authors declare that they have no conflict of interest.

**Acknowledgements**

This research was supported by the Fund for Scientific Research – Flanders (FWO aspirant grant to KVS; FWO postdoctoral fellowship to SV) and by the European Research Council grant ERC-SyG-610028 IMBALANCE-P. We also acknowledge support from the ClimMani COST Action (ES1308). The Swedish Forest Soil Inventory is part of the national environmental monitoring commissioned by the Swedish Environmental Protection Agency. EC–JRC–MARS provided precipitation data. We thank Ivan Janssens for his valuable comments on earlier versions of the manuscript.

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

**Table 1.** Overview of variables of the database used in the current study. Each plot for soil and vegetation analyses had a 10 m radius and was sampled once during the period 2003-2012. The (mostly managed) forests in the inventory represent a random sample of Swedish forests. Abbreviations: MAP = mean annual precipitation; TSUM = growing season temperature sum; SOC = soil organic carbon concentration; TEB = total exchangeable bases; $pH_w$ = pH measured in water; $pH_{KCl}$ = pH measured in KCl solution; TN = total nitrogen concentration; C:N ratio = carbon to nitrogen ratio.

| Available data | location | climate | soil[b] | vegetation |
|---|---|---|---|---|
| | latitude [° N] longitude [° E] elevation [m] | MAP [mm] TSUM[a] [° C days] | horizon thickness [cm] humus stock [ton ha$^{-1}$] humus depth [cm] SOC [%] texture [% sand, silt, clay] TEB [cmol$_+$ kg$^{-1}$ or cmol$_+$ m$^{-2}$] $pH_w$, $pH_{KCl}$ TN [%], C:N ratio, soil moisture[c] [classified] soil type[d] [classified] | age[e] [yrs], tree species composition [%] productivity[f] [m³ ha$^{-1}$ yr$^{-1}$] |

[a]TSUM was calculated for each data point based on its latitude, longitude and elevation.

[b]$n$ = 3; soil variables were determined using standard sampling and laboratory procedures (e.g. Olsson et al., 2009; Stendahl et al., 2010).

[c]Soil moisture was determined in the field based on indicators (e.g. groundwater depth, moisture at the surface, ground vegetation, elevated tree trunks, ....). The classification is representative of the average moisture conditions during the growing season.

[d]Taxonomic soil classification based on the World Reference Base for Soil Resources.

[e]Stand age ranged between 1 and 350 years, with an average of 65 years.

[f]Productivities (site quality) or mean annual volume increments (MAI) over a full rotation were estimated based on height development curves. *In situ* productivities may be lower, depending on the management.

**Table 2.** Associations between single soil variables and normalized productivity for Swedish spruce and pine forests. Significance (*P*-values) of single soil variable effects on residual productivity (mean annual increment - MAI [m³ ha⁻¹ yr⁻¹]) and actual/attainable MAI (for spruce only) across Sweden are given. For (near) significant variables (i.e. $P < 0.10$), parameter estimates ± s.e.m. and the proportion of variation explained ($R^2$) are shown as well. Abbreviations: N = north; M = middle; S = south; SOC = soil organic carbon concentration; C:N = soil carbon to nitrogen ratio; TEB = total exchangeable bases; quad = parameter estimate for quadratic term; lin = parameter estimate for linear term of a quadratic function. For actual/attainable MAI, the model including 0-10 cm soil C:N performed better than the one with 0-20 cm soil C:N (ms = 173 vs 159). No depths are given for soil texture (% sand and % clay), because these were previously calculated based on texture classes of the upper mineral soil with the help of particle size diagrams.

| Normalized productivity response | Region | ln SOC 0-20cm [%] | ln N stock 0-20cm [g m⁻²] | C:N 0-20cm | ln C:N 0-10 cm | Mineral soil sand [%] | Mineral soil clay [%] | ln TEB stock 0-20cm [cmol₊ m⁻²] | pH$_{KCl}$ 0-20cm |
|---|---|---|---|---|---|---|---|---|---|
| Residual MAI (method 1) | N (n = 542) | quad = -0.16 ± 0.02 P < 0.01 lin = 0.49 ± 0.08 P < 0.01 intercept = -0.19 ± 0.08 P = 0.03 $R^2_{tot}$ = 0.145 | slope = 0.29 ± 0.06 P < 0.01 intercept = -1.5 ± 0.3 P < 0.01 $R^2_{tot}$ = 0.012 | slope = -0.014 ± 0.004 P < 0.01 intercept = 0.3 ± 0.1 P < 0.01 $R^2$ = 0.021 | N/A | slope = 0.003 ± 0.001 P = 0.01 intercept = -0.2 ± 0.1 P = 0.03 $R^2_{tot}$ = 0.008 | slope = 0.009 ± 0.004 P = 0.02 intercept = -0.05 ± 0.03 P = 0.14 $R^2_{tot}$ = 0.002 | P = 0.11 | quad = -0.71 ± 0.06 P < 0.01 lin = 5.3 ± 0.4 P < 0.01 intercept = -9.7 ± 0.9 P < 0.01 $R^2_{tot}$ = 0.099 |
| | M (n = 777) | quad = -0.16 ± 0.02 P < 0.01 lin = 0.35 ± 0.08 P < 0.01 intercept = -0.03 ± 0.08 P = 0.71 $R^2_{tot}$ = 0.145 | slope = 0.29 ± 0.06 P < 0.01 intercept = -1.5 ± 0.3 P < 0.01 $R^2_{tot}$ = 0.012 | slope = -0.027 ± 0.005 P < 0.01 intercept = 0.7 ± 0.2 P < 0.01 $R^2$ = 0.029 | N/A | slope = 0.003 ± 0.001 P = 0.01 intercept = -0.23 ± 0.09 P = 0.01 $R^2_{tot}$ = 0.008 | slope = 0.009 ± 0.004 P = 0.02 intercept = -0.05 ± 0.03 P = 0.14 $R^2_{tot}$ = 0.002 | P = 0.11 | quad = -0.71 ± 0.06 P < 0.01 lin = 5.6 ± 0.4 P < 0.01 intercept = -10.8 ± 0.8 P < 0.01 $R^2_{tot}$ = 0.099 |
| | S (n = 946) | quad = -0.16 ± 0.02 P < 0.01 lin = 0.19 ± 0.09 P = 0.03 intercept = 0.5 ± 0.1 P < 0.01 $R^2_{tot}$ = 0.145 | slope = 0.29 ± 0.06 P < 0.01 intercept = -1.5 ± 0.3 P < 0.01 $R^2_{tot}$ = 0.012 | slope = -0.082 ± 0.007 P < 0.01 intercept = 2.0 ± 0.2 P < 0.01 $R^2$ = 0.112 | N/A | slope = 0.003 ± 0.001 P = 0.01 intercept = 0.00 ± 0.08 P = 0.98 $R^2_{tot}$ = 0.008 | slope = 0.009 ± 0.004 P = 0.02 intercept = -0.05 ± 0.03 P = 0.14 $R^2_{tot}$ = 0.002 | P = 0.11 | quad = -0.71 ± 0.06 P < 0.01 lin = 5.9 ± 0.4 P < 0.01 intercept = -11.5 ± 0.8 P < 0.01 $R^2_{tot}$ = 0.099 |
| Actual/attainable MAI (method 2) | entire Sweden (n = 955) | quad = -2.6 ± 0.4 P < 0.01 lin = 11 ± 2 P < 0.01 intercept = 32 ± 2 P < 0.01 $R^2$ = 0.048 | slope = 10.7 ± 0.8 P < 0.01 intercept = -18 ± 5 P < 0.01 $R^2$ = 0.146 | N/A | slope = -19 ± 5 P < 0.01 intercept = 100 ± 5 P < 0.01 $R^2$ = 0.131 | slope = -0.04 ± 0.02 P = 0.01 intercept = 42 ± 1 P < 0.01 $R^2$ = 0.005 | slope = 0.18 ± 0.06 P < 0.01 intercept = 39.2 ± 0.6 P < 0.01 $R^2$ = 0.008 | slope = 2.0 ± 0.5 P < 0.01 intercept = 32 ± 2 P < 0.01 $R^2$ = 0.014 | slope = 3 ± 1 P < 0.01 intercept = 29 ± 4 P < 0.01 $R^2$ = 0.009 |

**Table 3.** Estimates ± s.e.m. for parameters of the selected multiple regression equations linking soil variables to normalized productivity for Swedish conifer forests. Significance of the pattern ($P$ values) and proportion of variation explained ($R^2$) are given as well. Abbreviations: MAI = mean annual increment [$m^3$ $ha^{-1}$ $yr^{-1}$]; N = north; M = middle; S = south; SOC = soil organic carbon concentration; C:N = soil carbon to nitrogen ratio; TEB = total exchangeable bases; quad = parameter estimate for quadratic term; lin = parameter estimate for linear term of a quadratic function. Output of the model selection procedures for methods 1 and 2 is shown in Tables S7 and S8. Data for method 1 are for both spruce and pine, whereas actual/attainable MAI (method 2) was only available for spruce. No depths are given for soil texture (% sand), because these were previously calculated based on texture classes of the upper mineral soil with the help of particle size diagrams.

| Normalized productivity response | Region | ln SOC 0-20cm [%] | ln N stock 0-20cm [g m⁻²] | C:N 0-20cm | ln C:N 0-10 cm | Mineral soil sand [%] | ln TEB stock 0-20cm [cmol₊ m⁻²] | pH$_{KCl}$ 0-20cm | intercept | P and $R^2$ |
|---|---|---|---|---|---|---|---|---|---|---|
| Residual MAI (method 1) | N (n = 542) | quad = -0.16 ± 0.02 P < 0.01 lin = 0.34 ± 0.08 P < 0.01 | not selected | lin = -0.004 ± 0.007 P = 0.58 | N/A | not selected | lin = 0.13 ± 0.04 P < 0.01 | quad = 0.3 ± 0.2 P = 0.22 lin = -2 ± 2 P = 0.22 | 4 ± 3 P = 0.27 | P < 0.01 $R^2_{tot}$ = 0.180 |
| | M (n = 777) | quad = -0.16 ± 0.02 P < 0.01 lin = 0.34 ± 0.08 P < 0.01 | | lin = -0.014 ± 0.006 P = 0.03 | N/A | | lin = 0.13 ± 0.04 P < 0.01 | quad = 0.0 ± 0.1 P = 0.88 lin = 0.2 ± 0.9 P = 0.86 | 0 ± 2 P = 0.81 | |
| | S (n = 946) | quad = -0.16 ± 0.02 P < 0.01 lin = 0.34 ± 0.08 P < 0.01 | | lin = -0.050 ± 0.008 P < 0.01 | N/A | | lin = 0.13 ± 0.04 P < 0.01 | quad = -0.40 ± 0.08 P < 0.01 lin = 2.7 ± 0.6 P < 0.01 | -3 ± 1 P < 0.01 | |
| Actual/attainable MAI (method 2) | entire Sweden (n = 955) | quad = -2.3 ± 0.4 P < 0.01 lin = 9 ± 2 P < 0.01 | lin = 6 ± 1 P < 0.01 | N/A | lin = -15 ± 2 P < 0.01 | lin = -0.02 ± 0.02 P = 0.20 | not selected | lin = -3 ± 1 P < 0.01 | 64 ± 13 P < 0.01 | P < 0.01 $R^2$ = 0.215 |

**Table 4.** Evaluation of upgraded nutrient availability metric 1 for randomly selected nutrient availability gradients in Sweden (Fig. S2). Statistics indicate the relationship between productivity (mean annual increment - $m^3\,ha^{-1}\,yr^{-1}$) and the metric. For (near) significant variables (i.e. $P < 0.10$), parameter estimates ± s.e.m. and the proportion of variation explained ($R^2$) are given. For Norway spruce, no TEB gradient without substantial variation in climate was found, so that only for Scots pine, there was a gradient in TEB. Abbreviations: TEB = total exchangeable bases. Error bars represent the s.e.m.

| Dominant tree species | Soil moisture gradient | TEB gradient | Productivity gradient |
|---|---|---|---|
| Norway spruce | slope = 1.6 ± 0.4<br>$P < 0.01$<br>$R^2 = 0.125$<br>$n = 132$ | N/A | slope = 1.6 ± 0.4<br>$P < 0.01$<br>$R^2 = 0.150$<br>$n = 78$ |
| Scots pine | slope = 1.4 ± 0.2<br>$P < 0.01$<br>$R^2 = 0.208$<br>$n = 141$ | slope = 1.1 ± 0.3<br>$P < 0.01$<br>$R^2 = 0.205$<br>$n = 59$ | slope = 1.9 ± 0.3<br>$P < 0.01$<br>$R^2 = 0.350$<br>$n = 67$ |

**Table 5.** Evaluation of upgraded nutrient availability metric 2 for randomly selected nutrient availability gradients in Sweden (Fig. S2). Statistics indicate the relationship between productivity (mean annual increment - m³ ha$^{-1}$ yr$^{-1}$) and the metric. For (near) significant variables (i.e. $P < 0.10$), parameter estimates ± s.e.m. and the proportion of variation explained ($R^2$) are given. For Norway spruce, no TEB gradient without substantial variation in climate was found, so that only for Scots pine, there was a gradient in TEB. Abbreviations: C:N = soil carbon to nitrogen ratio; TEB = total exchangeable bases. Error bars represent the s.e.m.

| Dominant tree species | Soil moisture gradient | TEB gradient | Productivity gradient |
|---|---|---|---|
| Norway spruce | slope = 0.31 ± 0.08<br>$P < 0.01$<br>$R^2 = 0.092$<br>$n = 132$ | N/A | slope = 0.36 ± 0.08<br>$P < 0.01$<br>$R^2 = 0.188$<br>$n = 78$ |
| Scots pine | slope = 0.28 ± 0.05<br>$P < 0.01$<br>$R^2 = 0.177$<br>$n = 141$ | slope = 0.23 ± 0.06<br>$P < 0.01$<br>$R^2 = 0.213$<br>$n = 59$ | slope = 0.52 ± 0.08<br>$P < 0.01$<br>$R^2 = 0.383$<br>$n = 67$ |

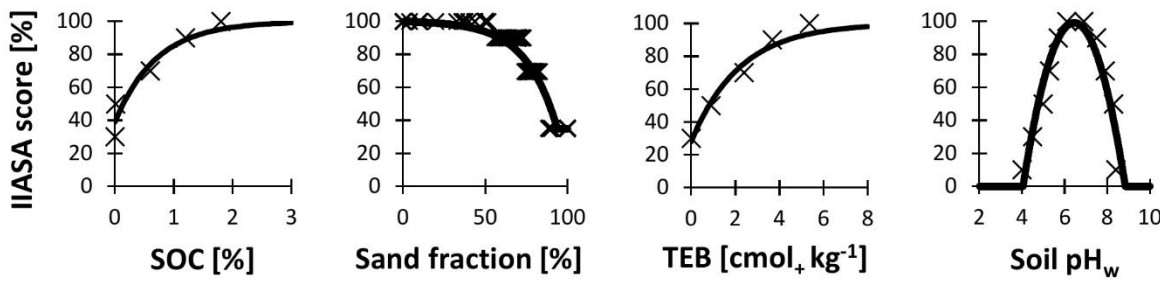

**Figure 1.** IIASA soil scores for soil organic carbon concentration (SOC), texture, total exchangeable bases (TEB) and pH measured in water (pH$_w$). The curves were drawn based on approximate functions through the points, which were derived from crop-specific scores in a look-up table ((http://webarchive.iiasa.ac.at/Research/LUC/GAEZv3.0/soil_evaluation.html, IIASA and FAO, 2012) and averaged over all crop species in the table.

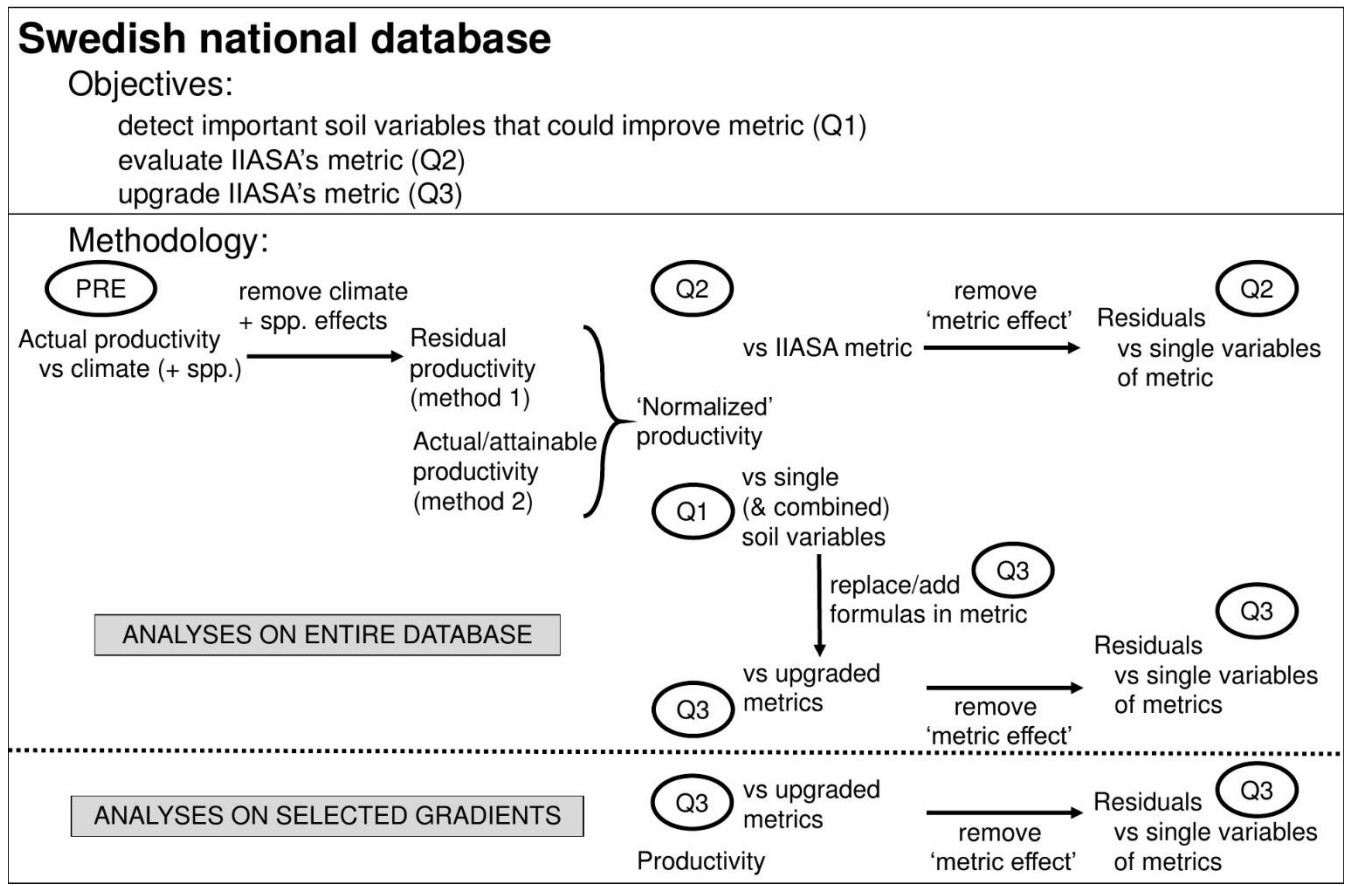

**Figure 2.** Objectives and methods followed in the current paper. PRE refers to a regression model of productivity vs climate and species; Q1, Q2 and Q3 refer to the research questions. Performance of the upgraded nutrient metrics was evaluated against the entire database, and against five nutrient availability gradients, selected from the database (Fig. S2).

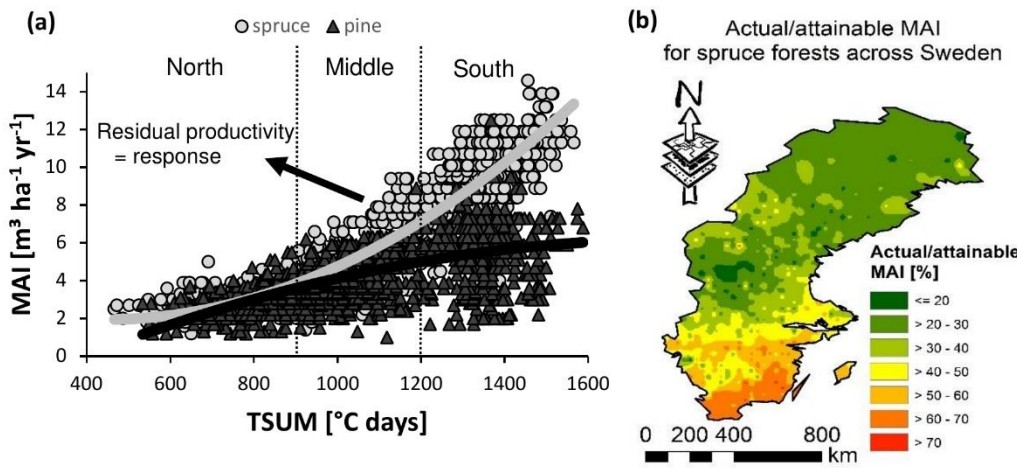

**Figure 3.** Normalized productivity was calculated in two alternative ways. (a) In method 1, residual values were taken from a regression model, explaining variation in mean annual increments (MAI) by climate (growing season temperature sum or TSUM and precipitation) and species. The selection procedure, equation and parameter estimates are given in the supplementary information (resp. Table S1, Eq. (S1) and Table S2). In order to avoid heteroscedasticity-induced artefacts, the dataset was split in a northern (TSUM < 900 °C days), middle (900 °C days < TSUM < 1200 °C days) and southern (TSUM > 1200 °C days) region for this approach. (b) In method 2, actual productivities for spruce were divided by theoretically attainable productivities, provided by Bergh et al. (2005).

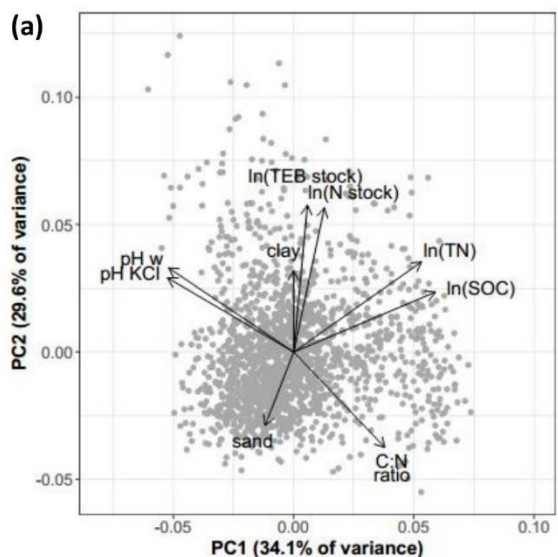

(a)

(b)

| soil variable | ln TN | ln N stock | C:N ratio | sand | clay | ln stock | TEB | pHw | pHKCl | ln N dep. |
|---|---|---|---|---|---|---|---|---|---|---|
| ln SOC | 0.97 | 0.38 | 0.39 | -0.18 | 0.05 | 0.33 | | -0.45 | -0.45 | 0.30 |
| ln TN | | 0.52 | 0.15 | -0.21 | 0.10 | 0.43 | | -0.34 | -0.34 | 0.34 |
| ln N stock | | | -0.42 | -0.16 | 0.19 | 0.58 | | 0.11 | 0.12 | 0.39 |
| C:N ratio | | | | 0.06 | -0.16 | -0.23 | | -0.52 | -0.53 | -0.10 |
| sand | | | | | -0.44 | -0.20 | | 0.01 | 0.03 | -0.21 |
| clay | | | | | | 0.21 | | 0.11 | 0.10 | 0.13 |
| ln TEB stock | | | | | | | | 0.39 | 0.22 | 0.13 |
| pHw | | | | | | | | | 0.89 | -0.18 |
| pHKCl | | | | | | | | | | -0.12 |

**Figure 4.** Correlation structure of a set of potential key soil variables for a soil depth of 0-20 cm. (a) = PCA biplot (sd for PC1 = 1.75, sd for PC2 = 1.63). (b) = correlation matrix, showing Pearson's r for the variable pairs, including correlations with nitrogen deposition.. Abbreviations: SOC = soil organic carbon concentration [%]; TN = soil total nitrogen [%], N stock = amount of nitrogen in the layer [g m$^{-2}$]; C:N ratio = soil carbon to nitrogen ratio; sand = % sand in the mineral soil; clay = % clay in the mineral soil; TEB = total exchangeable bases [cmol$_+$ m$^{-2}$]; pHw = pH measured in water; pHKCl = pH measured in KCl solution; N dep = nitrogen deposition.

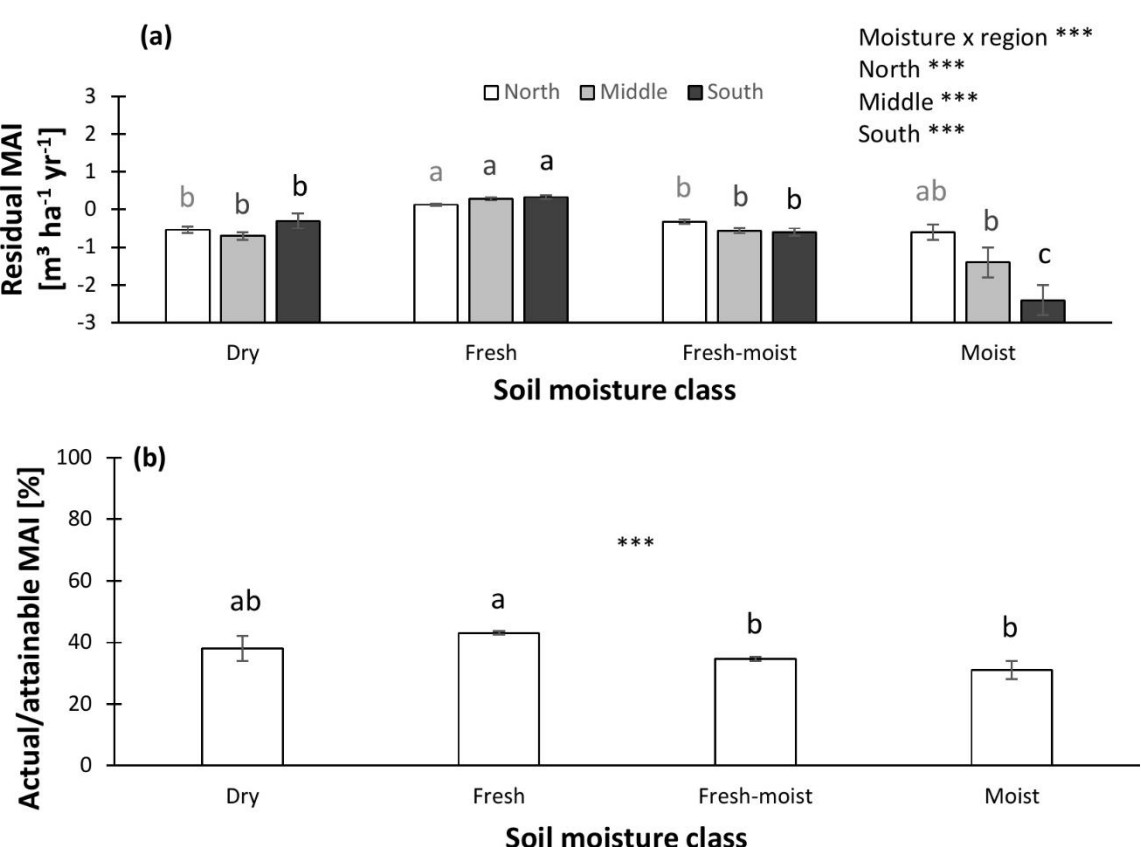

**Figure 5.** Normalized productivity per soil moisture class. (a) Productivity normalized following method 1 (residual mean annual increment - MAI) vs soil moisture. (b) Productivity normalized following method 2 (actual/attainable mean annual increment - MAI) vs soil moisture. In panel a, separate analyses were performed for northern, middle and southern Sweden, as the moisture effects differed among regions. *** indicates significant differences at the $P < 0.01$ level. Error bars represent the s.e.m.

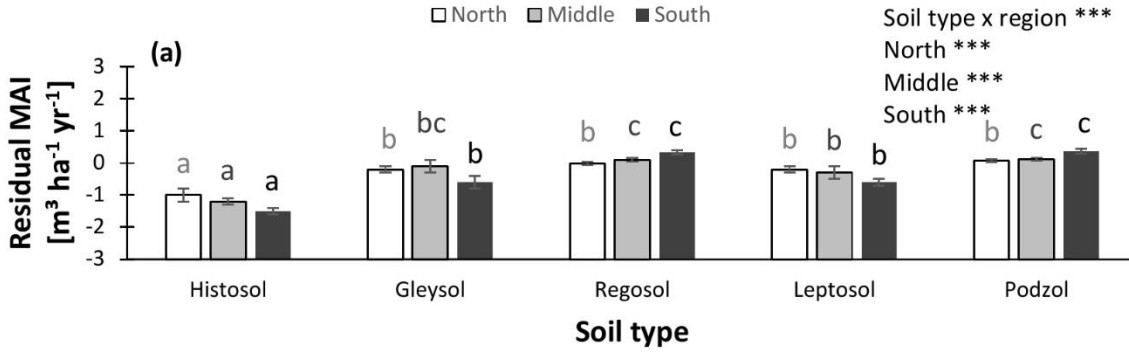

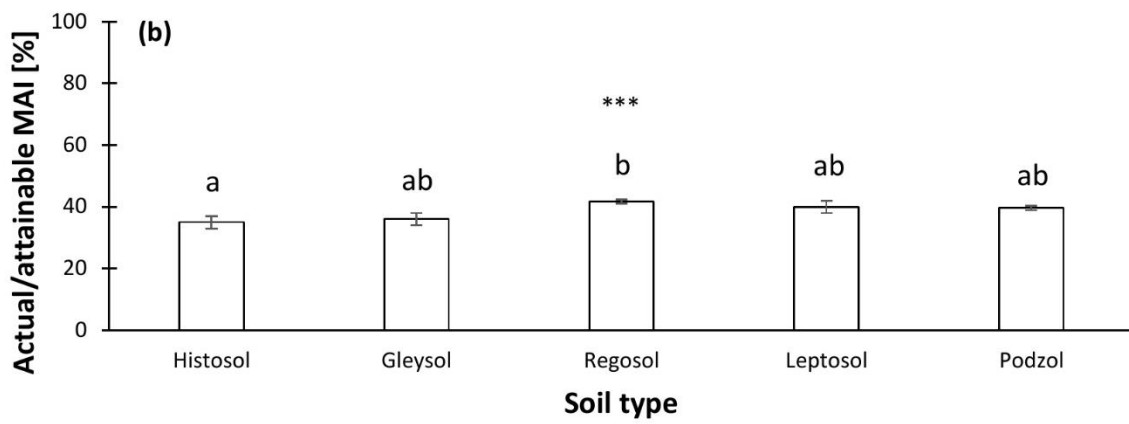

**Figure 6.** Normalized productivity per soil type. (a) Productivity normalized following method 1 (residual mean annual increment - MAI) vs soil type. (b) Productivity normalized following method 2 (actual/attainable mean annual increment – MAI for spruce) vs soil type. In panel a, separate analyses were performed for northern, middle and southern Sweden, as the soil type effects differed among regions. *** indicates significant differences at the $P < 0.01$ level. Error bars represent the s.e.m.

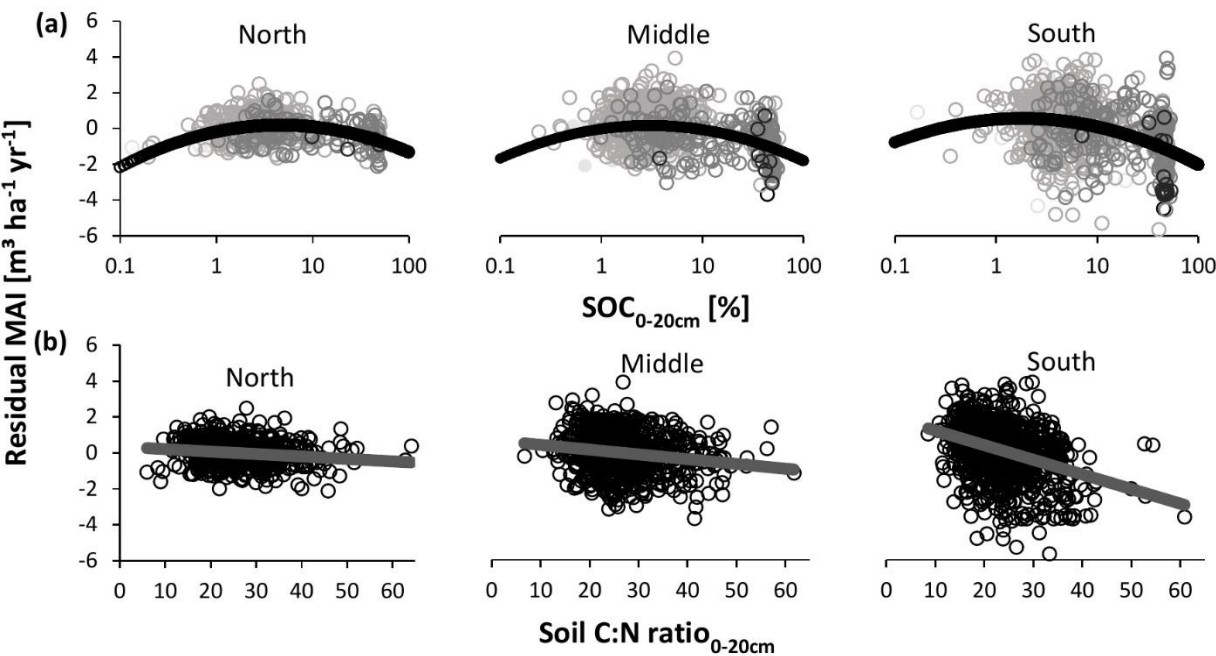

**Figure 7.** Relationship between normalized productivity following method 1 (residual mean annual increment - MAI) and, (a) log-transformed soil organic carbon (SOC) concentration, (b) soil carbon to nitrogen (C:N) ratio at depth 0-20 cm. Separate analyses were performed for northern, middle and southern Sweden, as the SOC and C:N effects differed among regions. Point darkness in panel a represents soil moisture (darker = moister). Statistics corresponding to the panels are presented in Table 2. Note that the horizontal axis for SOC covers a broader range here than in Fig. 1, as SOC varied widely in the Swedish database.

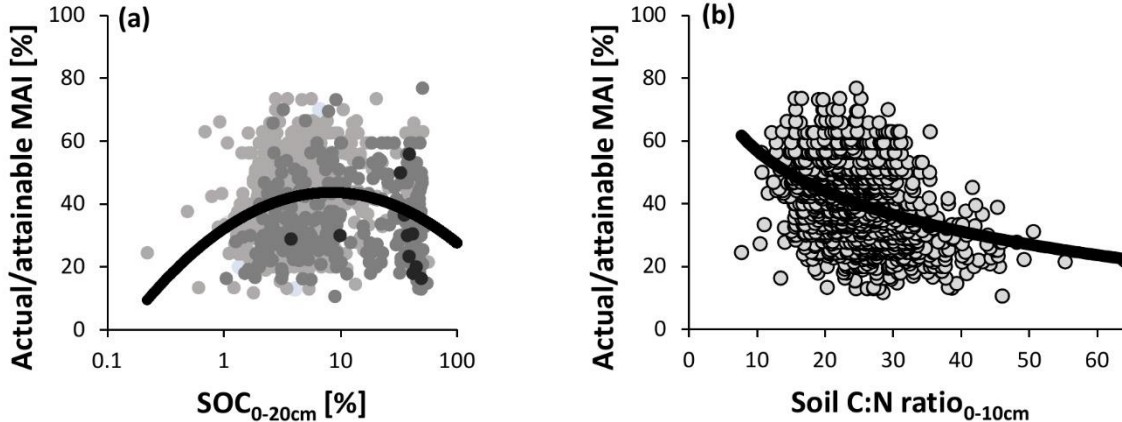

**Figure 8.** Relationship between normalized productivity following method 2 (actual/attainable mean annual increment – MAI for spruce) and, (a) log-transformed soil organic carbon (SOC) concentration, (b) soil carbon to nitrogen (C:N) ratio. Point darkness in panel a represents soil moisture (darker = moister). Statistics corresponding to the panels are presented in Table 2. Note that the horizontal axis for SOC covers a broader range here than in Fig. 1, as SOC varied widely in the Swedish database. Also note that the C:N ratio of the upper 10 cm was used instead of the upper 20 cm here, owing to a better description of variation in the response variable. Even though the C:N ratio roughly decreased southwards (Fig. S1d), it was only weakly correlated with the growing season temperature sum ($r = -0.13$ for C:N$_{0-20cm}$ and $r = -0.28$ for C:N$_{0-10cm}$).

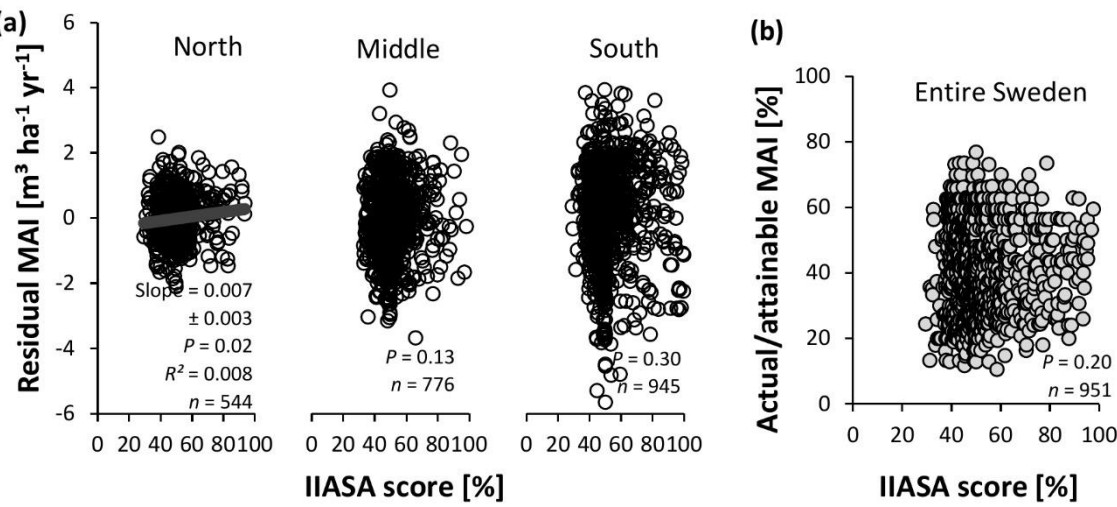

**Figure 9.** Evaluation of the IIASA-metric of constraints on nutrient availability for Swedish conifer forests. (a) Method 1 - association with residual mean annual increments (MAI) of the productivity-climate regression model (Fig. 3a, Eq. (S1) and Table S2), distinguishing northern, middle and southern Sweden. (b) Method 2 - association with actual/attainable MAI for the entire Swedish land area (Fig. 3b). Full line = significant slope ($P < 0.05$).

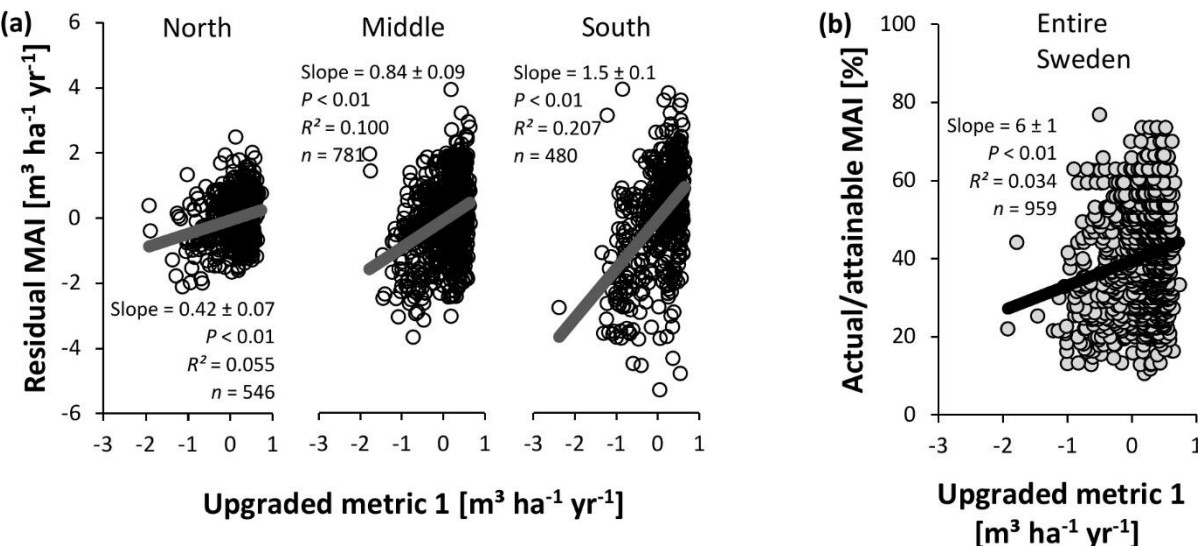

**Figure 10**. Evaluation of upgraded nutrient availability metric 1 for Swedish conifer forests. (a) Method 1 - association with residual mean annual increments (MAI) of the productivity-climate regression model (Fig. 3a, Eq. (S1) and Table S2), distinguishing northern, middle and southern Sweden. (b) Method 2 - association with actual/attainable MAI (Fig. 3b) for the entire Swedish land area. Full line = significant slope ($P < 0.05$).

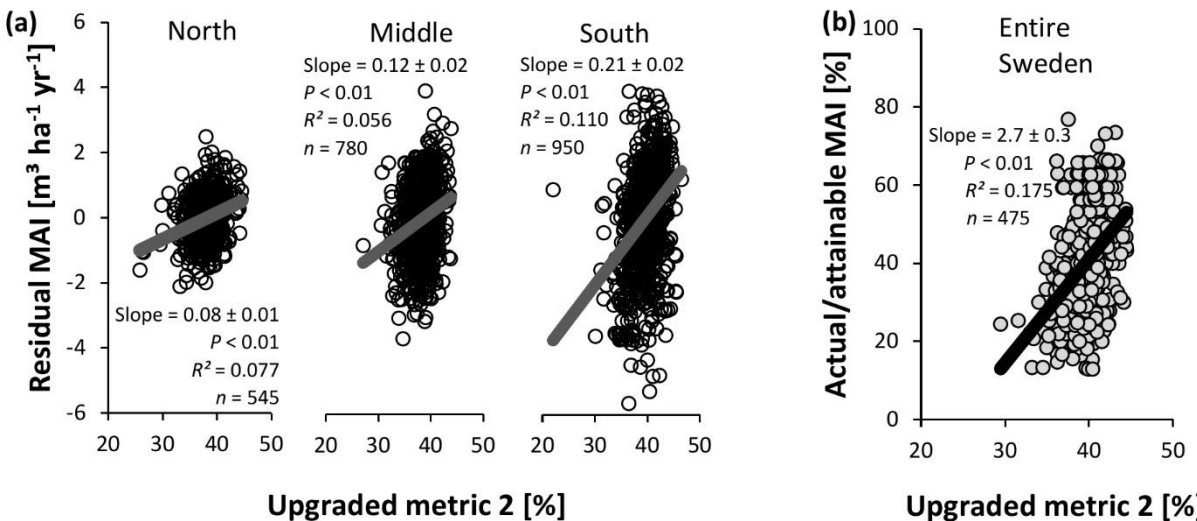

**Figure 11**. Evaluation of upgraded nutrient availability metric 2 for Swedish conifer forests. (a) Method 1 - association with residual mean annual increments (MAI) of the productivity-climate regression model (Fig. 3a, Eq. (S1) and Table S2), distinguishing northern, middle and southern Sweden. (b) Method 2 - association with actual/attainable MAI (Fig. 3b) for the entire Swedish land area. Full line = significant slope ($P < 0.05$).