# Peer review of "The influence of soil properties and nutrients on conifer forest growth in Sweden, and the first steps in developing a nutrient availability metric"

_Biogeosciences, 2017_

## Referee Comment (RC1) · Anonymous Referee #1 · 30 Nov 2017

General Comments There is growing awareness about the influence of nutrient availability on forest C cycling. To better understand this relationship at global scale, there is a real need to integrating information about soil fertility and ecosystem production at broad spatial scales. However, soil is a spatially and temporally complex growing medium, and efforts to develop simple, useful indices of the production potential of soils face numerous challenges. The authors of this paper seek to take the 'first steps' towards creating a single metric to define the relation between soil fertility and forest growth. The authors note that 'we often lack the soil data and metrics needed to accurately account for nutrient availability' and that 'Such a metric does not exist.' This prompts the important question of whether attempting to reduce the complexity of soil fertility into one metric with broad applicability is a realistic pursuit.

The central goal of this paper is to evaluate the utility of individual and combined soil parameters for predicted forest growth. The study leverages soil and forest growth data from > 2500 pine and spruce plots spanning much of Sweden's forest land. Soil x forest growth relations are partitioned along a N-S gradient with productivity normalized for corresponding climatic conditions. The strength of soil x production relations vary between and within regions. Soil moisture class (site wetness) is identified as another critical category that regulates both soil parameters (SOC %) and forest productivity. High SOC (wet sites) with low forest productivity (Fig 5a, 6a) stand out from N-M-S regional gradients. Owing to the strong influence of site wetness on SOC, it is advisable to stratify and analyzed soil fertility x productivity relations by wetness class, not just along N-S gradient. Further, N inputs are 4-5 times higher in Southern than Northern Sweden (Binkley and Hogberg 2016) and thus present a further factor that should be considered with evaluating nutrient and forest productivity across these sites (de Vries et al. 2014).

In addition to examining productivity and soil data from Swedish forests, the authors also evaluate the value of an existing global approach for assigning nutrient constraint metrics. The author's general intent to validate or modify an existing approach makes good sense, but the fact that the selected approach was developed for crops not forests and is 'yet unvalidated' is a bit counterintuitive. Interpretations of the utility of IIASA are reliant on some level of understanding about the strength and limitations of crop-focused IIASA approach. For example, how robust is the approach for predicting plant growth across various soil types? Based on the conditions under which IIASA was developed, how well might it be expected to perform on forest soils and with tree growth? Without clear description of the IIASA (currently some found in Intro, but lacking from Abstract), the paper presents a 'house of cards' based on a relatively unknown, and

possibly weak foundation. Finally, please justify why the simpler approach of originating with 'no' tool and creating one from the ground up with existing data, as outlined in Question 1, is not both adequate and preferable.

Specific and Technical Comments Line 8 Nutrients are one of the factors that influence C cycling. As presented this suggests they are the only or the primary factor. In spite of recen. Line 10 'ideally' The potential value of simplifying ecosystem complexity into interpretable and useful metrics is critically important. However, given the complexity of soils and forest ecosystems, it is worth questioning whether a single fertility metric is an attainable goal. Rather than assuming such an index is 'ideal' this paper might simply evaluate relationships between soil fertility properties and forest growth are robust and broadly useful. Line 10 'Such a metric does not exist' Is the lack of such a metric a major obstacle to understanding of forest production? An objective of this paper should be to quantify how such a metric improves our ability to predict forest growth. Line 12 Insert 'of' between 'combination' and 'soil' Change 'plant' to 'tree' Insert 'forest' before 'nutrient' Line 13-14 Developing this analysis using an unproven metric is counterintuitive. Justify why you did not start out with 'no' tool and build one from the ground up with existing data. Also, explain how the IIASA approach tool developed elsewhere for crops has application for Swedish forests. Clarify how advances from your analysis are independent of potential weakness or limitations of the IIASA approach. Line 19 'not well implemented' Define what that means, especially in light of whether or IIASA provides a useful platform to work with. Line 37/38 Site index is typically an estimate of site productivity, rather than soil fertility per se. The terms are certainly related, but key differences might include factors such as soil depth and hydrology. Line 47 'are more indicative' it's not clear of what. Please revise, clarify. Line 100 What was the size of the forest plots? Are these all plantations or are some natural stands? How are management interventions such as fertilization and thinning addressed and controlled for? What is the average and range of stand age and stand density or basal area?

Line 122 Before suggesting that metric scores can validly 'be assigned to any soil' it

would be useful to know more about the data that went into development of the IIASA and how well those soil types and crop conditions match the forest soils of Sweden. Line 123 This suggests that the IIASA metric is not sensitive to crop specific requirements. Line 284-285 Use soil moisture classes to stratify fertility by productivity. There appear to be abrupt differences between upland and 'wetland' forests that are at least as significant as the N-S gradients. Line 295 The high SOC of wet soils is less a limitation than the anaerobic rooting environment. Line 296 Consider optima for soil saturation, soil oxygen, or soil moisture like the one currently used for pH. This is independent of SOC. Line 301 It is ironic that the only nutrient examined is of limited value. It should be stressed that total N is a much larger pool than the N that's readily available to plants. The complications of inorganic N inputs across the N deposition gradient, and inorganic N losses (leaching and denitrification) across wetness/saturation gradient are likely considerable. Line 304 This relation appears to stem on the large difference in clay in Wet spruce sites (Fig S3e). Again, stratifying wetland from upland forests might isolate and better resolve these patterns. Line 309 Insert and stress "in the absence of direct soil nutrient data" Line 315/316 General differences between arable and Swedish forest soils could be used to define expectations regarding the utility of the IIASA method and probable modifications. Saturated soils, low-clay soils, low N soils might all define some of those differences. Line 324 High water table, low soil oxygen, soil saturation, wetland soils. Table 1 Units are included for everything but TN, and moisture. How was moisture determined and how well does it estimate relevant seasonal moisture dynamics? Table 2 Sample depths are listed for everything but sand and clay. Figure 1 Specify what 'Species-averaged' means or reword so that the caption and figures are interpretable independent of the text. The range of SOC data (1-3%) is nowhere near the ranges presented in Figure 5 and 6 (0.1 to 100%), where the data is presented on a log-scale axis. Explain or resolve the discrepancy.

Citations Binkley D, Högberg P. 2016. Tamm Review: Revisiting the influence of nitrogen deposition on Swedish forests. Forest Ecology and Management 368: 222-239.

de Vries W, Du E, Butterbach-Bahl K. 2014. Short and long-term impacts of nitrogen deposition on carbon sequestration by forest ecosystems. Current Opinion in Environmental Sustainability 9-10: 90-104.

---

## Referee Comment (RC2) · Anonymous Referee #2 · 8 Dec 2017

General comments The authors aim to improve an existing nutrient availability metric for forest ecosystems. They present their results well-structured and clearly. A nutrient availability metric would definitely be of great importance, however, the paper has two major drawbacks: 1. In the introduction the authors write that they aim at developing a globally valid metric while using data from Sweden only. It should be clarified that the upgraded metric is only valid for Sweden. Possibly, a global metric is not achievable at all, since nutrient availability is not limited by the same factors in different ecosystems worldwide. Hence, a metric for Sweden might be usable for other boreal, but not

for tropical ecosystems. These considerations should be discussed. 2. The performance of the metric is bad. Instead of discussing the – non-existing – relationships, the authors should rather discuss the possible reasons for the failure of the metric. One possible reason is data quality. The authors should describe the soil sampling design and the methods used for chemical analyses. Inventory data might not be suitable to find relationships between parameters even though they exist, because (soil) variability would require a large number of replications, which is often not affordable in inventories.

Specific comments l. 17 The coefficients of determination are that poor that you should not write "Normalized productivity increased with decreasing soil C:N ratio, while SOC exhibited an empirical optimum." l. 21 The coefficient of determination of the upgraded metric is still poor and should not be called "a significant fraction". l. 34 and l. 36 "among terrestrial ecosystems" and "global scale" is misleading. You should clarify that in the present paper only Sweden is considered. l. 47 "are more indicative" to what and for what? l. 48 What is meant with "the size of the soil solution"? l. 57 to 60 Yes! Perhaps the fact that a global metric might not be achievable at all, since nutrient availability is not limited by the same factors in different ecosystems worldwide, should be discussed here, too. l. 78 to 84 If the goal is a "global metric", data from the Swedish forest inventory service do not represent "a substantial variation in nutrient availability". Which were the additional variables? P? l. 89 Here you restrict your results on Swedish forests – you should already earlier mention that your goal is a metric for Sweden – not a global metric. l. 94 to 134 List all the parameters (soil, tree/productivity, climate/meteorology) and explain how they were measured. l. 121 to 130 Explain in more detail how the scores were derived and what you can find in the look-up tables. l. 137 to 138 You should state that you call the two alternative ways to calculate normalized productivity "method 1" and "method 2" in the following. l. 138 Name the advantages and drawbacks. l. 179 Is "method 1" referring to the method used to calculated the normalized productivity? l. 193 to 200 You used half of the dataset from southern Sweden to upgrade the metric and tested the upgraded metric

with the rest of the dataset including middle and northern Sweden, right? Did you also upgrade the metric using method 2 (half of the dataset for calibration, half for validation)? l. 216 to217 Why not using both SOC and TN in separate models and choosing the model that better fits the data? l. 225 ". . . related to normalized productivity (Table 2), however, the coefficient of determination was small (0,002 to 0,146)." l. 227 I cannot see from Fig. 5 that "the effect became more pronounced towards the south". l. 232 to 234 According to Table 3 also other variables than SOC, pH and C:N were included in the multiple regression models. Why? l. 239 to 240 On the one hand you write that SOC and C:N "consistently describe a distinct, clear effect on normalized productivity" on the other hand you write that $R^2$ is "at least a few percent" – this is contradictive. Both the figures and the $R^2$ show that there is no distinct, clear effect on normalized productivity. l. 225 to 241 This part should be rewritten since the variance in productivity is hardly explained by the soil variables (see also discussion l. 272 to 313). l. 251 to 252 The relationship between residuals and variables is very weak. l. 254 To my opinion, the results do not show "that SOC, C:N and pH are important factors influencing nutrient availability in Sweden." l. 264 to 265 From Figure 8 no enhancement of the upgraded metric can be deduced. The performance is still bad. l. 272 to 313 The variance in productivity is not explained well by the soil variables. Instead of discussing the (significant but very weak) relationships, you should rather discuss why you do not find relationships were you would expect them. What about the soil sampling design and the methods used for chemical analyses? You did not describe them in the material and methods section. Could the sampling design (number of replications) explain the bad model performance? How is the variability of soil variables in Sweden? l. 323 . . . but again with a very small coefficient of determination. l. 330 ". . . the nutrient availability metric was intended to be improved by . . ." l. 335 I cannot agree. The relationships were significant but only a very small part of the variance was explained by the variables used. l. 336 to 339 Why are especially stable N isotope signatures and ion exchange resin bags of interest? What about other methods? Table 2 Are you sure that $R^2$ is the same for all three regions for the parameters SOC, N stock, sand, clay

and pHKCl? This is quite unrealistic and the data shown in Figure 5 (SOC) lead to the assumption that R2 of the North is largest and that of the South smallest. In line 226 to 227 you write that the effect becomes more pronounced towards the south (C:N), however, in Figure 5 the relationship is worse for the south than for the other regions and a larger R2 – as written in Table 2 – seems to be quite unrealistic. Table 4 and 5 The coefficients of determination are similar and do not point on a better implementation of the parameters in the upgraded metric. Figure 1 I have problems understanding the legend of Figure 1.

Technical corrections l. 12 ...to test which combination of soil factors... l. 63 to 64 rephrase this sentence to avoid the twofold use of "recent(ly)" l. 96 explain the abbreviations l. 118 mass stock [kg m-2]; if really the mass is meant, the formula is wrong l. 157 "we therefore we split" delete one "we" l. 164 "SOC" – stock or content? "Total N" – total N content? "N stock" – total N stock? l. 167 Name the software used.

Please also note the supplement to this comment:
https://www.biogeosciences-discuss.net/bg-2017-372/bg-2017-372-RC2-supplement.pdf
* * *

---

## Author Comment (AC1) · 27 Dec 2017

Here, we provide a first short reply to the main points raised by Referee #1. Point-by-point responses to all General, Specific and Technical Comments and changes in the manuscript will be presented in a later stage of the peer review process.

Sincerely,

Kevin Van Sundert, on behalf of all co-authors.

[Figure]

REPLIES TO GENERAL COMMENTS

We thank the reviewer for his/her supportive and constructive assessment of our manuscript. We agree with most remarks made and believe incorporating the suggestions will further improve our manuscript.

One issue we would like to clarify upfront, and that may not have been sufficiently clear yet in the manuscript, is that the primary aim of the paper is not to improve predictions of forest growth (although our work may also help in that regard), but to investigate which soil properties and nutrients are the most critical determinants of conifer forest productivity. This as a first step in developing a metric of nutrient availability based on soil data. Tree productivity (normalized for climate) here serves as the most straightforward indicator of nutrient availability.

COMMENT: "The central goal of this paper is to evaluate the utility of individual and combined soil parameters for predicted forest growth. The study leverages soil and forest growth data from > 2500 pine and spruce plots spanning much of Sweden's forest land. Soil x forest growth relations are partitioned along a N-S gradient with productivity normalized for corresponding climatic conditions. The strength of soil x production relations vary between and within regions. Soil moisture class (site wetness) is identified as another critical category that regulates both soil parameters (SOC %) and forest productivity. High SOC (wet sites) with low forest productivity (Fig 5a, 6a) stand out from N-M-S regional gradients. Owing to the strong influence of site wetness on SOC, it is advisable to stratify and analyzed soil fertility x productivity relations by wetness class, not just along N-S gradient."

We fully agree with Referee #1's recommendation to perform our analyses for the separate soil moisture categories (based on the observation that soil moisture is a critical factor (Figs. 5 and 6)). We therefore performed a new analysis where we distinguished the four moisture classes represented in the database. This analysis confirmed the results and parameter estimates obtained in the previous analysis (Table 2 vs the attached Table in the supplement). Hence, these results indicate that the observed patterns are very robust across the database. We will include this additional analysis in the revised manuscript.

COMMENT: "N inputs are 4-5 times higher in Southern than Northern Sweden (Binkley and Hogberg 2016) and thus present a further factor that should be considered with evaluating nutrient and forest productivity across these sites (de Vries et al. 2014)."

Referee #1 correctly states that considering variables such as N deposition may further improve growth predictions. However, exploring the influence of N deposition is outside the scope of our study. Including N deposition as an extra predictor would even complicate our understanding of the influence of soil properties, because for example pH and soil C:N ratio are gradually altered by N deposition. Nonetheless, if the editor so wishes, we are willing to test the influence of N deposition separately and present it in the revised manuscript.

COMMENT: "In addition to examining productivity and soil data from Swedish forests, the authors also evaluate the value of an existing global approach for assigning nutrient constraint metrics. The author's general intent to validate or modify an existing approach makes good sense, but the fact that the selected approach was developed for crops not forests and is 'yet unvalidated' is a bit counterintuitive. Interpretations of the utility of IIASA are reliant on some level of understanding about the strength and limitations of crop-focused IIASA approach. For example, how robust is the approach for predicting plant growth across various soil types? Based on the conditions under which IIASA was developed, how well might it be expected to perform on forest soils and with tree growth? Without clear description of the IIASA (currently some found in Intro, but lacking from Abstract), the paper presents a 'house of cards' based on a relatively unknown, and possibly weak foundation. Finally, please justify why the simpler approach of originating with 'no' tool and creating one from the ground up with existing data, as outlined in Question 1, is not both adequate and preferable."
We used the IIASA-metric of constraints on nutrient availability because the structures of its formulas (Eqs.(6-9)) reflect general mechanisms that link soil properties to nutrient availability. Soil pH for example shows a typical optimum effect on nutrient availability, while SOC and TEB have a direct positive non-linear influence (IIASA and FAO, 2012). The final weighing of the four partial scores (Eq. (10)) finds its rationale in the idea that if a certain soil property is particularly suboptimal, it will be the most important determinant of productivity, with less influence of the other soil properties that are within the optimal range. This way of weighing can be considered as a type of interaction, but one that cannot be implemented in a simple linear regression model. Hence, our main reason for adopting the IIASA-metric as a starting point is that, in spite of its simplicity, it is based on theoretical considerations. The multiple regression equations we obtained, on the other hand, would be of less use for building a nutrient availability metric because such an entirely empirical approach would not allow for later updates of parameters or model structures based on data from other ecosystems.

We understand that the choice to start with the IIASA-metric in particular is somewhat counterintuitive, as this metric was initially developed for evaluating the soil fertility of agricultural ecosystems. Species and soil conditions of such ecosystems indeed greatly differ from the boreal forests investigated in the present study (e.g. we anticipated in advance that N availability would not yet be sufficiently explicit implemented, given the absence of variables such as C:N). However, we argue that this metric still offers the best option to serve as a starting point, because of (i) the reasons mentioned in the previous paragraph (simplicity, inclusion of mechanisms and interactions, and potential for updates), and because (ii) it is, to our knowledge, the only attempt so far to develop a generic nutrient availability metric (except for an older productivity index presented in Riquier et al., 1970, which however only considers linear effects without interactions). In other words, we had to rely on a metric originally developed for arable land, because nutrient metrics for other ecosystem types simply do not exist. In the revised manuscript, we will further clarify the reasons to start with the IIASA metric.

[Figure]

Citations:

IIASA and FAO.: Global Agro-ecological Zones (GAEZ v3.0), International Institute for Applied Systems Analysis, Laxenburg, Austria and Food and Agricultural Organization of the United Nations, Rome, Italy, 2012.

Riquier, J., Bramao, D.L., and Cornet, J.P.: A new system of soil appraisal in terms of actual and potential productivity, Food and Agriculture Organization of the United Nations, Rome, Italy, 1970.

Please also note the supplement to this comment:
https://www.biogeosciences-discuss.net/bg-2017-372/bg-2017-372-AC1-supplement.pdf

[Figure]

**Supplement:**

Subject: Reply to General and Specific Comments of Ref#1

Here, we provide a first short reply to the main points raised by Referee #1. Point-by-point responses to all General, Specific and Technical Comments and changes in the manuscript will be presented in a later stage of the peer review process.

Sincerely,

Kevin Van Sundert, on behalf of all co-authors.

**REPLIES TO GENERAL COMMENTS**

We thank the reviewer for his/her supportive and constructive assessment of our manuscript. We agree with most remarks made and believe incorporating the suggestions will further improve our manuscript.

One issue we would like to clarify upfront, and that may not have been sufficiently clear yet in the manuscript, is that the primary aim of the paper is not to improve predictions of forest growth (although our work may also help in that regard), but to investigate which soil properties and nutrients are the most critical determinants of conifer forest productivity. This as a first step in developing a metric of nutrient availability based on soil data. Tree productivity (normalized for climate) here serves as the most straightforward indicator of nutrient availability.

COMMENT: *"The central goal of this paper is to evaluate the utility of individual and combined soil parameters for predicted forest growth. The study leverages soil and forest growth data from > 2500 pine and spruce plots spanning much of Sweden's forest land. Soil x forest growth relations are partitioned along a N-S gradient with productivity normalized for corresponding climatic conditions. The strength of soil x production relations vary between and within regions. Soil moisture class (site wetness) is identified as another critical category that regulates both soil parameters (SOC %) and forest productivity. High SOC (wet sites) with low forest productivity (Fig 5a, 6a) stand out from N-M-S regional gradients. Owing to the strong influence of site wetness on SOC, it is advisable to stratify and analyzed soil fertility x productivity relations by wetness class, not just along N-S gradient."*

We fully agree with Referee #1's recommendation to perform our analyses for the separate soil moisture categories (based on the observation that soil moisture is a critical factor (Figs. 5 and 6)). We therefore performed a new analysis where we distinguished the four moisture classes represented in the database. This analysis confirmed the results and parameter estimates obtained in the previous analysis (Table 2 vs the attached Table in the supplement). Hence, these results indicate that the observed patterns are very robust across the database. We will include this additional analysis in the revised manuscript.

COMMENT: *"N inputs are 4-5 times higher in Southern than Northern Sweden (Binkley and Hogberg 2016) and thus present a further factor that should be considered with evaluating nutrient and forest productivity across these sites (de Vries et al. 2014)."*

Referee #1 correctly states that considering variables such as N deposition may further improve growth predictions. However, exploring the influence of N deposition is outside the scope of our study. Including N deposition as an extra predictor would even complicate our understanding of the influence of soil properties, because for example pH and soil C:N ratio

are gradually altered by N deposition. Nonetheless, if the editor so wishes, we are willing to test the influence of N deposition separately and present it in the revised manuscript.

COMMENT: *"In addition to examining productivity and soil data from Swedish forests, the authors also evaluate the value of an existing global approach for assigning nutrient constraint metrics. The author's general intent to validate or modify an existing approach makes good sense, but the fact that the selected approach was developed for crops not forests and is 'yet unvalidated' is a bit counterintuitive. Interpretations of the utility of IIASA are reliant on some level of understanding about the strength and limitations of crop-focused IIASA approach. For example, how robust is the approach for predicting plant growth across various soil types? Based on the conditions under which IIASA was developed, how well might it be expected to perform on forest soils and with tree growth? Without clear description of the IIASA (currently some found in Intro, but lacking from Abstract), the paper presents a 'house of cards' based on a relatively unknown, and possibly weak foundation. Finally, please justify why the simpler approach of originating with 'no' tool and creating one from the ground up with existing data, as outlined in Question 1, is not both adequate and preferable."*

We used the IIASA-metric of constraints on nutrient availability because the structures of its formulas (Eqs.(6-9)) reflect general mechanisms that link soil properties to nutrient availability. Soil pH for example shows a typical optimum effect on nutrient availability, while SOC and TEB have a direct positive non-linear influence (IIASA and FAO, 2012). The final weighing of the four partial scores (Eq. (10)) finds its rationale in the idea that if a certain soil property is particularly suboptimal, it will be the most important determinant of productivity, with less influence of the other soil properties that are within the optimal range. This way of weighing can be considered as a type of interaction, but one that cannot be implemented in a simple linear regression model. Hence, our main reason for adopting the IIASA-metric as a starting point is that, in spite of its simplicity, it is based on theoretical considerations. The multiple regression equations we obtained, on the other hand, would be of less use for building a nutrient availability metric because such an entirely empirical approach would not allow for later updates of parameters or model structures based on data from other ecosystems.

We understand that the choice to start with the IIASA-metric in particular is somewhat counterintuitive, as this metric was initially developed for evaluating the soil fertility of agricultural ecosystems. Species and soil conditions of such ecosystems indeed greatly differ from the boreal forests investigated in the present study (e.g. we anticipated in advance that N availability would not yet be sufficiently explicit implemented, given the absence of variables such as C:N). However, we argue that this metric still offers the best option to serve as a starting point, because of (i) the reasons mentioned in the previous paragraph (simplicity, inclusion of mechanisms and interactions, and potential for updates), and because (ii) it is, to our knowledge, the only attempt so far to develop a generic nutrient availability metric (except for an older productivity index presented in Riquier *et al.*, 1970, which however only considers linear effects without interactions). In other words, we had to rely on a metric originally developed for arable land, because nutrient metrics for other ecosystem types simply do not exist. In the revised manuscript, we will further clarify the reasons to start with the IIASA metric.

Citations:

IIASA and FAO.: Global Agro-ecological Zones (GAEZ v3.0), International Institute for Applied Systems Analysis, Laxenburg, Austria and Food and Agricultural Organization of the United Nations, Rome, Italy, 2012.

Riquier, J., Bramao, D.L., and Cornet, J.P.: A new system of soil appraisal in terms of actual and potential productivity, Food and Agriculture Organization of the United Nations, Rome, Italy, 1970.

**Supplementary Table.** Associations between single soil variables and normalized productivity of Table 2, stratified by soil moisture class (dry to moist). Significance (*P*-values) of single soil variable effects on residual productivity (mean annual increment - MAI [m³ ha⁻¹ yr⁻¹]) and actual/attainable MAI (for spruce only) across Sweden are given. For (near) significant variables (i.e. $P < 0.10$), parameter estimates ± s.e.m. and the proportion of variation explained ($R^2$) are shown as well. Abbreviations: N = north; M = middle; S = south; SOC = soil organic carbon concentration; C:N = soil carbon to nitrogen ratio; TEB = total exchangeable bases; quad = parameter estimate for quadratic term; lin = parameter estimate for linear term of a quadratic function.

| Normalized productivity response | Region | ln SOC 0-20cm [%] | | | |
|---|---|---|---|---|---|
| | | **Dry** | **Fresh** | **Fresh-moist** | **Moist** |
| Residual MAI (method 1) | N | $P = 0.32$ | quad = -0.13 ± 0.02 $P < 0.01$ lin = 0.39 ± 0.10 $P < 0.01$ intercept = -0.05 ± 0.10 $R^2_{tot} = 0.052$ | quad = -0.22 ± 0.06 $P < 0.01$ lin = 0.9 ± 0.3 $P < 0.01$ intercept = -0.8 ± 0.3 $P = 0.01$ $R^2_{tot} = 0.106$ | quad = -3 ± 1 $P = 0.03$ lin = 15 ± 7 $P = 0.03$ intercept = -23 ± 10 $P = 0.03$ $R^2_{tot} = 0.314$ |
| | M | $P = 0.75$ | quad = -0.13 ± 0.02 $P < 0.01$ lin = 0.33 ± 0.09 $P < 0.01$ intercept = 0.17 ± 0.09 $P = 0.06$ $R^2_{tot} = 0.052$ | quad = -0.22 ± 0.06 $P < 0.01$ lin = 0.8 ± 0.1 $P < 0.01$ intercept = -0.9 ± 0.3 $P < 0.01$ $R^2_{tot} = 0.106$ | quad = -3 ± 1 $P = 0.03$ lin = 13 ± 5 $P = 0.03$ intercept = -14 ± 6 $P = 0.02$ $R^2_{tot} = 0.314$ |
| | S | $P = 0.10$ | quad = -0.13 ± 0.02 $P < 0.01$ slope = 0.2 ± 0.1 $P = 0.09$ intercept = 0.5 ± 0.1 $P < 0.01$ $R^2_{tot} = 0.052$ | quad = -0.22 ± 0.06 $P < 0.01$ lin = 0.7 ± 0.3 $P = 0.03$ intercept = -0.6 ± 0.4 $P = 0.19$ $R^2_{tot} = 0.106$ | quad = -3 ± 1 $P < 0.01$ lin = 13 ± 6 $P = 0.05$ intercept = -16 ± 9 $P = 0.07$ $R^2_{tot} = 0.314$ |
| Actual/attainable MAI (method 2) | entire Sweden | $P = 0.66$ | quad = -1.6 ± 0.5 $P < 0.01$ lin = 9 ± 2 $P < 0.01$ intercept = 34 ± 2 $P < 0.01$ $R^2 = 0.043$ | quad = -3.4 ± 0.8 $P < 0.01$ lin = 16 ± 4 $P < 0.01$ intercept = 19 ± 4 $P < 0.01$ $R^2 = 0.056$ | $P = 0.53$ |

| Normalized productivity response | Region | ln N stock 0-20cm [$g\ m^{-2}$] | | | |
|---|---|---|---|---|---|
| | | **Dry** | **Fresh** | **Fresh-moist** | **Moist** |
| Residual MAI (method 1) | N | $P = 0.56$ | slope $= 0.22 \pm 0.06$ $P < 0.01$ intercept $= -1.0 \pm 0.3$ $P < 0.01$ $R^2_{tot} = 0.007$ | slope $= 0.26 \pm 0.08$ $P < 0.01$ intercept $= -1.5 \pm 0.4$ $P < 0.01$ $R^2_{tot} = 0.012$ | $P = 0.78$ |
| | M | $P = 0.56$ | slope $= 0.22 \pm 0.06$ $P < 0.01$ intercept $= -1.0 \pm 0.3$ $P < 0.01$ $R^2_{tot} = 0.007$ | slope $= 0.26 \pm 0.08$ $P < 0.01$ intercept $= -1.5 \pm 0.4$ $P < 0.01$ $R^2_{tot} = 0.012$ | $P = 0.78$ |
| | S | $P = 0.56$ | slope $= 0.22 \pm 0.06$ $P < 0.01$ intercept $= -1.0 \pm 0.3$ $P < 0.01$ $R^2_{tot} = 0.007$ | slope $= 0.26 \pm 0.08$ $P < 0.01$ intercept $= -1.5 \pm 0.4$ $P < 0.01$ $R^2_{tot} = 0.012$ | $P = 0.78$ |
| Actual/attainable MAI (method 2) | entire Sweden | $P = 0.10$ | slope $= 11 \pm 1$ $P < 0.01$ intercept $= -19 \pm 6$ $P < 0.01$ $R^2 = 0.149$ | slope $= 10 \pm 1$ $P < 0.01$ intercept $= -2 \pm 7$ $P < 0.01$ $R^2 = 0.212$ | $P = 0.12$ |

| Normalized productivity response | Region | C:N 0-20cm | | | |
|---|---|---|---|---|---|
| | | **Dry** | **Fresh** | **Fresh-moist** | **Moist** |
| Residual MAI (method 1) | N | $P = 0.20$ | slope $= -0.007 \pm 0.004$ $P = 0.09$ intercept $= 0.3 \pm 0.1$ $P < 0.01$ $R^2 = 0.005$ | slope $= -0.040 \pm 0.008$ $P < 0.01$ intercept $= 0.8 \pm 0.2$ $P < 0.01$ $R^2_{tot} = 0.184$ | $P = 0.13$ |
| | M | $P = 0.22$ | slope $= -0.015 \pm 0.006$ $P = 0.02$ intercept $= 0.7 \pm 0.2$ $P < 0.01$ $R^2_{tot} = 0.008$ | slope $= -0.039 \pm 0.010$ $P < 0.01$ intercept $= 0.5 \pm 0.3$ $P = 0.09$ $R^2_{tot} = 0.067$ | $P = 0.39$ |
| | S | $P = 0.28$ | slope $= -0.048 \pm 0.009$ $P < 0.01$ intercept $= 1.5 \pm 0.2$ $P < 0.01$ $R^2_{tot} = 0.041$ | slope $= -0.09 \pm 0.01$ $P < 0.01$ intercept $= 1.8 \pm 0.4$ $P < 0.01$ $R^2_{tot} = 0.170$ | $P = 0.14$ |
| Actual/attainable MAI (method 2) | entire Sweden | N/A | N/A | N/A | N/A |

| Normalized productivity response | Region | ln C:N 0-10 cm | | | |
|---|---|---|---|---|---|
| | | Dry | Fresh | Fresh-moist | Moist |
| Residual MAI (method 1) | N | N/A | N/A | N/A | N/A |
| | M | N/A | N/A | N/A | N/A |
| | S | N/A | N/A | N/A | N/A |
| Actual/attainable MAI (method 2) | entire Sweden | slope $= -33 \pm 15$ $P = 0.04$ intercept $= 143 \pm 47$ $P < 0.01$ $R^2 = 0.174$ | slope $= -17 \pm 2$ $P < 0.01$ intercept $= 96 \pm 6$ $P < 0.01$ $R^2 = 0.112$ | slope $= -19 \pm 3$ $P < 0.01$ intercept $= 95 \pm 8$ $P < 0.01$ $R^2 = 0.149$ | $P = 0.35$ |

| Normalized productivity response | Region | Mineral soil sand [%] Dry | Fresh | Fresh-moist | Moist |
|---|---|---|---|---|---|
| Residual MAI (method 1) | N | slope $= 0.021 \pm 0.009$ $P = 0.02$ intercept $= -1.3 \pm 0.8$ $P = 0.10$ $R^2_{tot} = 0.101$ | slope $= 0.004 \pm 0.001$ $P < 0.01$ intercept $= -0.2 \pm 0.1$ $P = 0.04$ $R^2_{tot} = 0.006$ | slope $= 0.006 \pm 0.002$ $P < 0.01$ intercept $= -0.4 \pm 0.1$ $P < 0.01$ $R^2_{tot} = 0.015$ | $P = 0.64$ |
|  | M | slope $= 0.021 \pm 0.009$ $P = 0.02$ intercept $= -1.0 \pm 0.6$ $P = 0.09$ $R^2_{tot} = 0.101$ | slope $= 0.004 \pm 0.001$ $P < 0.01$ intercept $= -0.15 \pm 0.09$ $P = 0.10$ $R^2_{tot} = 0.006$ | slope $= 0.006 \pm 0.002$ $P < 0.01$ intercept $= -0.6 \pm 0.1$ $P < 0.01$ $R^2_{tot} = 0.015$ | $P = 0.64$ |
|  | S | slope $= 0.021 \pm 0.009$ $P = 0.02$ intercept $= -1.6 \pm 0.5$ $P < 0.01$ $R^2_{tot} = 0.101$ | slope $= 0.004 \pm 0.001$ $P < 0.01$ intercept $= -0.01 \pm 0.08$ $P = 0.86$ $R^2_{tot} = 0.006$ | slope $= 0.006 \pm 0.002$ $P < 0.01$ intercept $= -0.5 \pm 0.1$ $P < 0.01$ $R^2_{tot} = 0.015$ | $P = 0.64$ |
| Actual/attainable MAI (method 2) | entire Sweden | $P = 0.44$ | slope $= -0.07 \pm 0.02$ $P < 0.01$ intercept $= 47 \pm 1$ $P < 0.01$ $R^2 = 0.014$ | slope $= -0.04 \pm 0.03$ $P = 0.09$ intercept $= 37 \pm 1$ $P < 0.01$ $R^2 = 0.006$ | $P = 0.25$ |

| Normalized productivity response | Region | Mineral soil clay [%] | | | |
|---|---|---|---|---|---|
| | | **Dry** | **Fresh** | **Fresh-moist** | **Moist** |
| Residual MAI (method 1) | N | slope = 0.10 ± 0.05
$P = 0.03$
intercept
 = -0.8 ± 0.3
$P = 0.02$
$R^2_{tot} = 0.072$ | slope = 0.011 ± 0.005
$P = 0.02$
intercept
 = 0.06 ± 0.04
$P = 0.14$
$R^2_{tot} = 0.003$ | $P = 0.97$ | slope = 0.05 ± 0.02
$P = 0.03$
intercept
 = -0.6 ± 0.2
$P = 0.03$
$R^2_{tot} = 0.124$ |
| | M | slope = 0.10 ± 0.05
$P = 0.03$
intercept
 = -0.8 ± 0.3
$P = 0.02$
$R^2_{tot} = 0.072$ | slope = 0.011 ± 0.005
$P = 0.02$
intercept
 = 0.06 ± 0.04
$P = 0.14$
$R^2_{tot} = 0.003$ | $P = 0.97$ | slope = 0.05 ± 0.02
$P = 0.03$
intercept
 = -0.6 ± 0.2
$P = 0.03$
$R^2_{tot} = 0.124$ |
| | S | slope = 0.10 ± 0.05
$P = 0.03$
intercept
 = -0.8 ± 0.3
$P = 0.02$
$R^2_{tot} = 0.072$ | slope = 0.011 ± 0.005
$P = 0.02$
intercept
 = 0.06 ± 0.04
$P = 0.14$
$R^2_{tot} = 0.003$ | $P = 0.97$ | slope = 0.05 ± 0.02
$P = 0.03$
intercept
 = -0.6 ± 0.2
$P = 0.03$
$R^2_{tot} = 0.124$ |
| Actual/attainable MAI (method 2) | entire Sweden | $P = 0.80$ | slope = 0.21 ± 0.07
$P < 0.01$
intercept
 = 41.8 ± 0.7
$P < 0.01$
$R^2 = 0.011$ | $P = 0.30$ | $P = 0.84$ |

| Normalized productivity response | Region | ln TEB stock 0-20cm [cmol$_+$ m$^{-2}$] | | | |
|---|---|---|---|---|---|
| | | **Dry** | **Fresh** | **Fresh-moist** | **Moist** |
| Residual MAI (method 1) | N | $P = 0.73$ | slope = 0.22 ± 0.05 $P < 0.01$ intercept = -0.6 ± 0.2 $P < 0.01$ $R^2 = 0.039$ | $P = 0.13$ | $P = 0.39$ |
| | M | $P = 0.73$ | slope = 0.20 ± 0.07 $P < 0.01$ intercept = -0.4 ± 0.2 $P = 0.09$ $R^2 = 0.015$ | $P = 0.13$ | $P = 0.39$ |
| | S | $P = 0.73$ | slope = -0.20 ± 0.06 $P < 0.01$ intercept = 1.3 ± 0.3 $P < 0.01$ $R^2 = 0.014$ | $P = 0.13$ | $P = 0.39$ |
| Actual/attainable MAI (method 2) | entire Sweden | $P = 0.51$ | slope = 2.7 ± 0.7 $P < 0.01$ intercept  = 32 ± 3 $P < 0.01$ $R^2 = 0.025$ | slope = 3.6 ± 0.8 $P < 0.01$ intercept  = 20 ± 4 $P < 0.01$ $R^2 = 0.060$ | $P = 0.68$ |

| Normalized productivity response | Region | $pH_{KCl}$ 0-20cm | | | |
|---|---|---|---|---|---|
| | | **Dry** | **Fresh** | **Fresh-moist** | **Moist** |
| Residual MAI (method 1) | N | $P = 0.80$ | quad = -0.54 ± 0.07 $P < 0.01$ lin = 3.9 ± 0.5 $P < 0.01$ intercept = -7 ± 1 $P < 0.01$ $R^2_{tot} = 0.043$ | quad = -0.4 ± 0.1 $P < 0.01$ lin = 3.4 ± 0.8 $P < 0.01$ intercept = -7 ± 2 $P < 0.01$ $R^2_{tot} = 0.121$ | quad = -1.8 ± 0.8 $P = 0.03$ lin = 12 ± 5 $P = 0.03$ intercept = -22 ± 9 $P = 0.02$ $R^2_{tot} = 0.415$ |
| | M | $P = 0.80$ | quad = -0.54 ± 0.07 $P < 0.01$ lin = 4.2 ± 0.5 $P < 0.01$ intercept = -8 ± 1 $P < 0.01$ $R^2_{tot} = 0.043$ | quad = -0.4 ± 0.1 $P < 0.01$ lin = 3.5 ± 0.8 $P < 0.01$ intercept = -8 ± 1 $P < 0.01$ $R^2_{tot} = 0.121$ | quad = -1.8 ± 0.8 $P = 0.03$ lin = 12 ± 5 $P = 0.03$ intercept = -23 ± 9 $P = 0.02$ $R^2_{tot} = 0.415$ |
| | S | $P = 0.80$ | quad = -0.54 ± 0.07 $P < 0.01$ lin = 4.2 ± 0.5 $P < 0.01$ intercept = -8 ± 1 $P < 0.01$ $R^2_{tot} = 0.043$ | quad = -0.4 ± 0.1 $P < 0.01$ lin = 3.8 ± 0.8 $P < 0.01$ intercept = -8 ± 1 $P < 0.01$ $R^2_{tot} = 0.121$ | quad = -1.8 ± 0.8 $P = 0.03$ lin = 12 ± 5 $P = 0.03$ intercept = -21 ± 9 $P = 0.03$ $R^2_{tot} = 0.415$ |
| Actual/attainable MAI (method 2) | entire Sweden | $P = 0.13$ | $P = 0.21$ | slope = 4 ± 2 $P = 0.02$ intercept = 23 ± 5 $P < 0.01$ $R^2 = 0.016$ | $P = 0.45$ |

---

## Author Comment (AC2) · 27 Dec 2017

Here, we present replies to the main comments of Referee #2. Point-by-point responses to all General, Specific and Technical Comments and changes in the manuscript will be presented in a later stage of the peer review process.

Sincerely,

Kevin Van Sundert, on behalf of all co-authors.

[Figure]

REPLIES TO GENERAL COMMENTS

We thank the reviewer for the insightful comments. We agree that the nutrient availability metric in its current form cannot just be applied to ecosystems globally, and that the development of a global metric is challenging, e.g. in part because of differential nutrient limitation across ecosystems. We also agree that the low $R^2$ of the upgraded metric indicates that it may not be good enough yet to describe a desirable proportion of the variation in nutrient availability, even for boreal forests, but this remains to be tested with other datasets. Instead, it should be seen as a starting point in a development process. We will discuss the type of data needed for further development and evaluations of the metric in the manuscript. For the soil properties and nutrients, however, we still find that, despite the limited $R^2$, our findings are generally robust (see below). In the discussion, we will therefore add a section on robustness of the associations we found (also see the reply to Ref#1's comments).

COMMENT: "In the introduction the authors write that they aim at developing a globally valid metric while using data from Sweden only. It should be clarified that the upgraded metric is only valid for Sweden. Possibly, a global metric is not achievable at all, since nutrient availability is not limited by the same factors in different ecosystems worldwide. Hence, a metric for Sweden might be usable for other boreal, but not for tropical ecosystems. These considerations should be discussed."

We agree that a metric developed or improved based on data from Swedish forests alone cannot be extrapolated to be used as a global nutrient metric. We will make this more explicit in our manuscript. Even though our ultimate aim is indeed to develop a globally applicable metric, the metric upgrade presented in this manuscript should only be seen as a first step in the development process, i.e. a step where we indicate the need for such a metric and show the development approach with its advantages and disadvantages. Furthermore, we will add a section in the discussion on additional data needed to develop a globally applicable metric, such as data from local gradients in nutrient availability (which have the advantage that no normalization for climate

is needed, see below). Evidently, further updates of the metric based on data from other ecosystems are also needed, not only because the proportion of the variation explained is rather low, even within Sweden (see below), but also because, as Referee #2 mentions, different ecosystems can be limited by different nutrients, while the current metric was developed based on only N limited forests.

Although we agree with Referee #2 that differential nutrient limitation across ecosystems poses an additional challenge to the development of a nutrient metric, we anticipate that the implementation of both N and P availability and perhaps other nutrients into one single metric is not necessarily an impossible task. In theory, it is perfectly possible to include multiple variables such as C:N (mainly relating to N availability), pH (among others a critical factor controlling P availability) and exchangeable bases in one single metric. In fact, the IIASA-metric is particularly useful in this regard, as it gives more weight to the soil factor with the lowest score and could therefore account for the type of nutrient limitation. For instance, if C:N is high, indicating N limitation, the metric score will be substantially reduced by this high C:N, while at low C:N other limiting factors can dominate the metric score. Based on an example like this one, we will discuss the challenge of differential nutrient limitation in the manuscript.

COMMENT: "The performance of the metric is bad. Instead of discussing the – non-existing – relationships, the authors should rather discuss the possible reasons for the failure of the metric. One possible reason is data quality. The authors should describe the soil sampling design and the methods used for chemical analyses. Inventory data might not be suitable to find relationships between parameters even though they exist, because (soil) variability would require a large number of replications, which is often not affordable in inventories."

First, the low $R^2$s lead us to ask the question how large they should be in order to sufficiently describe variation in nutrient availability. We agree with the referee that the performance of the upgraded metric is likely still too low for applications (although it can already describe > 20% of the variation in southern Sweden), but we should be

cautious with using $R^2$ as the main criterion to evaluate the metric: the unexplained variation depends on multiple factors not directly related to nutrient availability that could not be accounted for in our analyses (e.g. imperfect productivity estimates and the normalization procedure add to this unexplained variation, see below). Instead, we also considered the significance of the relationships, and the presence or absence of a remaining association between residuals and soil variables in the metric as additional criteria for testing its performance. Besides these criteria, the metric should obviously also succeed in explaining variation in complementary datasets. We thank the reviewer for this critical assessment, which inspired us to include in the revised manuscript a paragraph discussing the criteria of a good nutrient availability metric.

Besides factors such as management that increase variation, we think that the low $R^2$ values also follow from (i) a lack of soil and nutrient data more closely related to N availability (the primary limiting nutrient in these boreal forests), and (ii) inevitable uncertainty related to the response variable, i.e. "climate-normalized" aboveground productivity. The first point implies that variables such as C:N may be insufficient. This requires further testing with datasets that contain multiple indicators of N availability (not available for our dataset). We will briefly discuss the potential of a few other N availability indicators in the revised manuscript. The second point includes uncertainty in the original estimates of productivity, and uncertainty related to its normalization for climate: by taking residuals of the productivity vs climate regression model, we for instance unintentionally not only remove the direct effect of climate on productivity, but also its indirect effect on productivity through nutrient availability. In contrast, the approach taking actual/attainable productivity as a response variable does not suffer from this issue, but the estimates of attainable productivity are uncertain. As a consequence, low $R^2$ values are not just a result of variables failing to explain variation in the real nutrient availability, but also because the normalization procedure has shortcomings that can only be overcome by using datasets where climate does not vary but nutrient availability does (e.g. local gradients).

Finally, although the relationships between normalized productivity and soil variables and nutrients have low $R^2$ values, the significant relationships still point at the role of these variables in shaping nutrient availability. Especially the observations that for example SOC and C:N appear as significant factors, across both normalized productivity approaches and across soil moisture classes (see the reply to Ref#1's comments), confirm the robustness of our statements. Hence, we do not agree that the relationships are "non-existent", but we do agree that certain claims on for instance the importance of the variables should be described with more caution, and that the use of the upgraded metric remains to be evaluated, even within the boundaries of Sweden.

We agree with Referee #2's main remarks on generalizability of the current metric and on the fact that the difficulties in its development should be discussed. We will thus process the answers given in this reply in the manuscript, and make further clarifications and corrections based on the Specific and Technical Comments.

Please also note the supplement to this comment:
https://www.biogeosciences-discuss.net/bg-2017-372/bg-2017-372-AC2-supplement.pdf

---

## Author Response (AR1)

Dear editor,

We hereby submit the revised version of our manuscript. We are grateful to you and both referees for the insightful comments that helped us to significantly improve our manuscript.

Below, we provide point-by-point responses to all General, Specific and Technical Comments. A marked-up version of the manuscript is available at
https://www.dropbox.com/s/kf5qqqkxrau6ofq/KevinVanSundert_etal_Biogeosciences_MarkedUp.7z?dl=0.

Sincerely,

Kevin Van Sundert, on behalf of all co-authors.

**OVERVIEW OF MAIN CHANGES**

Based on several excellent suggestions of the referees and the editor, the revised manuscript now includes new analyses in which the influence of soil type, soil moisture and nitrogen deposition is thoroughly explored. We added a section 'Identifying potentially confounding factors', in which we investigate and discuss (confounding) factors that should be taken into account when evaluating nutrient-productivity relationships. Referee 1 and the editor for example advised to stratify by soil moisture and soil type. In the new section, we first present how soil properties and normalized productivity are related to soil moisture and soil type (Sections 2.3.1, 3.1 and 4.1). In the next sections we then present and interpret the stratified analyses in view of our research questions (e.g. Lines 306-308, 337-340, 487-488, 524-526). Referee 1 and the editor also emphasized that the role of N deposition should be discussed. We therefore investigated how N deposition correlated with soil variables and (normalized) productivity (e.g. Lines 189-198, 280-289, 391-402). This revealed a strong positive correlation between N deposition and productivity. However, this correlation disappeared when productivity was normalized for climate following method 1 (i.e. "residual productivity" - Fig. 3a), while it remained for the normalization according to method 2 (i.e. actual/attainable productivity), since actual/attainable productivity increases from north to south in Sweden, together with N deposition. In other words, confounding N deposition effects on nutrient-productivity relationships were neutralized for method 1, whereas for method 2, confounding N deposition should be kept in mind when evaluating nutrient-productivity relationships.

In the previous version of our manuscript and in our replies, we had mentioned the future use of data from natural gradients in nutrient availability. The editor wondered why we did not already evaluate our metrics[1] against data from gradients within the Swedish data. We thank the editor for this excellent idea, and have now evaluated our upgraded metrics against data from five nutrient availability/productivity gradients in Sweden. These gradients were selected to cover a range in nutrient availability and productivity within the same/very similar climate. The results from these analyses indicated that tree productivity was significantly related to our metrics for all gradients (Tables 4 and 5). Moreover, the $R^2$s of these relationships were generally higher than was the case when considering the complete database ($R^2$s were up to 38
* * *
[1]We now present two upgraded nutrient metrics instead of one, following referee 2's suggestion (COMMENT 2.16) to develop a second metric using productivity normalized according to method 2 (actual/attainable productivity - Fig. 3b), apart from the other metric (using productivity normalized following method 1, i.e. residual productivity - Fig. 3a) already presented in the previous version of the manuscript.

%). This suggests that the (necessary) procedure to normalize productivity for factors like climate in the previous analyses reduced the predictive power of the metrics. This last aspect also relates to referee 2's comment on the low $R^2$'s, which we have now addressed in a section on sources of uncertainty and future challenges in the discussion (Section 4.5).

**POINT-BY-POINT REPLIES TO THE REFEREE AND EDITOR COMMENTS**

GENERAL COMMENTS RAISED BY REFEREE 1

COMMENT 1.1: *"(…) Owing to the strong influence of site wetness on SOC, it is advisable to stratify and analyze soil fertility x productivity relations by wetness class, not just along N-S gradient."*

We followed the referee's advice to stratify the main analyses by soil moisture, and we have also stratified by soil type (separately), as suggested by the editor. In the revised manuscript, we present the stratified regression analyses (on soil variables + N deposition) and evaluations of the upgraded metrics. The regression outputs (Tables S3-S6 and S12-S15) generally confirmed the robustness of our results. We briefly discuss outcomes of the stratified analyses in the manuscript (e.g. Lines 306-308, 337-340, 487-488, 524-526).

COMMENT 1.2: *"N inputs are 4-5 times higher in Southern than Northern Sweden (Binkley and Hogberg 2016) and thus present a further factor that should be considered with evaluating nutrient and forest productivity across these sites (de Vries et al. 2014)."*

We tested the influence of N deposition on productivity, and on climate-normalized productivity. As explained on Lines 280-289 in the manuscript, N deposition indeed showed a strong positive relationship with productivity, because productivity increases in the north-south direction, as do N deposition, temperature and incident light. Whether N deposition correlated with normalized productivity depended on the normalization procedure (Fig. 3): for method 1 (which normalized productivity for species and climate [+ indirectly N deposition!]), there were mostly no positive associations with N deposition, while for method 2 (actual/attainable productivity of spruce), there was a strong positive relationship. This positive relationship followed logically from the increase of the ratio actual/attainable productivity from north to south Sweden. These issues, and their consequences, are discussed in the revised manuscript on Lines 391-402:

"Many studies have shown the strong influence of N deposition on forest productivity (e.g. Binkley and Hogberg, 2016; From et al., 2016). As expected, N deposition correlated to some extent with some of the soil variables considered in the present study, such as the total soil N stock and concentration (Fig. 4b). Furthermore, N deposition was strongly positively related to productivity. However, this effect of N deposition on productivity cannot be separated from the influence of climate and light, as all these factors increase together in the north-south direction. Nevertheless, we argue that for the goals of this study, i.e. investigating soil nutrient-productivity relationships across Sweden and developing a nutrient metric, the spatially varying N deposition is not problematic, since the normalization for climate and species according to method 1 (Fig. 3a) at the same time also removed the influence of the confounding N deposition on productivity. Accordingly, "Residual productivity" was generally not correlated with N deposition (Table S3). The response variable derived from method 2 (i.e. actual/attainable productivities for spruce - Fig. 3b) in contrast, correlated strongly with N deposition (Table S4), because both actual/attainable productivity and N deposition increased from north to south. Consequently, relationships between actual/attainable productivity and soil data for this method were unavoidably confounded by N deposition."

COMMENT 1.3: *"(…) Based on the conditions under which IIASA was developed, how well might it be expected to perform on forest soils and with tree growth? Without clear description of the IIASA (currently some found in Intro, but lacking from Abstract), the paper presents a 'house of cards' based on a relatively unknown, and possibly weak foundation. Finally, please justify why the simpler approach of originating with 'no' tool and creating one from the ground up with existing data, as outlined in Question 1, is not both adequate and preferable."*

We agree with the referee that this choice required further clarification. In the revised manuscript, we therefore explain the basic principle of the IIASA-metric in more detail in the abstract (Lines 17-21), methods (Lines 130-146) and discussion (Lines 446-451):

"For the metric, we started from a (yet unvalidated) metric for constraints on nutrient availability that was previously developed by IIASA (Laxenburg, Austria). This IIASA-metric - initially developed for evaluating potential productivity of arable land - consists of soil properties that are indicative of nutrient availability and is based on theoretical considerations that are also generally valid for non-agricultural ecosystems."

"The IIASA-metric of constraints on nutrient availability, originally meant for use on arable land, incorporates four crop specific scores (estimated for SOC, texture, TEB and pHw) that can be assigned to a soil (IIASA and FAO, 2012). These scores, which can be found in look-up tables (http://webarchive.iiasa.ac.at/Research/LUC/GAEZv3.0/soil_evaluation.html), were derived from crop growth data on different agricultural soils. Given that we were analyzing boreal forests and not crops, we averaged the scores of the different crop species for each of the four soil properties. As such, we thus removed crop specific requirements, but generally known relationships between the soil variables and plant performance (not only valid for agro-ecosystems), such as an optimum for pH, remained. (…)"

"(…) the structures of its formulas (Eqs.(6-9)) reflect general mechanisms that link soil properties to nutrient availability, which are also valid for non-agricultural ecosystems. Soil pH for example shows a typical optimum effect on nutrient availability, while SOC and TEB have a direct positive non-linear influence (IIASA and FAO, 2012). The final weighing of the four partial scores (Eq. (10)) finds its rationale in the idea that if a certain soil property is particularly suboptimal, it will be the most important nutrient-related determinant of productivity, with less influence of the other soil properties that are within the optimal range. (…)"

and added paragraphs in the discussion on why (i) we started with this particular metric (formulas were based on theoretical considerations, it indirectly includes interactions, is upgradeable, and it is the only generic nutrient metric - Lines 444-455):

"Although the IIASA-metric of constraints of nutrient availability was originally designed for evaluating constraints on nutrient availability of arable lands, we opted to start with this metric for a couple of reasons. Apart from the fact that to our knowledge, it represents only attempt so far to develop a generic nutrient metric, the structures of its formulas (Eqs.(6-9)) reflect general mechanisms that link soil properties to nutrient availability, which are also valid for non-agricultural ecosystems. Soil pH for example shows a typical optimum effect on nutrient availability, while SOC and TEB have a direct positive non-linear influence (IIASA and FAO, 2012). The final weighing of the four partial scores (Eq. (10)) finds its rationale in the idea that if a certain soil property is particularly suboptimal, it will be the most important nutrient-related determinant of productivity, with less influence of the other soil properties that are within the optimal range. This way of weighing can be considered as a type of interaction, but one that cannot be implemented in a simple linear regression model. Hence, our main reason for adopting the IIASA-metric as a starting point is that, in spite of its simplicity, it is based on theoretical considerations. Moreover, adopting this structure allows for updating, in contrast to multiple regression equations (see below)."

and why (ii) multiple regression equations are not practical (i.e. not upgradeable with data from outside Sweden – Lines 494-505):

"Variation in normalized productivity explained by the upgraded metrics ($R^2 = 0.03–0.21$ and $R^2 = 0.06–0.18$) was similar to the variation explained by multiple regression equations ($R^2 = 0.18–0.22$) that contained the same (and

more) soil variables than the metrics. The metrics can, however, easily be updated, while equations derived from multiple regressions would be of less use for building a nutrient availability metric because such an entirely empirical approach does not allow for later updates of parameters or model structures based on data from other ecosystems. In order to further enhance performance of the metrics, and to test to what extent they can describe variation in nutrient availability outside of Swedish conifer forests, additional datasets with productivity and soil information are needed. (…)"

**SPECIFIC AND TECHNICAL COMMENTS RAISED BY REFEREE 1**

COMMENT 1.4: *"Line 8 - Nutrients are one of the factors that influence C cycling. As presented this suggests they are the only or the primary factor. (...)"*

We rephrased the sentence (Lines 8-9).

COMMENT 1.5: "*Line 10 - 'ideally' The potential value of simplifying ecosystem complexity into interpretable and useful metrics is critically important. However, given the complexity of soils and forest ecosystems, it is worth questioning whether a single fertility metric is an attainable goal. Rather than assuming such an index is 'ideal' this paper might sim-ply evaluate relationships between soil fertility properties and forest growth are robust and broadly useful.*"

We were glad to see that the referee appreciates our effort to develop a common nutrient availability metric. Whether or not a global, generic nutrient availability metric can be developed remains an open question. In any case, even if this turns out too complicated, a valuable alternative could be to develop a metric for forests only for example. In response to this comment, we have specified some challenges in the development (e.g. differential nutrient limitation and low $R^2s$ resulting from the normalization of productivity) in Section 4.5 in the revised manuscript.

Moreover, although the development of a nutrient metric is a key part of our study, our manuscript also provides important information about factors related to soil fertility in general. In the paper, we demonstrated for example that the soil variables most critical for predicting productivity across Swedish forests were soil C:N ratio and SOC.

COMMENT 1.6: *"Line 10 - 'Such a metric does not exist' Is the lack of such a metric a major obstacle to understanding of forest production? An objective of this paper should be to quantify how such a metric improves our ability to predict forest growth."*

The need for a common metric of nutrient availability goes far beyond predicting forest productivity. A generic nutrient availability metric would allow to make comparisons in the nutrient status among monitoring sites and field experiments. This would for instance greatly increase the statistical power of synthesis studies that aim to elucidate the role of nutrients in ecosystem functioning (e.g. Lines 32-50):

"Nutrients determine structure and functioning at all levels of biological organization. The availability of mineral elements influences for example plant growth (von Liebig, 1840), patterns of biodiversity (Fraser et al., 2015) and ecosystem processes (e.g. Janssens et al., 2010; Vicca et al., 2012; Fernández-Martínez et al., 2014). Moreover, nutrient availability can modify ecosystem responses to global atmospheric and climatic changes, such as nitrogen (N) deposition (From et al., 2016), increasing $CO_2$ levels (Norby et al., 2010; Terrer et al., 2016), warming (Dieleman et al., 2012) and drought (Friedrich et al., 2012). Given the crucial role of nutrients in terrestrial carbon cycling and in shaping the magnitude and direction of its feedbacks to climate change, nutrient availability should be taken into account in global analyses and in Earth system models (Goll et al., 2012; Thomas et al., 2015; Wieder

et al., 2015). This is, however, not yet common practice because we often lack the soil data and metrics needed to accurately account for nutrient availability.

Comparing nutrient availability among terrestrial ecosystems is thus difficult for two reasons: comprehensive and harmonized data on soil properties and nutrients are not usually available from experimental and observational sites, and no standardized quantitative metric exists to compare the nutrient statuses of terrestrial ecosystems at the global scale, or even at a national scale (e.g. for Sweden, which is considered in the present paper). In the absence of a standardized nutrient availability metric, studies comparing nutrient availability across sites have previously described soil fertility related approximations such as the height of 100 year old trees (which, however, also depends on other factors such as soil depth and hydrology - Hägglund and Lundmark, 1977) or have manually classified sites as low, medium, and high nutrient availability based on existing site information (Vicca et al., 2012; Fernández-Martínez et al., 2014). The absence of a more nuanced expression impedes elucidating the role of nutrient availability in ecosystem processes and functioning (Cleveland et al., 2011) and how these respond to global change, and precludes investigating non-linear effects of nutrient availability."

COMMENT 1.7: *"Line 12 - Insert 'of' between 'combination' and 'soil' Change 'plant' to 'tree' Insert 'forest' before 'nutrient'"*

Done. (Lines 11-13)

COMMENT 1.8: "*Line 13-14 - Developing this analysis using an unproven metric is counterin-tuitive. Justify why you did not start out with 'no' tool and build one from the ground up with existing data. Also, explain how the IIASA approach tool developed elsewhere for crops has application for Swedish forests. Clarify how advances from your analysis are independent of potential weakness or limitations of the IIASA approach.*"

In the revised manuscript, we now better explain why we opted to start with the IIASA-metric: the metric's scores for soil properties are based on generally (qualitatively) known relationships between soil properties and soil fertility, which are to some extent valid across ecosystem types (Lines 17-21, 130-142, 444-455). We now also clarified that we anticipated in advance that N availability (~ soil C:N ratio) should be better represented in the metric (Lines 459-465). We further stress that the metrics are upgradeable - in contrast to multiple regression equations, see Lines 494-505 - and can be tested against data from outside Sweden. Also see COMMENT 1.3.

COMMENT 1.9: *"Line 19 - 'not well implemented' Define what that means, especially in light of whether or IIASA provides a useful platform to work with."*

If soil variables are properly implemented in a nutrient metric, they should not be correlated with residuals obtained by regressions of (normalized) productivity vs that metric. We clarified this in the manuscript (Lines 22-23, 161-163, 213-216, 312-318, 340-346, 349-351, 466-469) and in the captions of Tables S9 and S16-19.

COMMENT 1.10: *"Line 37/38 - Site index is typically an estimate of site productivity, rather than soil fertility per se. The terms are certainly related, but key differences might include factors such as soil depth and hydrology."*

We agree and thank the reviewer for notifying. We rephrased the sentence (Lines 45-47).

COMMENT 1.11: *"Line 47 - 'are more indicative' it's not clear of what. Please revise, clarify."*

Soil properties such as soil texture, SOM and pH are more indicative of the general nutrient status, because nutrient availability is determined by the interplay between different soil

nutrients and influenced by soil properties like pH. We rephrased this in the manuscript (Lines 54-57).

COMMENT 1.12: *"Line 100 - What was the size of the forest plots? Are these all plantations or are some natural stands? How are management interventions such as fertilization and thinning addressed and controlled for? What is the average and range of stand age and stand density or basal area?"*

We added this information to Table 1 in the "Methods" section. As the inventory represents a random sample of all Swedish forests, the majority is managed. There are typically 1-2 thinnings per rotation, while fertilization was only carried out for a minor proportion of the forests. The thinnings did in all likelihood not bias our results, because we used rotation-averaged productivities, based on height development curves of dominant trees, as our response variable.

COMMENT 1.13: *"Line 122 - Before suggesting that metric scores can validly 'be assigned to any soil' it would be useful to know more about the data that went into development of the IIASA and how well those soil types and crop conditions match the forest soils of Sweden."*

We fully agree with the referee. In the revised manuscript, we now explain that the IIASA-metric's formulas represent general relationships between productivity and a few soil properties (Lines 17-21, 130-142, 444-455). These equations were calibrated based on data from different agricultural soils and crop growth. Although forest soils differ in various aspects (e.g. Lines 461-465), general relationships between SOC, soil texture, TEB, and pH on the one hand and productivity on the other hand should in principle remain qualitatively the same, unless other confounding factors (such as soil moisture) mask these relationships.

COMMENT 1.14: *"Line 123 - This suggests that the IIASA metric is not sensitive to crop specific requirements."*

We rephrased this (Lines 133-137). The IIASA-metric is crop-specific, but general, logical patterns between soil properties and the partial scores remain after averaging scores of different species.

COMMENT 1.15: *"Line 284-285 - Use soil moisture classes to stratify fertility by productivity. There appear to be abrupt differences between upland and 'wetland' forests that are at least as significant as the N-S gradients."*

We thank the referee for this insightful comment. We have incorporated the suggested analyses in the manuscript, and in addition stratified by soil type (as suggested by the editor).

We added a section 'Identifying potentially confounding factors', in which we investigate and discuss (confounding) factors that should be taken into account when evaluating nutrient-productivity relationships. In the new section, we first present how soil properties and normalized productivity are related to soil moisture and soil type (Sections 2.3.1, 3.1 and 4.1). In the next sections we then present and interpret the stratified analyses in view of our research questions (e.g. Lines 306-308, 337-340, 487-488, 524-526). In general, the stratified analyses confirmed the robustness of our results.

COMMENT 1.16: *"Line 295 - The high SOC of wet soils is less a limitation than the anaerobic rooting environment."*

We agree with the referee and have rephrased the sentence accordingly (Lines 421-424).

COMMENT 1.17: *"Line 296 - Consider optima for soil saturation, soil oxygen, or soil moisture like the one currently used for pH. This is independent of SOC."*

We agree with the referee that if we were to develop a productivity index, including soil moisture or oxygen should be considered. Our goal, however, is to develop a metric on nutrient availability. Distinguishing nutrient availability (metrics) from soil moisture (metrics) is essential if the eventual metric is to be used to disentangle drivers of ecosystem functioning and to separate and quantify the influence of nutrient availability and soil moisture. Although soil saturation/moisture/oxygen influence productivity, we therefore did not include these factors in the upgraded metrics.

COMMENT 1.18: *"Line 301 - It is ironic that the only nutrient examined is of limited value. It should be stressed that total N is a much larger pool than the N that's readily available to plants. The complications of inorganic N inputs across the N deposition gradient, and inorganic N losses (leaching and denitrification) across wetness/saturation gradient are likely considerable."*

We fully agree with the referee and have rephrased the sentence to make clear that available N is just a small fraction of total N (Lines 429-431).

COMMENT 1.19: *"Line 304 - This relation appears to stem on the large difference in clay in Wet spruce sites (Fig S3e). Again, stratifying wetland from up-land forests might isolate and better resolve these patterns."*

It is not entirely clear to us what the referee refers to here: there was a significant positive relationship between the mineral soil clay fraction and normalized productivity (albeit with very low $R^2$s). The referee correctly observed that mineral soils at wet spruce sites had the highest clay contents, but based on this and the fact that productivity was suppressed at wet sites, we would rather expect a negative association between mineral soil clay and normalized productivity.

Following the referee's advice, we stratified analyses by soil moisture. The positive relationship between mineral soil clay and normalized productivity remained, irrespective of the moisture class, or became non-significant (Table S5).

COMMENT 1.20: *"Line 309 - Insert and stress "in the absence of direct soil nutrient data""*

We followed the referees suggestion and have rephrased the sentence (Line 438).

COMMENT 1.21: *"Line 315/316 - General differences between arable and Swedish forest soils could be used to define expectations regarding the utility of the IIASA method and probable modifications. Saturated soils, low-clay soils, low N soils might all define some of those differences."*

We thank the referee for this suggestion and added these differences between arable and conifer forest soils to the discussion about the performance of the IIASA-metric (Lines 462-465).

COMMENT 1.22: *"Line 324 - High water table, low soil oxygen, soil saturation, wetland soils."*

We clarified this link now at various places in the manuscript (Lines 184-185, 372-374, 417-420, 470-473, 543-544).

COMMENT 1.23: *"Table 1 - Units are included for everything but TN, and moisture. How was moisture determined and how well does it estimate relevant seasonal moisture dynamics?"*

We thank the referee for noticing this and added the unit for TN [%]. Soil moisture and soil type do not have a unit, but are represented in different classes. This is now indicated in Table 1. We provided relevant information about the plots, sampling, lab analyses and classification in the caption of the table, and in the notes underneath it. The classification of soil wetness is representative of the average moisture conditions during the growing season.

COMMENT 1.24: *"Table 2 - Sample depths are listed for everything but sand and clay."*

Soil texture data were available as texture classes for the upper mineral soil - not specifying soil depth. We now explain this in the caption of Tables 2 and 3.

COMMENT 1.25: *"Figure 1 - Specify what 'Species-averaged' means or reword so that the caption and figures are interpretable independent of the text. The range of SOC data (1-3%) is nowhere near the ranges presented in Figure 5 and 6 (0.1 to 100%), where the data is presented on a log-scale axis. Explain or resolve the discrepancy."*

This remark relates to the different SOC range for the Swedish database versus the range used in the IIASA-metric (the latter being much smaller). To clarify this issue, we now mention the different SOC axis under Figs. 7 and 8, and emphasize in these captions that SOC varied widely in the Swedish database.

We also rephrased the caption in Fig. 1 to make clear that the scores were averaged over all crop species.

GENERAL COMMENTS RAISED BY REFEREE 2

COMMENT 2.1: *"In the introduction the authors write that they aim at developing a globally valid metric while using data from Sweden only. It should be clarified that the upgraded metric is only valid for Sweden. Possibly, a global metric is not achievable at all, since nutrient availability is not limited by the same factors in different ecosystems worldwide. Hence, a metric for Sweden might be usable for other boreal, but not for tropical ecosystems. These considerations should be discussed."*

We agree with the referee that the metrics presented in the current manuscript are only valid for Sweden and have emphasized this in the revised manuscript (Lines 14, 91-93, 498-500, 530, 548-553). Future evaluations of the current metric against data from ecosystems (forests) outside Sweden are needed to verify its applicability, and to further update its formulation. We also agree that serious challenges are ahead, such as differential nutrient limitation and especially extrapolation to other ecosystem types. In the revised manuscript, we added a new Section 4.5 discussing the main challenges and provide an outlook for how to deal with issues such as differential nutrient limitation:

"Even though normalized productivity was significantly related to soil properties, and to our upgraded metrics, much of the variation in normalized productivity remains unexplained. The considerable unexplained variation may have multiple reasons. Apart from a possible lack of soil and nutrient data more closely related to N availability than the ones available in our database, another possible factor reducing R²s could be the quality of

the data in the database. This could for instance be due to an insufficient number of replicates sampled per data point (n = 3 for the soils), although this is probably of limited importance because of the large number of data points in the database itself. A more important source of uncertainty is probably the inevitable uncertainty related to the response variable, i.e. "climate-normalized" aboveground productivity. This not only includes uncertainty in the original productivity estimates (for which for example differences in management or disturbances likely increased variability), but also uncertainty related to the normalization for climate. By taking residuals of the productivity vs climate regression model (method 1), we for instance unintentionally not only removed the direct effect of climate on productivity, but also its indirect effect through nutrient availability. Normalized productivity based on this method thus mainly represents productivity as influenced by regional variation in nutrient availability. The approach taking actual/attainable productivity as a response variable (method 2) does not suffer from this issue, but there the estimates of attainable productivity come with a high uncertainty. As a consequence, the low $R^2$ values are party due to shortcomings of the normalization procedure that can only be overcome by using datasets where climate does not vary but nutrient availability does. Such datasets are provided by local gradients, such as the five local nutrient availability gradients that we randomly selected from our database for additional evaluation of our upgraded metrics.

Despite the limited $R^2$, similar significant results for different methods (1 and 2) and subsets of the database (regions, soil moisture classes and soil types) indicated that the findings about the soil properties and nutrients are generally robust. The upgraded metrics explained up to 21 % of the variation in normalized productivity. It is unclear to what degree the influence of nutrient availability is covered by this percentage. Future studies, where additional soil data can be included, will need to verify this. In any case, the significant relationships with normalized productivity, the better implementation of the soil variables and the capability of the metrics to explain up to 38 % of the variation in productivity across different gradients imply a significant improvement compared to the original IIASA-metric for this database.

A key challenge in the further development of a metric describing spatial variation in nutrient availability both within and outside the boreal biome is differential nutrient limitation. Eventually, we want to be able to compare for example N-limited and P-limited systems. The original structure of the IIASA-metric, which was kept in our upgraded metrics, facilitates this by allowing the inclusion of multiple soil variables such as C:N (mainly relating to N availability), pH (among others a critical factor controlling P availability) and exchangeable bases in one single metric. In fact, the IIASA-metric is particularly useful in this regard, as it gives more weight to the soil factor with the lowest score. This corresponds to reality and enables accounting for the type of nutrient limitation. For instance, if C:N is high, indicating N limitation, the metric score will be substantially reduced by this high C:N, while at low C:N other limiting factors can dominate the metric score."

COMMENT 2.2: *"The performance of the metric is bad. Instead of discussing the – non-existing – relationships, the authors should rather discuss the possible reasons for the failure of the metric. One possible reason is data quality. The authors should describe the soil sampling design and the methods used for chemical analyses. Inventory data might not be suitable to find relationships between parameters even though they exist, because (soil) variability would require a large number of replications, which is often not affordable in inventories."*

We thank the referee for his/her critical assessment, which has helped us in interpreting the current metric's performance and was very useful for developing future perspectives.

We realize that the predictive power of the upgraded metric was rather low. Data quality may to some extent contribute to the seemingly inadequate performance of the metric. However, we identified other reasons for the low $R^2$s that are probably more important than data quality. First, the necessary normalization procedure for climate increased the uncertainty on normalized productivity and lowered coefficients of variation. This became clear in our new analyses using randomly selected local nutrient availability gradients. Within such a gradient, normalization for climate was not necessary and the predictive power of the metric increased up to 38 %.

On the other hand, the real proportion of variation explained by nutrient availability is simply unknown and hence it is impossible to set a target for the $R^2$. Some of the unexplained variation is, for example, likely due to differences in management and disturbances. Given that our new analyses on the local gradients reveal that a substantial proportion of variation is explained by our upgraded nutrient metric, we consider these metrics a useful starting point for follow-up studies evaluating (and further upgrading) the metric.

In the revised manuscript, we added a new section (Section 4.5) about "sources of uncertainty and future challenges". We mention there data quality as a possible reason for the low $R^2$s, and explain that especially the normalization procedure also contributed substantially to the uncertainty (given the substantially higher $R^2$s for the local gradients where normalization was not needed).

As requested, we have also added more information about the forest plots, replicates etc. in the "Methods" section and in the caption and notes of Table 1. Finally, we refer to other papers that offer more detailed information about soil sampling and laboratory analyses in this table.

SPECIFIC COMMENTS RAISED BY REFEREE 2

COMMENT 2.3: *"17 - The coefficients of determination are that poor that you should not write "Normalized productivity increased with decreasing soil C:N ratio, while SOC exhibited an empirical optimum.""*

We rephrased this sentence (Lines 16-17), and added a section to the discussion on sources of uncertainty, including reasons why $R^2$s were low (Section 4.5).

COMMENT 2.4: *"21 - The coefficient of determination of the upgraded metric is still poor and should not be called "a significant fraction"."*

We rephrased this sentence to make it more careful (Lines 25-26).

COMMENT 2.5: *"34 and 36 - "among terrestrial ecosystems" and "global scale" is misleading. You should clarify that in the present paper only Sweden is considered."*

Even though we aim to further upgrade the metrics from this paper with data from ecosystems outside the boreal biome, we agree that in the previous version of our manuscript, it was not always sufficiently clear that only Swedish data were considered. In the revised manuscript, we therefore stress this at multiple places in the text (Lines 14, 91-93, 498-500, 530, 548-553), and rephrased the sentence indicated by the referee (Lines 43-44):

"Comparing nutrient availability among terrestrial ecosystems is thus difficult for two reasons: comprehensive and harmonized data on soil properties and nutrients are not usually available from experimental and observational sites, and no standardized quantitative metric exists to compare the nutrient statuses of terrestrial ecosystems at the global scale, or even at a national scale (e.g. for Sweden, which is considered in the present paper)."

COMMENT 2.6: *"47 - "are more indicative" to what and for what?"*

Soil properties such as soil texture, SOM and pH are more indicative of the general nutrient status, which is determined by the interplay between different soil nutrients and influenced by soil properties like pH. We rephrased this in the manuscript (Lines 54-57).

COMMENT 2.7: *"48 - What is meant with "the size of the soil solution"?*

We meant the "size of the soil solution pool [of a particular nutrient], exchange sites and unavailable soil pools (…)", and correspondingly clarified this in the manuscript (Lines 56-57).

COMMENT 2.8: *"57 to 60 Yes! Perhaps the fact that a global metric might not be achievable at all, since nutrient availability is not limited by the same factors in different ecosystems worldwide, should be discussed here, too."*

As answered to COMMENT 2.1, we agree with the referee that differential nutrient limitation poses a challenge in the development of a global nutrient metric. The retained structure of the original IIASA formula seems, however, particularly useful for tackling this issue. This metric allows including multiple soil properties and nutrients and gives more weight to the soil factor that is most limiting (i.e. the one with the lowest score). We incorporated our answer in the discussion of the revised manuscript (Lines 531-538):

"A key challenge in the further development of a metric describing spatial variation in nutrient availability both within and outside the boreal biome is differential nutrient limitation. Eventually, we want to be able to compare for example N-limited and P-limited systems. The original structure of the IIASA-metric, which was kept in our upgraded metrics, facilitates this by allowing the inclusion of multiple soil variables such as C:N (mainly relating to N availability), pH (among others a critical factor controlling P availability) and exchangeable bases in one single metric. In fact, the IIASA-metric is particularly useful in this regard, as it gives more weight to the soil factor with the lowest score. This corresponds to reality and enables accounting for the type of nutrient limitation. For instance, if C:N is high, indicating N limitation, the metric score will be substantially reduced by this high C:N, while at low C:N other limiting factors can dominate the metric score."

COMMENT 2.9: *"78 to 84 If the goal is a "global metric", data from the Swedish forest inventory service do not represent "a substantial variation in nutrient availability". Which were the additional variables? P?"*

We fully agree with the referee that the variation in nutrient availability across Sweden is much smaller than global variation in nutrient availability. The reason for using the Swedish forest inventor data was primarily the fact that this dataset met the main data needs, i.e., a database that combines the necessary information on soil properties and nutrients with data on plant productivity, while also covering a substantial variation in nutrient availability and containing a large number of sites to increase statistical power. We have clarified this in the revised manuscript (Lines 91-93) and also emphasize now in several places that the current metric applies to Sweden only and further analyses are needed to develop a globally applicable metric (Lines 14, 91-93, 498-500, 530, 548-553).

The additional soil variables referred to were soil total N concentration/stock and especially soil C:N ratio. We rephrased this sentence in the manuscript (Lines 89-91).

COMMENT 2.10: *"89 - Here you restrict your results on Swedish forests – you should already earlier mention that your goal is a metric for Sweden – not a global metric."*

Done (Lines 14, 43-44, 91-93).

COMMENT 2.11: *"94 to 134 - List all the parameters (soil, tree/productivity, climate/meteorology) and explain how they were measured."*

We listed the parameters in Table 1 of the revised manuscript, and mentioned the most relevant sampling, analysis, … details in the caption and underneath the table. We provided references for further details on soil sampling and analyses.

COMMENT 2.12: *"121 to 130 - Explain in more detail how the scores were derived and what you can find in the look-up tables."*

We rephrased the paragraph (Lines 130-137), and the caption of Fig. 1. We also added a hyperlink to the look-up tables.

COMMENT 2.13: *"137 to 138 - You should state that you call the two alternative ways to calculate normalized productivity "method 1" and "method 2" in the following."*

Done (Lines 148-159).

COMMENT 2.14: *"138 - Name the advantages and drawbacks."*

We now explain the main disadvantages of each method in this paragraph (Lines 155-159).

COMMENT 2.15: *"179 - Is "method 1" referring to the method used to calculated the normalized productivity?"*

Yes, it is. This should be clearer now that we rewrote the paragraph about the normalization methods (Lines 148-159, 168-176). We also referred to Fig. 3 here.

COMMENT 2.16: *"193 to 200 - You used half of the dataset from southern Sweden to upgrade the metric and tested the upgraded metric with the rest of the dataset including middle and northern Sweden, right? Did you also upgrade the metric using method 2 (half of the dataset for calibration, half for validation)"*

The referee's interpretation is correct. Thanks to this comment, we realized that this paragraph, where we explained the metric upgrades, was not sufficiently clear. We therefore rewrote the paragraph (Lines 218-232). We now also developed a second upgraded metric, based on method 2.

COMMENT 2.17: *"216 to 217 - Why not using both SOC and TN in separate models and choosing the model that better fits the data?"*

Using SOC always resulted in higher $R^2$s than using TN, irrespective of the normalization method on productivity ($R^2$ was 0.145 [method 1] or 0.048 [method 2] for SOC, while it was 0.116 [method 1] or 0.046 [method 2] for TN). We did not explicitly mention this in the manuscript, since we preferred to continue with SOC anyway because (i) it is a component of the IIASA-metric, and (ii) SOC/M better represents the organic matter content, which not only acts as a nutrient reservoir, but also provides cation and anion exchange sites. We clarified our *a priori* choice in the paragraph (Lines 261-264).

COMMENT 2.18: *"225 - "… related to normalized productivity (Table 2), however, the coefficient of determination was small (0,002 to 0,146)."*

Done (Line 293), and we also rewrote our assertions more carefully. See also the remarks on $R^2$s (COMMENTS 2.2, 2.3, 2.4, …).

COMMENT 2.19: *"227 - I cannot see from Fig. 5 that "the effect became more pronounced towards the south".*

With this sentence we mean that both the slope and $R^2$ increase from north to south (also shown in Table 2). We clarified this in the manuscript (Lines 294-295). See also COMMENT 2.31: we reassessed whether the $R^2$ in the north was lowest, and confirm that the information provided in the figure, table and text is correct.

COMMENT 2.20: *"232 to 234 - According to Table 3 also other variables than SOC, pH and C:N were included in the multiple regression models. Why?"*

Even though the relationships between the other soil variables and normalized productivity were less clear than for SOC, pH and C:N, they were also included by the model selection procedure. In other words, the minimum cross-validation mean squares were reached for models with additional soil variables, but the role of these variables in the simple and multiple regression formulas was less consistent.

COMMENT 2.21: *"239 to 240 - On the one hand you write that SOC and C:N "consistently describe a distinct, clear effect on normalized productivity" on the other hand you write that R2 is "at least a few percent" – this is contradictive. Both the figures and the R2 show that there is no distinct, clear effect on normalized productivity."*

We resolved this discrepancy in the paragraph (Lines 303-305):

"In summary, SOC and the soil C:N ratio were the only soil factors that showed a similar trend according both methods with an $R^2$ of at least a few percent, and were thus included in the multiple regression models for both methods 1 and 2 (Table 3)."

COMMENT 2.22: *"225 to 241 - This part should be rewritten since the variance in productivity is hardly explained by the soil variables (see also discussion l. 272 to 313)."*

We rewrote sentences in this part and elsewhere more carefully now (Lines 291-305). See also the notes on low $R^2$s in our replies (COMMENTS 2.2, 2.3, 2.4, 2.18) and in the manuscript (Section 4.5).

COMMENT 2.23: *"251 to 252 - The relationship between residuals and variables is very weak."*

If residuals of the (normalized) productivity-metric associations are still related to soil variables already included in the metric, this could point to a suboptimal implementation of the variables. We clarified this better in the manuscript (Lines 22-23, 161-163, 213-216, 312-318, 340-346, 349-351, 466-469). We also refer to our answers on the comments on low $R^2$s (COMMENTS 2.2, 2.3, 2.4, 2.18, 2.22 and Section 4.5).

COMMENT 2.24: *"254 - To my opinion, the results do not show "that SOC, C:N and pH are important factors influencing nutrient availability in Sweden."*

We formulated this more carefully. The sentence now reads as follows:

"From the statistical analyses for *question 1*, we deduce that SOC, soil C:N and pH each play a role in influencing nutrient availability in Sweden."

COMMENT 2.25: *"264 to 265 - From Figure 8 no enhancement of the upgraded metric can be deduced. The performance is still bad."*

In contrast to the IIASA-metric (Fig. 9), both upgraded metrics are significantly positively related to normalized productivity across the database (Figs. 10 and 11). Moreover, our new analyses demonstrate that our metrics each explain (more) variation in productivity for five randomly selected nutrient availability gradients (Tables 4 and 5). The metrics are thus an improvement, at least for the Swedish boreal forests considered in the present study. They should not be seen as an endpoint, but rather as a first step in a development process.

COMMENT 2.26: *"272 to 313 - The variance in productivity is not explained well by the soil variables. Instead of discussing the (significant but very weak) relationships, you should rather discuss why you do not find relationships were you would expect them. What about the soil sampling design and the methods used for chemical analyses? You did not describe them in the material and methods section. Could the sampling design (number of replications) explain the bad model performance? How is the variability of soil variables in Sweden?"*

In response to the referee's concern, we discuss in the revised manuscript the potential sources of uncertainty (Section 4.5). We also provide (references to) more information about sampling methods in the methods section (Table 1). See also our more elaborate reply to the COMMENT 2.2.

COMMENT 2.27: *"323 - … but again with a very small coefficient of determination."*

See other replies to comments on low $R^2$s (COMMENT 2.2, …).

COMMENT 2.28: *"330 - "… the nutrient availability metric was intended to be improved by …"*

At least for the Swedish database, the upgraded metrics are an improvement compared to the IIASA-metric. See also other replies to comments on metric performance (COMMENT 2.25) and low $R^2$s (COMMENT 2.2, …).

COMMENT 2.29: *"335 - I cannot agree. The relationships were significant but only a very small part of the variance was explained by the variables used."*

We rephrased the sentence to make it more careful. In the revised manuscript, we state that the upgraded metrics explained some variation (especially for gradients - Lines 482-493):

"In contrast to the original metric developed by IIASA, the upgraded metrics described some variation across all approaches using the full database (Figs. 10 and 11). Variables were generally properly implemented, at least for upgraded metric 1 (Table 5), while for metric 2, significant associations emerged between residuals of normalized productivity and SOC and pH (Table S17). These associations had, however, opposite trends depending on whether normalization method 1 or 2 was considered, meaning that the methods did not agree on whether the metric score was either under- or overestimated for the soil variable considered. The stratified analyses confirm that the metrics are an improvement, at least for those soil moisture classes and soil types with sufficient data points (Tables S12-15). Moreover, each metric could describe spatial variation in productivity for five randomly selected nutrient availability gradients (Tables 4 and 5). The coefficients of determination were generally higher for these gradients than for the database analyses, likely because the gradients did not require a normalization for climate (the latter increased the uncertainty on the response variable, see section 4.5 on sources of uncertainty and future challenges). Lastly, the gradients generally confirmed the correct implementation of soil variables in upgraded metric 1 (Table S18), whereas for metric 2, scores for high SOC might be overestimated (Table S19)."

See also the other replies to comments on low $R^2$s (COMMENT 2.2, …).

COMMENT 2.30: *"336 to 339 - Why are especially stable N isotope signatures and ion exchange resin bags of interest? What about other methods?"*

N isotope signatures and ion exchange resins are just two examples. We clarify this better in the revised manuscript and also mention a few other options (Lines 502-505).

COMMENT 2.31: *"Table 2 - Are you sure that R² is the same for all three regions for the parameters SOC, N stock, sand, clay and pHKCl? This is quite unrealistic and the data shown in Figure 5 (SOC) lead to the assumption that R2 of the North is largest and that of the South smallest. In line 226 to 227 you write that the effect becomes more pronounced towards the south (C:N), however, in Figure 5 the relationship is worse for the south than for the other regions and a larger R2 – as written in Table 2 – seems to be quite unrealistic."*

In cases where $R^2_{tot}$ was written instead of $R^2$, we presented outputs of ANCOVA models of which the cross-validation mean squared error was minimal when excluding a soil variable[2] x region interaction. Consequently, results for all three regions were based on one model. When $R^2$ was written, there was a (significant/selected) soil variable x region interaction, and splitting up the analyses among regions yielded separate parameter estimates (here, we could have provided a joined $R^2_{tot}$ as well, while slope and intercept estimates would have remained the same, but splitting up provides more information on exactly these $R^2$s).

We reassessed whether the $R^2$ in the north was lowest, and confirm that the information provided in the figure, table and text is correct.

```
Call:
lm(formula = Resid[Region == "N"] ~ CNup[Region == "N"])

Residuals:
    Min      1Q  Median      3Q     Max
-2.03337 -0.51938  0.01981  0.53435  2.52545

Coefficients:
                   Estimate Std. Error t value Pr(>|t|)
(Intercept)        0.343074   0.104818   3.273 0.001131 **
CNup[Region == "N"] -0.013609  0.003858  -3.527 0.000455 ***
* * *
Signif. codes:  0 '***' 0.001 '**' 0.01 '*' 0.05 '.' 0.1 ' ' 1

Residual standard error: 0.7218 on 545 degrees of freedom
  (64 observations deleted due to missingness)
Multiple R-squared:  0.02232,   Adjusted R-squared:  0.02052
F-statistic: 12.44 on 1 and 545 DF,  p-value: 0.0004555
```

```
Call:
lm(formula = Resid[Region == "M"] ~ CNup[Region == "M"])

Residuals:
    Min      1Q  Median      3Q     Max
-3.2659 -0.7302  0.0153  0.7555  3.9562

Coefficients:
                   Estimate Std. Error t value Pr(>|t|)
(Intercept)        0.692602   0.151206   4.581 5.40e-06 ***
CNup[Region == "M"] -0.026695  0.005439  -4.908 1.12e-06 ***
* * *
Signif. codes:  0 '***' 0.001 '**' 0.01 '*' 0.05 '.' 0.1 ' ' 1

Residual standard error: 1.117 on 780 degrees of freedom
  (67 observations deleted due to missingness)
Multiple R-squared:  0.02996,   Adjusted R-squared:  0.02871
F-statistic: 24.09 on 1 and 780 DF,  p-value: 1.12e-06
```

```
Call:
lm(formula = Resid[Region == "S"] ~ CNup[Region == "S"])

Residuals:
    Min      1Q  Median      3Q     Max
-5.3262 -0.7532  0.1444  0.9772  6.3101

Coefficients:
                   Estimate Std. Error t value Pr(>|t|)
(Intercept)        2.04169    0.18359   11.12   <2e-16 ***
CNup[Region == "S"] -0.08163   0.00742  -11.00   <2e-16 ***
* * *
Signif. codes:  0 '***' 0.001 '**' 0.01 '*' 0.05 '.' 0.1 ' ' 1

Residual standard error: 1.441 on 949 degrees of freedom
  (110 observations deleted due to missingness)
Multiple R-squared:  0.1131,    Adjusted R-squared:  0.1122
F-statistic: 121.1 on 1 and 949 DF,  p-value: < 2.2e-16
```

COMMENT 2.32: *"Table 4 and 5 - The coefficients of determination are similar and do not point on a better implementation of the parameters in the upgraded metric."*

This comment presumably concerns a misunderstanding. Here, we examined the implementation of the soil variables in the metric. If residuals of the (normalized) productivity-metric associations are still related to soil variables already included in the metric, this could point at a suboptimal implementation of the variables. We clarified this better in the manuscript (Lines 22-23, 161-163, 213-216, 312-318, 340-346, 349-351, 466-469) and in the captions of Tables S9 and S16-19. We also refer to our answers on the comments on low $R^2$s (COMMENTS 2.2, 2.3, 2.4, 2.18, 2.22 and Section 4.5).

For upgraded metric 1, only few relationships between residuals of normalized productivity and the soil variables were significant (Table S16), while the opposite was true for the IIASA-metric (Table S9). As now discussed in the manuscript (Lines 483-487), residuals of normalized productivity were significantly related to both SOC and pH for upgraded metric 2, but the sign of the relationship depended on the normalization method (Fig. 3), i.e. methods 1 and 2 did not agree on whether partial scores for SOC and pH would be over- or underestimated. Analyses of the gradients generally confirmed the correct implementation of

soil variables in upgraded metric 1 (Table S18), whereas for metric 2, scores for high SOC might be overestimated (Table S19).

COMMENT 2.33: *"Figure 1 - I have problems understanding the legend of Figure 1."*

We rephrased the caption of Fig. 1.

TECHNICAL CORRECTIONS SUGGESTED BY REFEREE 2

COMMENT 2.34: *"12 - ...to test which combination of soil factors..."*

Done. (Line 12)

COMMENT 2.35: *"63 to 64 - rephrase this sentence to avoid the twofold use of "recent(ly)"*

Done. (Line 69-71)

COMMENT 2.36: *"96 - explain the abbreviations"*

Done. (Line 104-105)

COMMENT 2.37: *"118 - mass stock [kg m-2]; if really the mass is meant, the formula is wrong"*

Corrected. (Line 127)

COMMENT 2.38: *"157 - "we therefore we split" delete one "we"*

Done. (Line 173)

COMMENT 2.39: *"164 - "SOC" – stock or content? "Total N" – total N content? "N stock" – total N stock?"*

We clarified this in the manuscript. (Line 179-180)

COMMENT 2.40: *"167 - Name the software used."*

Done. (Line 181)

COMMENTS RAISED BY THE ASSOCIATE EDITOR

COMMENT 3.1: *"Reviewer 1 makes the excellent suggestion to stratify before you do your analyses. This reviewer specifically suggests to stratify according to wetness class before exploring soil fertility x productivity relations. I think that this is an important suggestion and deserves a more careful consideration than simply doing the same analysis for four different site wetness classes. I would even argue that you can only expect soil fertility to directly influence productivity if the site is well drained. Of course the IIASA approach does not consider this since agricultural sites are preferable located in better drained sites (unless it is rice), but with forests this is obviously different. Since you have a large database, you should consider to explore the suggestion to stratify further and explore whether you find stronger relations if you stratify according to soil type. At least you are then trying to find relations for sites with comparable soil forming factors."*

We followed the advice of referee 1 to stratify the main analyses by soil moisture, and we have also stratified by soil type (separately), as suggested here by the editor. In the revised manuscript, we added a section 'Identifying potentially confounding factors', in which we first present how soil properties and normalized productivity are related to soil moisture and soil type (Sections 2.3.1, 3.1 and 4.1). In the next sections we then present the stratified regression analyses (on soil variables + N deposition) and evaluations of the upgraded metrics. The regression outputs (Tables S3-S6 and S12-S15) generally confirmed the robustness of our results. We briefly discuss outcomes of the stratified analyses in the manuscript (e.g. Lines 306-308, 337-340, 487-488, 524-526).

COMMENT 3.2: *"Reviewer 1 also suggests to include N deposition in your analysis. Your write 'Nonetheless, if the editor so wishes, we are willing to test the influence of N deposition separately and present it in the revised manuscript`. My answer is: yes, I wish that you analyse whether N deposition affects productivity. It is something that you should have done after this suggestion was made and before writing your answer. Please do not forget to stratify here as well, since I predict that site wetness and other soil forming factors will interact with any N-deposition x productivity relation. There is a wealth of literature showing how strong the effect of N deposition is on forest productivity and just ignoring this by declaring it is out of the scope of this study is not helpful if N deposition affects productivity in your dataset."*

Both referee 1 and the editor emphasized that the role of N deposition should be discussed. We therefore investigated the influence of N deposition on soil variables, productivity, and on climate-normalized productivity (e.g. Lines 189-198, 280-289, 391-402). As explained on Lines 280-289 in the manuscript, N deposition indeed showed a strong positive relationship with productivity, because productivity increases in the north-south direction, as do N deposition, temperature and incident light. Whether N deposition correlated with normalized productivity depended on the normalization procedure (Fig. 3): for method 1 (which normalized productivity for species and climate [+ indirectly N deposition!]), there were mostly no positive associations with N deposition, while for method 2 (actual/attainable productivity of spruce), there was a strong positive relationship. This positive relationship followed logically from the increase of the ratio actual/attainable productivity from north to south Sweden. These issues, and their consequences, are discussed in the revised manuscript on Lines 391-402:

"Many studies have shown the strong influence of N deposition on forest productivity (e.g. Binkley and Hogberg, 2016; From et al., 2016). As expected, N deposition correlated to some extent with some of the soil variables considered in the present study, such as the total soil N stock and concentration (Fig. 4b). Furthermore, N deposition was strongly positively related to productivity. However, this effect of N deposition on productivity cannot be separated from the influence of climate and light, as all these factors increase together in the north-south direction. Nevertheless, we argue that for the goals of this study, i.e. investigating soil nutrient-productivity relationships across Sweden and developing a nutrient metric, the spatially varying N deposition is not problematic, since the normalization for climate and species according to method 1 (Fig. 3a) at the same time also removed the influence of the confounding N deposition on productivity. Accordingly, "Residual productivity" was generally not correlated with N deposition (Table S3). The response variable derived from method 2 (i.e. actual/attainable productivities for spruce - Fig. 3b) in contrast, correlated strongly with N deposition (Table S4), because both actual/attainable productivity and N deposition increased from north to south. Consequently, relationships between actual/attainable productivity and soil data for this method were unavoidably confounded by N deposition."

COMMENT 3.3: *"While I understand your arguments why you chose the IIASA approach, it would still be interesting if a purely statistical approach would lead to better results. Why don't you present this as well (and don't forget to stratify here as well)?"*

In our revised manuscript, we present results from multiple regression analyses (Table 3) and compare them with our upgraded metrics (Eqs. (11–13), (14–16) and (10)) in the discussion (Lines 494-505, also see COMMENT 1.3):

"Variation in normalized productivity explained by the upgraded metrics ($R^2 = 0.03–0.21$ and $R^2 = 0.06–0.18$) was similar to the variation explained by multiple regression equations ($R^2 = 0.18–0.22$) that contained the same (and more) soil variables than the metrics. The metrics can, however, easily be updated, while equations derived from multiple regressions would be of less use for building a nutrient availability metric because such an entirely empirical approach does not allow for later updates of parameters or model structures based on data from other ecosystems. In order to further enhance performance of the metrics, and to test to what extent they can describe variation in nutrient availability outside of Swedish conifer forests, additional datasets with productivity and soil information are needed. (…)"

COMMENT 3.4: *"Reviewer 2 points at the limitations of your study for developing a global metric. I agree with this reviewer that this is not possible but I do think that a metric for e.g. 'boreal forests on well drained soils' would already be a major improvement compared to the present situation. I think you should address this more specifically in your revised manuscript and acknowledge that a global metric is not realistic."*

Both referees and the editor expressed doubts on whether developing a global metric of nutrient availability is a realistic goal. We fully agree that this is an ambitious objective that might not be fully attainable (e.g. due to too large differences among vegetation types). However, no one has ever made a serious attempt. We thus consider it too early to "acknowledge that a global metric is not realistic", and have therefore kept the two main aims of our paper, which were to (i) detect important soil variables related to nutrient availability in Sweden, and (ii) start the development of a nutrient availability metric.

In any case, even if development of a global metric turns out too complicated, a valuable alternative could be to develop a metric for (boreal) forests only for example. In response to this comment and related COMMENTS 1.5, 2.1 and 2.8, we added a new Section 4.5, discussing the main challenges, and provide an outlook for how to deal with issues such as differential nutrient limitation, e.g.:

"A key challenge in the further development of a metric describing spatial variation in nutrient availability both within and outside the boreal biome is differential nutrient limitation. Eventually, we want to be able to compare

for example N-limited and P-limited systems. The original structure of the IIASA-metric, which was kept in our upgraded metrics, facilitates this by allowing the inclusion of multiple soil variables such as C:N (mainly relating to N availability), pH (among others a critical factor controlling P availability) and exchangeable bases in one single metric. In fact, the IIASA-metric is particularly useful in this regard, as it gives more weight to the soil factor with the lowest score. This corresponds to reality and enables accounting for the type of nutrient limitation. For instance, if C:N is high, indicating N limitation, the metric score will be substantially reduced by this high C:N, while at low C:N other limiting factors can dominate the metric score."

COMMENT 3.5: *"Reviewer 2 also makes the excellent suggestion that data quality (including comparability of data potentially analyzed with different chemical methods) may be an issue. In your answer you are not addressing this but it is still a very valid consideration. Were uniform methods used? How many laboratories were involved in the data? Is anything reported about detection limits? How are missing data treated? These are all valid questions that you have to address in your manuscript. You cannot expect people in dig into the original dataset to find this out."*

Based on this suggestion and the related COMMENT 2.2, we added a new Section 4.5 about "sources of uncertainty and future challenges" in the revised manuscript. We mention data quality there as a possible reason for the low $R^2$s, and explain that especially the normalization procedure also contributed substantially to the uncertainty (given the substantially higher $R^2$s for the local gradients where normalization was not needed).

As requested, we have also added more information about the forest plots, replicates etc. in the "Methods" section and in the caption and notes of Table 1. Finally, we refer to other papers that offer more detailed information about soil sampling and laboratory analyses in this table.

COMMENT 3.6: *"In you answer you refer twice to the need for using datasets where climate does not vary but nutrient availability does (e.g. local gradients). I was wondering why are you not exploring this? If you have a dataset with 2500 points I am sure that there are selections that can be considered local gradients. In my opinion you have the data to test this; it is not something that should be postponed to the future."*

We thank the editor for this excellent idea, and have now evaluated our upgraded metrics against data from five nutrient availability/productivity gradients in Sweden. These gradients were selected to cover a range in nutrient availability and productivity within the same/very similar climate. The results from these analyses indicated that tree productivity was significantly related to our metrics for all gradients (Tables 4 and 5). Moreover, the $R^2$s of these relationships were generally higher than was the case when considering the complete database ($R^2$s were up to 38 %). This suggests that the (necessary) procedure to normalize productivity for factors like climate in the previous analyses reduced the predictive power of the metrics. This last aspect also relates to referee 2's comment on the low $R^2$'s, which we have now addressed in a section on sources of uncertainty and future challenges in the discussion (Section 4.5).

---

## Author Response (AR2)

Dear editor,

We hereby submit the final version of our manuscript. We thank you and the referees for the useful comments and suggestions made throughout the peer-review process.

Below, we provide point-by-point responses to your comments and those of the referees. A marked-up version of the manuscript is available at https://www.dropbox.com/s/n63wjuwjbv0hnaf/VanSundertetal_BG_MarkedUp.7z?dl=0.

Sincerely,

Kevin Van Sundert, on behalf of the co-authors.

**OVERVIEW OF MAIN CHANGES**

Following the suggestions made by the referees and the Associate Editor, we now substantially reduced the emphasis on the formerly defined goal to ultimately develop one single, global nutrient metric. Instead, in the Abstract we introduce our study as a first example ("manual") of how a nutrient availability metric may be evaluated and adjusted with inventory data. We also changed the wording to indicate that the metrics are adjustments of the IIASA-metric rather than improvements/updates per se, as the latter would require testing also for agricultural systems for which the IIASA-metric was initially developed. Overall, we agreed with most referee comments and we have made the necessary changes in the manuscript. In few cases where we did not agree (e.g. on the removal of gradients from the manuscript – COMMENTS 3.4 and 3.37), we thoroughly explain why.

**POINT-BY-POINT REPLIES TO THE EDITOR AND REFEREE REVIEWS**

**COMMENTS RAISED BY THE ASSOCIATE EDITOR**

COMMENT E.1: *"I agree with reviewer #1 that you should adjust your abstract. My recommendation follows the recommendation of this reviewer: to write a short convincing story that clarifies the reality of developing this kind of indices."*

We fully agree with the editor and reviewer and have rewritten the abstract accordingly. We first explain the need for nutrient availability metrics and discuss the research questions (i.e. detection of important soil variables, evaluation of the IIASA-metric and adjustment of the metric to the Swedish data), and then emphasize that our paper may serve as a manual for further evaluations and adjustments of nutrient metrics in the future (instead of stressing that we aim to develop one single, global metric (e.g. Lines 11-15 and 31-32)):

"Here, we use a Swedish forest inventory database that contains soil data and tree growth data for > 2500 forests across Sweden to (i) test which combination of soil factors best explains variation in tree growth, (ii) evaluate an existing metric of constraints on nutrient availability, and (iii) adjust this metric for boreal forest data. With (iii), we thus aimed to provide an adjustable nutrient metric, applicable for Sweden and with potential for elaboration to other regions."

"This study thus shows for the first time how nutrient availability metrics can be evaluated and adjusted for a particular ecosystem type, using a large-scale database."

As a side note, we want to mention here that development of a nutrient availability metric for forests not just limited to the boreal biome may not be so unrealistic. We are currently evaluating the metrics from the present paper against European ICP forest data, and preliminary analyses show that it is possible to further adjust our metrics such that they describe variation in normalized productivity across ICP data, without losing explanatory power for the Swedish inventory.

COMMENT E.2: *"Many details in your manuscript sometimes distract from the main story. I recommend to critically look if all the details are necessary."*

Based on this comment by the Associate Editor and remarks made by Ref#3, we removed repetitions and details, especially in the Discussion of the manuscript. We now for example only explain the normalization procedure once in the main text (COMMENT 3.17), removed unnecessary details in the discussion on soil C:N, and also shortened the discussion regarding research question 2 (i.e. evaluation of the IIASA-metric – COMMENT 3.2).

COMMENT E.3: *"Reviewer #3 does not like the term 'upgraded'. While I see the point made by this reviewer, I think this is mainly semantics. You may consider to use the term 'adjusted' or some other term, but if you give me a good reason why the term 'upgraded' is preferable, I am open for it."*

Following the suggestion made by Ref#3, we replaced the term "upgraded" by "adjusted" everywhere in the manuscript (in the text, captions and in Figs. 2, 10, 11, S5 and S6). Ref#3 states that "adjusted" is more appropriate than "upgraded", because our changes to the metric would merely represent a reparameterization, applied to a different land use type (i.e. Swedish boreal forest instead of agricultural field). Although we not only reparameterized, but also improved performance of the metric by including the soil C:N ratio, we understand Ref#3's comment and thus followed his/her advice.

COMMENTS RAISED BY REFEREE 1

COMMENT 1.1: *"Abstract - The abstract has not been sufficiently updated to accommodate changes in the revised manuscript. With the exception of the detail regarding the local fertility gradients, it reads largely like the original. Include information that describes some of the variables that were included/evaluated in the various metrics (other than SOC and C:N). Include mention of the value of stratifying based on N deposition and upland/wetland landscapes. Some discussion how the study sorted out co-varying factors such as N deposition and N-S climate would strengthen the study's utility to researchers working elsewhere with multiple covarying factors."*

Based on the comments by both referees and the Associate Editor, we rewrote parts of and added more information to the Abstract where necessary.

We now mention all variables included in the IIASA-metric (Lines 21-23):

"This IIASA-metric requires information on soil properties that are indicative of nutrient availability (SOC, soil texture, total exchangeable bases - TEB and pH) and is based on theoretical considerations that are also generally valid for non-agricultural ecosystems."

Moreover, before explaining the results in the Abstract, we now explicitly mention that methods for dealing with covarying climate, N deposition and soil oxygen availability are used in the manuscript (Lines 15-16). We however preferred not to explain the stratification and

normalization procedures themselves in the abstract, because this would make us add too many details:

"While taking into account confounding factors such as climate, N deposition and soil oxygen availability, our analyses revealed that (…)"

In the final sentence, we now explain that our paper may serve as an example of how nutrient metrics can be evaluated and adjusted (Lines 31-32):

"This study thus shows for the first time how nutrient availability metrics can be evaluated and adjusted for a particular ecosystem type, using a large-scale database."

COMMENT 1.2: "*Line 10 - In my opinion, the contention that one metric is a goal remains rather over simplistic. Given the vastly different sorts of soils, climates, landscapes and production systems and management practices, a diversity of metrics seems more appropriate. I do agree that evaluation of the IIASA metric for forest systems in Northern Europe is worthwhile, but would suggest that the applicability of the study is more in showing how any particular metric is evaluated and modified, rather than how to develop a single metric.*"

Development of one single, globally applicable metric is a great challenge, and may even be impossible. We therefore carefully changed our wording in the Abstract and Conclusion of the manuscript, and now refer to metrics applicable at large spatial scales (e.g. Lines 8-11 and 554-558):

"**Abstract.** The availability of nutrients is one of the factors that regulate terrestrial carbon cycling and modify ecosystem responses to environmental changes. Nonetheless, nutrient availability is often overlooked in climate-carbon cycle studies because it depends on the interplay of various soil factors that would ideally be comprised into metrics applicable at large spatial scales. Such metrics do currently not exist."

"The current nutrient availability metrics were developed based on data from Swedish conifer forests only, and can therefore not as such be extrapolated outside the boreal biome. In order to find out if development of a metric that compares the nutrient status across sites also beyond the boreal biome is feasible, the adjusted metrics developed in this study now need to be validated and further modified based on other forests for which the necessary soil information is available. In a later stage, this approach can then be expanded to other ecosystem types."

We also explain in the Abstract that our paper may serve as an example of how nutrient metrics can be evaluated and adjusted (Lines 31-32):

"This study thus shows for the first time how nutrient availability metrics can be evaluated and adjusted for a particular ecosystem type, using a large-scale database."

COMMENT 1.3: "*Line 16 - Define how forest productivity is measured (MAI m3/ha/yr??).*"

We added the units in the Abstract and made clear that mean annual volume increment was used as a proxy of forest productivity (Lines 17-18).

COMMENT 1.4: "*Line 19 - Define IIASA.*"

Since the abbreviation "IIASA" is first mentioned in the Abstract, we now provide the full name there (Lines 20-21).

COMMENT 1.5: "*Line 21 - Insert 'forest' between normalized and productivity.*"

We inserted the word "forest" as requested (Lines 18 and 24).

COMMENT 1.6: "*Line 33 - Remove 'for example'*"

Done. (Line 35)

COMMENT 1.7: "*Line 41 - Remove 'thus'*"

Done. (Line 43)

COMMENT 1.8: *"Line 45 - Replace 'have previously described' with 'commonly use'"*

Done. (Line 47)

COMMENT 1.9: *"Line 45 - Insert hyphen 'fertility-related'"*

Done. (Line 47)

COMMENT 1.10: "*Line 74 - Oxygen availability and relevance to wetland vs upland soils should be included in the uncertainty/applicability/future challenges.*"

Done (Lines 513-517). We incorporated this in the paragraph on "Sources of uncertainty", since soil wetness and related oxygen availability increases variation in productivity, independent of nutrient availability (although we tried to account for this effect by performing stratified analyses):

"A more important source of uncertainty [than the number of replicates per data point] is probably the inevitable uncertainty related to the response variable, i.e. "climate-normalized" aboveground productivity. This includes uncertainty in the original productivity estimates (for which for example differences in management or disturbances likely increased variability) and additional variation caused by soil moisture effects on oxygen availability (which we accounted for by also performing analyses on split datasets). However, there is also uncertainty related to the normalization for climate: (…)"

COMMENT 1.11: *"Line 74-75 - Mention these [… environmental characteristics such as climate, rooting conditions and soil oxygen availability …] in the abstract"*

Done. Before explaining the results in the Abstract, we now explicitly mention that methods for dealing with covarying climate, N deposition and soil oxygen availability are used in the manuscript (Lines 15-16):

"While taking into account confounding factors such as climate, N deposition and soil oxygen availability, our analyses revealed that (…)"

COMMENT 1.12: "*Line 89 - Specify that it [the Swedish dataset] includes site information that allows comparison of the implications across soil oxygen/wetland, N deposition and climate gradients. Also integrate information regarding the local fertility gradients.*"

We now specified this in the paragraph (Lines 89-95):

"Such a unique dataset – that comprises > 2500 conifer forest plots and thus provides sufficient statistical power for an evaluation of the metric – is provided by the Swedish forest inventory service. Moreover, it contains additional variables of interest related to N availability, such as soil total N stock and concentration, and especially the soil C:N ratio, which we expected to be an important factor in explaining variation in nutrient availability. This large dataset also allows evaluating our country-scale findings against local gradients in nutrient availability that avoid confounding effects of covarying factors such as climate and N deposition."

COMMENT 1.13: *"Line 148 - Insert: N deposition, soil landscape (wetland vs upland)"*

Done (Lines 151-153):

"Forest productivity across Sweden depends not only on soil nutrient availability, but also on climate, soil wetness and N deposition. Before evaluating the metric, we removed the influence of climate on forest productivity ("PRE" in Fig. 2). The influence of soil moisture and N deposition are considered in further analyses (see section 2.3.1)."

COMMENT 1.14: *"Line 274 - State the obvious parallels between the analysis of soil moisture class and the wetland soil types (histosol/podzol)."*

We now more explicitly show the parallels between soil moisture and type (Lines 288-294):

"Soil properties not only differed among soil moisture classes, but also among soil types. Especially histosols and podzols could be distinguished from the other soils: histosols (which largely overlapped with the wet soil moisture classes) were characterized by a low $pH_{KCl}$, high SOC and soil C:N ratio, while podzols were sandy and had a low TEB stock (Fig. S4). Differences in normalized productivity among soil types were observed as well. Histosols in particular showed reduced productivities compared to other soil types (Fig. 6). Hence, the wetness of a site and its type of soil (partly in parallel with wetness) could confound observed patterns in productivity associated with the soil variables and are therefore taken into account in the further analyses and their interpretation."

COMMENT 1.15: *"Line 285 - Clearer less arbitrary wording would specify that "N deposition had a strong positive effect on normalized productivity based on method 2, but no significant effect with method 1."*

We rephrased the sentences (Lines 299-303):

"However, N deposition did generally not have a significant effect on productivity normalized with method 1 (i.e. residual productivity), while with method 2 (i.e. actual/attainable productivity), there was a strong positive relationship with N deposition. The increasing N deposition along the north-south gradient in Sweden (e.g. Olsson et al., 2009) should thus be kept in mind when interpreting effects of soil variables on productivity when normalized following method 2."

COMMENT 1.16: *"Line 297 - If the variables were not correlated, it does not matter what the trend was. This sentence should be removed."*

Done. (Line 311)

COMMENT 1.17: *"Line 349 - Remove 'y' in (Tables 4 and 5)"*

Done. (Line 365)

COMMENT 1.18: *"Line 385 - Good to connect soil moisture and soil type."*

We appreciate the referee's positive comment on this paragraph, where we link soil moisture and type.

COMMENT 1.19: *"Line 507 - Good place to mention the effects of variable forest management and age. Productivity relates to stand density (thinning operations) and stand age, and it would be good to acknowledge if/how those factors were addressed. The same goes for and more importantly for stand fertilization and soil drainage. Just review if this information was factored out (by site/data selection) or left in and part of unaccounted for variation."*

By taking the mean annual increment (MAI) over a rotation period as a proxy for aboveground productivity, the influence of stand age was largely removed. When we started the analyses for our paper, we formally tested if there was any remaining MAI-age relationship. This was not the case ($P > 0.05$).

Since we did not exclude any management type from the database, differences in management may explain part of the remaining variation. This information is included on Lines 513-517 of the revised manuscript:

"A more important source of uncertainty [than the number of replicates per data point] is probably the inevitable uncertainty related to the response variable, i.e. "climate-normalized" aboveground productivity. This includes uncertainty in the original productivity estimates (for which for example differences in management or disturbances likely increased variability) and additional variation caused by soil moisture effects on oxygen availability (which we accounted for by also performing analyses on split datasets). However, there is also uncertainty related to the normalization for climate: (…)"

Of the 23 Mha of productive forest in Sweden, only 44 000 ha was fertilized annually during 2006-2016 (https://www.slu.se/en/Collaborative-Centres-and-Projects/the-swedish-national-forest-inventory/forest-statistics/forest-statistics/). Stand fertilization is thus unlikely a major contributor to uncertainty in our database.

COMMENT 1.20: *"Fig. 4's Caption - Remove extra period after 'nitrogen deposition'."*

Done. (Fig. 4)

COMMENT 1.21: *"Fig. 4 - Check 'ln stock' between clay and TEB. Is that correct/complete? Add to caption?"*

Due to the alignment, it was unclear that ln(TEB stock) was meant here. We changed the alignment in Fig. 4b.

GENERAL COMMENTS RAISED BY REFEREE 3

COMMENT 3.1: *"The paper reads quite well, however, with some repetitions."*

Based on this remark and a comment by the Associate Editor, we removed some repetitions and details, especially in the Discussion. We now for example explain the normalization procedure only once in the main text (COMMENT 3.17), removed unnecessary details in the discussion on soil C:N, and also shortened the discussion regarding research question 2 (i.e. evaluation of the IIASA-metric – COMMENT 3.2).

COMMENT 3.2: *"Research question 2 seems to be a little bit overstressed as it should be clear from the dataset used, that the predominant part of the dataset is outside the range defined for the original IIASA-metric. This suggests, that the original IIASA-metric isn't applicable to the dataset. Thus, Q2 may be answered very shortly."*

We agree with the referee that the evaluation of the original IIASA-metric was not the main point of the study, and have therefore merged the two paragraphs on metric performance in the discussion into one (Lines 574-580):

"The IIASA-metric of constraints on nutrient availability does not clarify much variation in normalized productivity among Swedish forests. Moreover, SOC, soil texture, TEB and $pH_w$ were apparently not optimally implemented. A low performance of the IIASA-metric in its current form for the Swedish database was expected, as it was initially developed for evaluating (constraints on) the soil fertility of agricultural ecosystems, and the Swedish database contains variable values outside the ranges to which the metric is sensitive. Soil conditions of agro-ecosystems indeed greatly differ from the boreal forests investigated in the present study. Many Swedish forest soils are for instance coarse-textured, and in addition, the database contains wet-soil forests, while arable soils are typically not water saturated."

COMMENT 3.3: *"In my view, Q3 should be regarded as a reparametrization or adjustment of the original IIASA-metric, not as an improvement."*

We agree with the referee and have replaced the term "upgraded" by "adjusted" throughout the manuscript.

COMMENT 3.4: *"I'm not very supportive of the analyses of the new metrics based on selected gradients as these – as far as I understood - contain no additional information, because they are sub-datasets of the complete dataset. Because they are – in my view – systematically (not randomly) selected, a somewhat higher degree of explained variance is not very surprising. Therefore, I suggest to omit this part of the manuscript."*

Based on an earlier remark by the Associate Editor, we decided to search for local nutrient availability/productivity gradients in the database, which offer the advantage that no (potentially $R^2$ reducing) normalization for climate would be necessary. If we would have chosen data points close to each other entirely randomly, $R^2$s would, as the referee suggests, probably be lower than was the case in the current analyses. However, in such case, the data points would together not represent a nutrient availability gradient anymore, and consequently variation in productivity would likely be dominated to a larger extent by "noise" such as uncertainty in productivity estimates, management, … We therefore did not remove the gradient analyses from the manuscript.

COMMENT 3.5: *"References are sufficient and up to date. However, more references dealing with forest issues would be an asset."*

We thank the referee for the positive evaluation of our use of references. Regarding citations on forest issues, we now included the studies by Laubhann et al. (2009) and de Vries et al. (2014) suggested by the referee, and in addition refer to more papers on boreal, European and global analyses of forest data, e.g.:

"Nutrients determine structure and functioning at all levels of biological organization. The availability of mineral elements influences plant growth (von Liebig, 1840), patterns of biodiversity (Fraser et al., 2015) and ecosystem processes (e.g. Janssens et al., 2010; Vicca et al., 2012; Fernández-Martínez et al., 2014). Moreover, nutrient availability can modify ecosystem responses to global atmospheric and climatic changes, such as nitrogen (N) deposition (**Nohrstedt, 2001; Hyvönen et al., 2008; Vadeboncoeur, 2010**), (…)" (Lines 34-38)

"Numerous studies have shown the strong influence of N deposition on forest productivity (e.g. **Laubhann et al., 2009; Solberg et al., 2009; de Vries et al., 2014**; Binkley and Högberg, 2016; **Wang et al., 2017**). Although N deposition can influence the soil properties considered in our analyses, it may also influence productivity without immediate changes in these soil properties (i.e. there is a time lag - Novotny et al., 2015). (…)" (Lines 192-195)

"Soil factors other than the soil C:N ratio and SOC either exhibited only a marginal influence on normalized productivity or their effect depended on the approach (Table 2). N stocks could explain variation across both methods, but their explanatory power was rather modest for method 1. We anticipate that if we aim to develop metrics applicable beyond the boreal biome, including N stock will be of limited value, as this variable is only loosely related to N availability (**Högberg et al., 2017**)." (Lines 446-449)

SPECIFIC COMMENTS RAISED BY REFEREE 3

COMMENT 3.6: *"Rows 24, 28, 91, 99, … - I wonder if 'upgraded' is the appropriate term here as it is more an adjustment from one land use type to another."*

We replaced the term "upgraded" by "adjusted" throughout the manuscript.

COMMENT 3.7: *"Row 35 - From et al. 2016 isn't explicitly a study on ecosystems responses to N deposition. It rather studies the effects of enhanced TN on productivity, which is later addressed in the manuscript. Probably, Laubhann et al. 2009 FORECO 258, de Vries et a. 2014 Curr Opinion Environ Sust 9-10 or related papers are more appropriate here."*

In response to this comment, we have replaced the citation to From et al. by Nohrstedt (2001), Hyvönen et al. (2008) and Vadeboncoeur (2010) to explain that ecosystem responses to N deposition depend on the background nutrient status (Lines 36-38). Even though these studies do not explicitly report on effects of N deposition, they refer to experimental N(PK) additions, which obviously show analogies with N deposition. These papers explain more clearly than From et al. (2016) that effects on tree growth, soil respiration and C sequestration are function of initial site fertility. To the best of our knowledge, the influence of background nutrient status on N deposition effects has never been assessed.

We now refer to Laubhann et al. (2009), Solberg et al. (2009), de Vries et al. (2014) and Wang et al. (2017) in the Methods (Lines 192-193) and Discussion (Lines 415-416) sections.

COMMENT 3.8: *"Row 70 - Abbreviations (TEB) need to be explained only at its first appearance."*

With this comment, the referee pointed to the line where both the full name and abbreviation of IIASA (not TEB) were mentioned for the first time in the Introduction. Since IIASA was already mentioned in the Abstract before, we removed the full name in Line 72. With the changes in our manuscript, TEB now appears for the first time in the Abstract too (Line 22), hence the full name + abbreviation is now given there.

COMMENT 3.9: *"Row 81-82 - Please explain, why the 'realism of the metric' is increased?"*

The IIASA-metric and our adjusted metrics give more weight to the soil variable with the lowest score. Following Liebig's Law of the Minimum, which states that it is the most limiting element that determines productivity, this way of weighing should improve realism of the metric. We now added an example why this is realistic to the paragraph (Lines 79-84):

"The species-specific score of the metric depends on four measurable soil variables, related to soil fertility: SOC (%), soil texture, TEB (cmol+ kg-1 dw) and pH measured in water (pHw). The metric score combines the scores of each of these four attributes (provided in a look-up table), but giving more weight to the attribute with the lowest score. Together with the non-linear relationships (e.g. for pH and SOC - see Methods), this increases the realism of the metric (cf. Liebig's Law of the Minimum (von Liebig, 1840); e.g. at optimal pH, the limiting effect of low SOC on plant growth will be stronger than in soils with very low or high pH where plant growth becomes more likely to be P-limited)."

In the context of differential nutrient limitation, we also provide another example with soil C:N in the discussion (Lines 539-541):

"This [giving more weight to the variable with the lowest score] corresponds to reality and enables accounting for the type of nutrient limitation. For instance, if soil C:N is high, indicating N limitation, the metric score will be substantially reduced by this high C:N, while at low C:N other limiting factors can dominate the metric score."

COMMENT 3.10: *"Row 89 - ... contains an additional variables ...' -> '... contains additional variables ..."*

Done. (Line 91)

COMMENT 3.11: *'Row 98 - ... in Sweden, ...' -> '... in Swedish forest soils, ...'*

The final part of the last introductory paragraph was edited and does not include the expression "in Sweden" anymore.

COMMENT 3.12: *"Row 114-117 - Why not use ERA-interim data for the determination of temperature sums (TSUM) to be consistent with precipitation data? As TSUM is determined here only from latitude, longitude and elevation using a parametrization obviously based on data prior to 1983, the consistency with other climate data is questionable."*

Prior to our further analyses such as normalization of productivity for climate etc., we tested which combination of climatological variables best described spatial variation in productivity. In the ERA-interim data, mean annual temperature (MAT) was available besides precipitation, but the regression model including TSUM instead of MAT performed better (i.e. it had a lower cross-validation based mean-squared error).

Odin et al. (1983) provide the original equation to calculate TSUM in Sweden based on latitude and elevation, but the actual equation given in our manuscript was based on a recent reparametrization. We now explicitly mention this in the text (Lines 113-116):

"To quantify the influence of climate on productivity across Sweden (question 1), we first determined the annual growing season temperature sum (TSUM) following a recently reparameterized version of the equation given in Odin et al. (1983), available on www.kunskapdirekt.se: (…)"

COMMENT 3.13: *"Row 119 - Please use consistent nomenclature throughout the manuscript (e.g. 'total nitrogen' or 'total N'; 'soil C.N ratio' or 'C:N ratio') and define abbreviations at its first appearance."*

Done. We now consistently refer to soil organic carbon concentration as SOC, to total N concentration as TN, to soil carbon to nitrogen ratio as soil C:N ratio etc.

COMMENT 3.14: *"Row 120 - Is the organic layer included in the depth intervals? Please clarify."*

Yes, the organic layer is included in the intervals. We now clarified this better in the text (Lines 122-124):

"(…) to values representative of the upper 10 cm (i.e. the 0–10 cm layer) and the upper 20 cm (i.e. the 0–20 cm layer) of the soil, including the organic layer."

COMMENT 3.15: *"Row 122 - Equation 3: Is 'humus stock' and 'humus depth' here synonymous to 'organic layer stock' and 'organic layer depth', respectively?"*

Yes. In order to be consistent with terminology across the manuscript, we renamed "humus" to "organic layer" in the equation (Line 125)

$$BD_{organic\ horizon}\ [kg\ m^{-3}] = \frac{organic\ layer\ stock\ [kg/m^2]}{organic\ layer\ depth\ [m]}\ , \tag{3}$$

and in Table 1.

COMMENT 3.16: *"Row 152 - '... with ArcGIS (ESRI, 2011) ...' seems not necessary here."*

We decided to keep the citation to correctly refer to the software used (Lines 157, 199 and 241).

COMMENT 3.17: *"Row 173-176 - This is a repetition of row 150 to 154 and row 168 to 169."*

The referee correctly points to information that was unnecessarily provided twice in the manuscript: the two methods for normalizing productivity were already introduced in Section 2.1 (General approach). In the following Section (Data analyses), we therefore shortened the paragraph to focus better on the main new point, i.e. that due to heteroscedasticity, we split the database by region for method 1, but not for method 2 (Lines 173-178):

"As explained in the paragraphs above, productivity was normalized using two methods. Method 1 considers the residuals to reflect deviations in productivity imposed by spatial variation in nutrient availability and in the absence of climate effects. However, residuals deviated more strongly from zero towards the warmer south (Fig. 3a), thus causing heteroscedasticity and a potential bias in the further analyses if not properly accounted for. For further analyses, we therefore split the database into three TSUM groups (north, middle and south; Fig. 3a). For method 2, considering the ratio actual/attainable productivity, this separation of different regions was not required."

COMMENT 3.18: *"Row 179 - '... (SOC concentration, total ...' -> '... (SOC, total ...'."*

As explained in COMMENT 3.13, we now consistently refer to soil organic carbon concentration as SOC, to total N concentration as TN, to soil carbon to nitrogen ratio as soil C:N etc.

COMMENT 3.19: "*Row 183 - 'Soil type' according to which classification?"*

Soil type was classified following the World Reference Base for Soil Resources. We now mention this in the paragraph (Line 190), in addition to the information added earlier under Table 1:

"We therefore tested if the selected soil variables and normalized productivity differed among soil moisture classes (…) and the most common World Reference Base for Soil Resources based soil types (histosols, gleysols, regosols, leptosols and podzols) …"

COMMENT 3.20: *"Row 186 - 'Soil moisture classes' according to which classification?"*

As specified under Table 1, soil moisture was determined in the field based on various indicators (e.g. groundwater depth, moisture at the surface, ground vegetation, elevated tree trunks, ....). The classification, specific for the Swedish inventories, is representative of the average moisture conditions during the growing season. We now mention that moisture classification is based on field indicators in the manuscript, and provide the references (Lines 187-189):

"We therefore tested if the selected soil variables and normalized productivity differed among soil moisture classes (dry, fresh, fresh-moist and moist, as available from the database and derived from a combination of indicators such as groundwater depth – Olsson, 1999; Olsson et al., 2009)"

COMMENT 3.21: *"Row 187-188 - Which classification was used here for soil type? If WRB is meant here (Table 1), the specifiers used (e.g. 'peaty', 'wetland') aren't valid."*

The referee correctly assumes that WRB was used for classification, as now explained in the text and under Table 1. We added this information to the paragraph (see COMMENT 3.19).

Although the specifiers we added to the soil types (peaty histosols, wetland gleysols, weakly developed regosols, gravel-rich leptosols and sandy podzols) are roughly correct, we agree that they were defined quite bluntly when WRB is used for classification. We therefore agreed with the referee's request to remove the specifiers, but instead stressed the link between soil type and wetness better in the Results (Lines 288-294) and Discussion (Lines 406-414), as demanded by Ref#1:

"Soil properties not only differed among soil moisture classes, but also among soil types. Especially histosols and podzols could be distinguished from the other soils: histosols (which largely overlapped with the wet soil moisture classes) were characterized by a low $pH_{KCl}$, high SOC and soil C:N ratio, while podzols were sandy and had a low TEB stock (Fig. S4). Differences in normalized productivity among soil types were observed as well. Histosols in particular showed reduced productivities compared to other soil types (Fig. 6). Hence, the wetness of a site and its type of soil (partly in parallel with wetness) could confound observed patterns in productivity associated with the soil variables and are therefore taken into account in the further analyses and their interpretation."

"In the same way as for soil moisture, stratification by soil type might help in resolving nutrient-productivity relationships. Soil properties and productivity differed among the five most common soil types in the database (i.e. histosols, gleysols, regosols, leptosols and podzols - Fig. S4). To some extent, these differences among soil types overlapped with these observed for soil moisture classes (e.g. wet histosols had the highest SOC, soil C:N and the lowest productivity), but additional patterns emerged as well (e.g. podzols had a particularly low TEB stock). Although actual differences in nutrient availability among soil types will in part underlie the variations in productivity, other factors related to soil type (e.g. wetness, soil depth or the rooting environment) may also influence productivity (Binkley and Hart, 1989). The main analyses of the current study were therefore stratified by both soil moisture and type to test the robustness of associations between nutrient related soil properties and normalized productivity."

COMMENT 3.22: *"Row 179-180 - What is the rationale behind using total N concentration and total N stock as different variables, but only SOC concentration? First appearance of clay and sand fraction. Were sand and clay fractions derived from texture designation in the field or from lab analyses?"*

The idea of not only using TN (concentration), but also the amount of N per m² (stock) is that the latter may better represent the N present per unit soil volume. This matters for example when comparing SOM-rich soils with low bulk density with SOM-poor soils with high bulk density. In such case, higher TN might be observed for the former, while N stock is the same for both. For SOC, on the other hand, this is less of a problem, because soils with high SOC are roughly also the ones with the highest carbon stocks.

Originally, soil texture was present in the database as texture classes. In an earlier version of the database, however, these were approximately converted to percentages sand, silt and clay to facilitate analyses such as the ones performed in our study. We now mention this under Table 1 (Line 724):

"c In an earlier version of the database, percentages of sand, silt and clay were approximated from field based soil texture class."

COMMENT 3.23: "*Row 194-195 - Please amend the time period for which N deposition was available.*"

We used the most recent deposition data available for our analyses (i.e. from 2015). We added this information to our manuscript (Lines 196-199):

"To verify whether N deposition confounded our analyses, we extracted N deposition data of 2015 from a map available at http://www.smhi.se/sgn0102/miljoovervakning/kartvisare.php?lager=15DTOT_NOY___ (Swedish Meteorological and Hydrological Institute, 2018), using the ArcGIS software (ESRI, 2011)."

COMMENT 3.24: "*Row 204-206 - I don't understand the sentence (what is meant with 'each time'?). Please reformulate!*"

Like AIC(c), cross-validation can be used as a method to perform model selection. In the model selection procedure, we started with a model containing all explanatory variables, e.g.

$$Y \sim a * X_1 + b * X_2 + c * X_3 + d * X_4 + e * X_5 \qquad \text{(model 1)}$$

Then, the term with the lowest significance was removed, e.g. if this was c, then the model became:

$$Y \sim a * X_1 + b * X_2 + d * X_4 + e * X_5 \qquad \text{(model 2)}$$

To verify is model 2 was indeed better than model 1, we then compared the cross-validation based mean squared errors (mse) of the two models. If mse of model 2 < mse of model 1, we continued the procedure with model 2. With "each time", we thus meant that the selection procedure was carried out multiple times on different models, until the model with minimal mse was found. To make this clearer, we reformulated this in our manuscript (Lines 207-210):

"Starting from the full model containing all explanatory variables, the least significant term was removed, resulting in a simplified model. Performance of the full and simplified model was then compared using the mean squared error (mse), based on cross-validation (package DAAG - Maindonald and Braun, 2015). We repeated this model simplification procedure until mse stopped decreasing."

COMMENT 3.25: *"Row 233-245 - The selection of the 'local gradients' should be specified correctly. At row 235 and 239 it is claimed, that the local gradients were selected randomly, whereas at row 239 it is mentioned, that these were manually selected (for which it is hard to imagine that this is possible randomly). Moreover, at row 242 to 243, it is mentioned that 'we searched specifically for clear soil C:N gradients', again indicating a systematic selection procedure."*

We thank the reviewer for pointing out this apparent contradiction. We indeed manually (systematically) looked for gradients in soil moisture, TEB, productivity (and soil C:N) in ArcGIS, but within each "pool of potential gradients", we did not have a specific reason to prefer one gradient over another, which is why we wrote they were selected "randomly". In order to avoid confusion, we removed the word "random" in the revised manuscript, including Table and Figure captions (e.g. Lines 29, 237, 493, … and captions of Tables 4 and 5, Fig. S2).

COMMENT 3.26: *"Row 247-248 - Which 'additional tests' from which 'packages'?"*

This information was redundant and was therefore removed from the revised manuscript (the packages were explained in the following sentences - Lines 249-253):

"Moreover, for all regressions, potential non-linearities were detected with histograms of all variables' distributions and generalized additive models from the mgcv package (Wood, 2006). Data were log-transformed if their distribution was right-skewed, while polynomial (e.g. quadratic) functions were included in the model selection procedure where the general additive models suggested non-linear patterns. The variance inflation factor (package car - Fox and Weisberg, 2011) assessed possible multicollinearity."

COMMENT 3.27: *"Row 260-261 - 'TN' wasn't explained before."*

As explained in COMMENT 3.13, we now consistently refer to soil organic carbon concentration as SOC, to total N concentration as TN, to soil carbon to nitrogen ratio as soil C:N etc. We now provide the full name of TN at first appearance in the "Methods" section (Lines 121-124), and in the caption of (a.o.) Table 1, which introduces all variables:

"In order to facilitate between-site comparisons and to allow calculating the nutrient availability metric, we converted the soil measurements (SOC, soil texture, TEB, $pH_w$, $pH_{KCl}$, total nitrogen concentration (TN) and soil C:N ratio) taken per horizon to values representative of the upper 10 cm (i.e. the 0–10 cm layer) and the upper 20 cm (i.e. the 0–20 cm layer) of the soil, including the organic layer."

COMMENT 3.28: *"Row 262-263 - Is 'total N' the same as 'TN'?"*

Yes, we meant TN (total nitrogen concentration) with "total N" here. We replaced it by TN to be consistent across the manuscript.

COMMENT 3.29: *"Row 294 - Fig. 7b appears prior to Fig. 7a. Please change the numbering of the respective Figures accordingly."*

Fig. 7a/b represents the relationship between residual productivity and SOC/soil C:N. In an analogous way, Fig. 8a/b shows the relationship between actual/attainable productivity and SOC/soil C:N. In order to maintain this analogy between Figs. 7 and 8, we preferred to change the order of the sentences referring to SOC and soil C:N, rather than switching Figs 7a and 7b (Lines 307-310):

"For both SOC (Fig. 7a) and $pH_{KCl}$, the relationship with normalized productivity showed an optimum (i.e. an empirical quadratic relationship fitted better than a linear model). Normalized productivity was significantly

negatively correlated with the soil C:N ratio (Fig. 7b), for which the effect became more pronounced towards the south (i.e. slopes and R²s increased; $F_{2,2274} = 34.23$, $P < 0.01$).”

COMMENT 3.30: “*Row 295-296 + 303 - The relationship between SOC (pH) and normalized productivity isn't necessarily quadratic by nature (I guess, it only gives a better fit than linear). Please change wording. // Please use other wording for explanation of the functional type of empirical relations (i.e. '… was not quadratic…').*”

We agree with the referee that the better fitting quadratic relationships of SOC and pH with normalized productivity are entirely empirical, and do not imply that the relationships are quadratic in reality. However, both show an optimum, and these optima do make sense for theoretical reasons discussed in the manuscript. As suggested by the referee, we rephrased earlier mentions to quadratic relationships and instead report that the variables showed an optimum effect on normalized productivity (e.g. Lines 307-308, 316-317 and 435):

“For both SOC (Fig. 7a) and $pH_{KCl}$, the relationship with normalized productivity showed an optimum (i.e. an empirical quadratic relationship fitted better than a linear model).”

“However, the function for $pH_{KCl}$ did not show an optimum, but was linear with a significantly positive slope (Table 2).”

“The relationship of ln SOC with normalized productivity, which showed an optimum (Figs. 7a and 8a), (…)”

COMMENT 3.31: “*Row 303-305 - However, more than these two variables were included in the multiple regressions (Table 3).*”

We now made this clearer the manuscript (Lines 317-320):

“In summary, SOC and the soil C:N ratio were the only soil factors that showed a similar trend according both methods with an $R^2$ of at least a few percent, and were thus included in the multiple regression models for both methods 1 and 2 (these models also included other variables resulting from the stepwise regression analysis; Table 3).”

COMMENT 3.32: “*Row 349 - Is Table 5 meant here instead of Y5?*”

Done. (Line 365)

COMMENT 3.33: “*Row 383-384 - the specifiers of the soil type (e.g. peaty, wetland) don't appear in Fig. S4.*”

In accordance with the referee's earlier COMMENT 3.21, we removed the specifiers from the main text.

COMMENT 3.34: “*Row 395-398 - However, as N deposition is affecting nutrient availability (which is demonstrated by the positive correlation to SOC and TN – Fig. 4), also the effect of – in this case – SOC may be partly removed by the normalization procedure (method 1).*”

We agree with the referee that some minor influence of SOC on productivity may be removed by normalizing, because by removing climate influence following method 1, also N deposition influence is largely removed, and SOC correlates with N deposition (Pearson's $r = 0.30$). However, there is considerable variation in SOC irrespective of N deposition, such that the main effect of SOC on productivity nevertheless remains.

COMMENT 3.35: *"Row 466-474 - The 'not optimal implementation' of SOC (and other variables) is not very surprising as the database contains a substantial part outside the range of the original IIASA-metric. Extrapolation beyond the range of empirical functions should usually be avoided."*

IIASA does not explicitly provide information on what soil variable ranges the metric can be applied, but we agree with the referee that data that fall outside of the original metric's ranges explain in part why the variables seem suboptimally implemented. We therefore added this to the paragraph (Lines 474-480):

"The IIASA-metric of constraints on nutrient availability does not clarify much variation in normalized productivity among Swedish forests. Moreover, SOC, soil texture, TEB and pHw were apparently not optimally implemented. A low performance of the IIASA-metric in its current form for the Swedish database was expected, as it was initially developed for evaluating (constraints on) the soil fertility of agricultural ecosystems, and the Swedish database contains variable values outside the ranges to which the metric is sensitive. Soil conditions of agro-ecosystems indeed greatly differ from the boreal forests investigated in the present study. Many Swedish forest soils are for instance coarse-textured, and in addition, the database contains wet-soil forests, while arable soils are typically not water saturated."

COMMENT 3.36: *"Row 476-481 - Basically, this is a reparametrization of the original IIASA-metric."*

We agree with the referee that the altered equations for SOC and pH are actually reparametrizations. However, we also included the soil C:N ratio, which is new to the metric and which, as we demonstrate, is an important indicator of nutrient availability for forests in Sweden. Based on this and other comments of the referee, we have reworded this sentence to indicate that the metric was "adjusted" instead of "improved" (Lines 482-484):

"Based on results of the analyses for question 1, the nutrient availability metric was adjusted by i) including an empirical optimum in the influence of SOC on normalized productivity, and ii) including soil C:N, thus more explicitly incorporating the availability of N."

COMMENT 3.37: *"Row 488-489 + 489-491 – This [the fact that the adjusted metrics described variation in productivity for gradients] is again not surprising as the gradients are sub-datasets of the complete dataset. // I fear, the higher R²s are most likely because they were not randomly selected. If they were randomly selected, I would guess that the probability of higher or lower R²s is almost similar."*

Based on an earlier remark by the Associate Editor, we decided to search for local nutrient availability/productivity gradients in the database, which offer the advantage that no (potentially $R^2$ reducing) normalization for climate would be necessary. If we would have chosen data points close to each other entirely randomly, $R^2$s would, as the referee suggests, probably be lower than was the case in the current analyses. However, in such case, the data points would together not represent a nutrient availability gradient anymore, and consequently variation in productivity would likely be dominated to a larger extent by "noise" such as uncertainty in productivity estimates, management, … We therefore did not remove the gradient analyses from the manuscript.

COMMENT 3.38: *"Row 496-498 - As far as I understood, both approaches are entirely empirical. Moreover, I would expect that a multiple regression approach can be updated to datasets from other ecosystems as well."*

We agree with the referee that the adjusted metrics are definitely not mechanistic models, but rather a weighted set of empirical regression equations. Reparametrizing the metrics however remains more obvious than doing the same for multiple regressions, especially with regards to differential nutrient limitation, as also mentioned under COMMENT 3.9 and in the Discussion (Lines 539-541):

"This [giving more weight to the variable with the lowest score] corresponds to reality and enables accounting for the type of nutrient limitation. For instance, if soil C:N is high, indicating N limitation, the metric score will be substantially reduced by this high C:N, while at low C:N other limiting factors can dominate the metric score."

With multiple regression equations, the underlying model structure (including non-linear functions and interactions) would likely completely shift depending on the type of nutrient limitation. Consequently, this would require more than a simple reparametrization or addition of a variable not yet included. To clarify the need for the more complicated structure compared to regressions, we have included a few sentences on this in the discussion (Lines 498-506):

"Variation in normalized productivity explained by the adjusted metrics ($R^2$ = 0.03–0.21 and $R^2$ = 0.06–0.18) was similar to the variation explained by multiple regression equations ($R^2$ = 0.18–0.22) that contained the same (and more) soil variables than the metrics. The metrics, however, have the advantage that they can be updated more easily than equations from multiple regressions, especially if additional soil parameters need to be included for other ecosystems. Moreover, the interaction effect – with the highest weight for the least optimal soil parameter – cannot be mimicked with a multiple regression approach. In order to further adjust the metrics, and to test to what extent they can already describe variation in nutrient availability outside of Swedish conifer forests, additional datasets with productivity and soil information are needed. Such datasets include large-scale inventories such as the one considered in the present study, but also local gradients and nutrient manipulation experiments. The latter two have lower generalizability, but offer the advantage that normalization for climate is not needed."

COMMENT 3.39: *"Row 518-520 - What are the main sources of uncertainty in the attainable productivity estimates?"*

Attainable productivity was extracted from a map, provided by Bergh et al. (2005). The authors estimated attainable productivity for Norway spruce in Sweden by combining climate/environmental information with an experimentally derived relationship between productivity and intercepted radiation. This relationship was based on only a few nutrient optimization experiments. Hence, large uncertainty can be expected in the national generalization. We shortly added this information to the manuscript (Lines 517-526):

"However, there is also uncertainty related to the normalization for climate: by taking residuals of the productivity vs climate regression model (method 1), we for instance unintentionally not only removed the direct effect of climate on productivity, but also its indirect effect through nutrient availability. Normalized productivity based on this method thus mainly represents productivity as influenced by regional variation in nutrient availability. The approach taking actual/attainable productivity as a response variable (method 2) does not suffer from this issue, but there the estimates of attainable productivity come with a high uncertainty, as they were based on only limited experimental data to establish a relationship between productivity and intercepted radiation. As a consequence, the low $R^2$ values are partly due to shortcomings of the normalization procedure that can only be overcome by using datasets where climate does not vary but nutrient availability does. Such datasets are provided by local gradients, such as the five local nutrient availability gradients that we selected from our database for additional evaluation of our adjusted metrics."

COMMENT 3.40: *"Figure 2 - '... vs climate and species; ...'-> '... vs climate and species (spp.); ...'."*

Done.